

# Generalized hydrodynamics in complete box-ball system for $U_q(\widehat{sl}_n)$

**Atsuo Kuniba[1]⋆, Grégoire Misguich[2]† and Vincent Pasquier[2]‡**

**1** Institute of Physics, University of Tokyo, Komaba, Tokyo 153-8902, Japan
**2** Institut de Physique Théorique, Université Paris Saclay, CEA,
CNRS UMR 3681, 91191 Gif-sur-Yvette, France

⋆ atsuo.s.kuniba@gmail.com, † gregoire.misguich@ipht.fr,
‡ vincent.pasquier@ipht.fr

## Abstract

We introduce the complete box-ball system (cBBS), which is an integrable cellular automaton on 1D lattice associated with the quantum group $U_q(\widehat{sl}_n)$. Compared with the conventional $(n-1)$-color BBS, it enjoys a remarkable simplification that scattering of solitons is totally diagonal. We also submit the cBBS to randomized initial conditions and study its non-equilibrium behavior by thermodynamic Bethe ansatz and generalized hydrodynamics. Excellent agreement is demonstrated between theoretical predictions and numerical simulation on the density plateaux generated from domain wall initial conditions including their diffusive broadening.



# 1 Introduction

The box-ball system (BBS) [1] and its generalizations are paradigm examples of integrable cellular automata on 1D lattice connected to Yang-Baxter solvable vertex models [2], Bethe ansatz [3], crystal base theory of quantum groups at $q = 0$ [4], ultradiscretization and tropical geometry etc. See for example the review [5] and the references therein.

In this paper we introduce a new version of BBS associated with $U_q(\widehat{sl}_n)$, which we call the *complete* box-ball system (cBBS). It possesses a number of distinguished features compared with the conventional BBS with $(n-1)$ kinds of balls. We also consider a protocol where the cBBS is initially prepared in some random initial conditions corresponding to two different ball densities in the left and in the right halves. The associated non-equilibrium behavior is then studied by thermodynamic Bethe ansatz [6,7] and generalized hydrodynamics [8–10]. The results generalize the earlier work [11] for $n = 2$, where the cBBS itself reduces to the original one [1].

Let $B^{k,l}$ be the set of semistandard tableaux on the $k \times l$ rectangular Young diagram over the alphabet $\{1, 2, \ldots, n\}$ [12,13]. It is a labelling set of the basis of the irreducible $sl_n$ module corresponding to the mentioned Young diagram. By the definition only $1 \le k < n$ is relevant and the simplest $B^{1,1}$ is identified with the set $\{1, 2, \ldots, n\}$. A typical or conventionally most studied BBS with $(n-1)$ kinds of balls [14] is a dynamical system on

$$\cdots \otimes B^{1,1} \otimes B^{1,1} \otimes B^{1,1} \otimes \cdots, \tag{1.1}$$

where $\otimes$ can just be regarded as the product of sets in this paper. One interprets it as an array of boxes which is empty for $1 \in B^{1,1}$ or contains a color $a$ ball for $a = 2, 3, \ldots, n \in B^{1,1}$. All the distant boxes are assumed to be $1 = $ empty. By using the crystal theory of $U_q(\widehat{sl}_n)$, one can formulate an integrable dynamics on such states yielding solitons [15, 16].[1] Here is

---

[1]In this paper we will not use the crystal theory of $U_q(\widehat{sl}_n)$ extensively since it is not the main theme, and only

an example with $n = 4$, where $t$ is the discrete time variable (the symbol $\otimes$ is omitted for simplicity):

$t = 0$:  111122221111113321114311111111111111111111111111111

$t = 1$:  111111112222111133211143111111111111111111111111111

$t = 2$:  111111111111222211113321143111111111111111111111111

$t = 3$:  111111111111111222211133243111111111111111111111111

$t = 4$:  111111111111111111112222113243311111111111111111111

$t = 5$:  111111111111111111111111222132243311111111111111111

$t = 6$:  111111111111111111111111111221132243321111111111111

$t = 7$:  111111111111111111111111111111122113221433211111111

$t = 8$:  111111111111111111111111111111111221111322114332111111111

$t = 9$:  111111111111111111111111111111111111221111132211143321111

The incoming solitons $2222$, $332$ and $43$ get close and undergo messy collisions, but eventually they come back nicely in the very original amplitude (or size) $2, 3$ and $4$. Solitons possess internal degrees of freedom. They have changed nontrivially as $2222 \times 332 \times 43 \to 22 \times 322 \times 4332$ like the interchange of quarks in hadron collisions. Thus, the scattering in this model is *non-diagonal* meaning that the internal labels of solitons are not conserved and change nontrivially.

The cBBS we propose and study in this paper is obtained, among other things, by replacing (1.1) with

$$\cdots \otimes B \otimes B \otimes B \otimes \cdots, \qquad \text{with} \qquad B = B^{1,1} \otimes B^{2,1} \otimes \cdots \otimes B^{n-1,1}. \qquad (1.2)$$

Apparently it looks more involved than (1.1) since now each site has a larger internal structure $B$ than $B^{1,1}$ for $n \geq 3$. However, it turns out to enjoy much simpler and elegant features than the conventional BBS as we explain below.

First, solitons in cBBS can be labeled just with color $a \in \{1, 2, \ldots, n-1\}$ and amplitude $i \in \mathbb{Z}_{\geq 1}$ as $S_i^{(a)}$. Moreover they exhibit totally *diagonal* scattering $S_i^{(a)} \times S_j^{(b)} \to S_j^{(b)} \times S_i^{(a)}$ whose nontrivial effect is integrated into a phase shift on their asymptotic trajectories. This is a drastic simplification compared with the conventional BBS. For example in the notation explained in Sec. 2, scattering of $S_3^{(1)} = {}_{22}^{\ 1} \otimes {}_{22}^{\ 1} \otimes {}_{22}^{\ 1}$ and $S_4^{(2)} = {}_{13}^{\ 1} \otimes {}_{13}^{\ 1} \otimes {}_{13}^{\ 1} \otimes {}_{13}^{\ 1}$ for $n = 3$ under a time evolution $T_l^{(1)}$ with $l \geq 3$ looks as follows (a bank signifies the vacuum background ${}_{12}^{\ 1}$):

$$
\begin{array}{cccccccccccccccccccccc}
, & {}^{1}_{22}, & {}^{1}_{22}, & {}^{1}_{22}, & , & , & , & {}^{1}_{13}, & {}^{1}_{13}, & {}^{1}_{13}, & {}^{1}_{13}, & , & , & , & , & , & , & , & , & , & , & , \\
, & , & , & , & {}^{1}_{22}, & {}^{1}_{22}, & {}^{1}_{22}, & {}^{1}_{13}, & {}^{1}_{13}, & {}^{1}_{13}, & {}^{1}_{13}, & , & , & , & , & , & , & , & , & , & , & , \\
, & , & , & , & , & , & {}^{1}_{22}, & {}^{2}_{23}, & {}^{1}_{13}, & {}^{1}_{13}, & {}^{1}_{13}, & , & , & , & , & , & , & , & , & , & , & , \\
, & , & , & , & , & , & , & {}^{2}_{33}, & {}^{1}_{23}, & {}^{1}_{13}, & , & , & , & , & , & , & , & , & , & , & , & , \\
, & , & , & , & , & , & , & {}^{2}_{13}, & {}^{1}_{33}, & {}^{1}_{32}, & , & , & , & , & , & , & , & , & , & , & , & , \\
, & , & , & , & , & , & , & , & {}^{1}_{13}, & {}^{1}_{13}, & {}^{1}_{33}, & {}^{1}_{22}, & {}^{1}_{22}, & , & , & , & , & , & , & , & , & , \\
, & , & , & , & , & , & , & , & {}^{1}_{13}, & {}^{1}_{13}, & {}^{1}_{13}, & {}^{1}_{13}, & , & , & {}^{1}_{22}, & {}^{1}_{22}, & {}^{1}_{22}, & , & , & , & , & , \\
, & , & , & , & , & , & , & , & {}^{1}_{13}, & {}^{1}_{13}, & {}^{1}_{13}, & {}^{1}_{13}, & , & , & , & , & {}^{1}_{22}, & {}^{1}_{22}, & {}^{1}_{22}, & , & , \\
\end{array}
$$

The initial solitons $S_3^{(1)}$ and $S_4^{(2)}$ have speed 3 and 0, and they do regain the original forms after the collision except the phase shift $-3$. See around (2.40) for the definition and the neat general formula of the phase shift. More examples with $n = 4$ are available in Example 2.4–2.6.

Second, cBBS can accept a full family of commuting time evolutions $T_l^{(r)}$ $(r \in \{1, 2, \ldots, n-1\}, l \in \mathbb{Z}_{\geq 1})$ naturally without introducing an artificial "barrier" which was necessary for the conventional BBS to prevent balls from escaping from the system for $r \geq 2$. See a remark after [17, eq. (3)]. This is assured by the stability (2.23) which is another benefit of the choice of $B$ in (1.2).

---

quote relevant results casually.

Third, the complete set of conserved quantities of cBBS are given as an $(n-1)$-tuple of Young diagrams $\mu^{(1)}, \ldots,$ $\mu^{(n-1)}$, and they all admit most transparent meaning; $\mu^{(a)}$ is nothing but the list of amplitude $i$ of the color $a$ solitons $S_i^{(a)}$. Such a direct interpretation of the conserved Young diagrams in terms of solitons was possible only for the first one $\mu^{(1)}$ in the conventional BBS. Our cBBS puts all of $\mu^{(1)}, \ldots, \mu^{(n-1)}$ on an equal footing achieving the democracy of the conserved Young diagrams.

Given all these fascinating features which have escaped notice so far, we regard cBBS as the most natural as well as decent generalization of the original BBS [1] along $U_q(\widehat{sl}_n)$ differing from the conventional ones for $n \geq 3$. The nomenclature "complete" BBS is meant to indicate that the complete list $B^{1,1}, B^{2,1}, \ldots, B^{n-1,1}$ of (the labeling set of the basis of) the fundamental representations of $sl_n$ have been gathered into $B$ (1.2) at each site. Put in the other way, the somewhat unsatisfactory aspects of the conventional higher rank BBS mentioned in the above may be attributed to the "incomplete" choice of $B$. We expect similar stories also in integrable cellular automata associated with the other quantum affine algebras.

After clarifying the nature of cBBS in Sec. 2, the rest of the paper is devoted to the study of a randomized version of cBBS by thermodynamic Bethe ansatz (TBA) and generalized hydrodynamics (GHD) extending the previous work on $n = 2$ case [11]. *Randomized* means here that some measures over random initial conditions are considered, but we stress that for a given initial microscopic configuration the dynamics remains completely deterministic. We focus on the i.i.d. (independent and identically distributed) randomness including $(n-1)$ temperature generalized Gibbs ensemble (3.15). It corresponds to assigning (relative) fugacities $z_1, z_2, \ldots, z_n$ to the letters $1, 2, \ldots, n$ in the semistandard tableaux. We formulate the TBA and GHD equations on the so-called Y-function, the string/hole densities and the effective speed of solitons under any time evolution. They are fully solved in terms of the Schur functions with fugacity entries $z_1, z_2, \ldots, z_n$. These results provide a quantitative description of the cBBS soliton gas in a homogeneous system.

One of the central ideas in GHD is that the TBA Y-variable plays the role of normal mode in the Euler-scale hydrodynamics of an "integrable fluid". We apply it to the non-equilibrium dynamics of the cBBS soliton gas started from domain wall initial conditions. This is a typical setting in Riemann problem called partitioning protocol. See [10, 18, 19] and the references therein. As with the $n = 2$ case [11], density profile of solitons exhibits a rich plateaux structure depending on the type of solitons, time evolutions and the fugacity controlling the inhomogeneity of the system. We derive their position and height including the diffusive broadening by synthesizing all the ingredients in the preceding sections. They are shown to agree with extensive numerical simulations.

Let us comment on some other extensions of the basic setting (1.1) than (1.2) in the literature. The best known example is $\cdots \otimes B^{1,l_{i-1}} \otimes B^{1,l_i} \otimes B^{1,l_{i+1}} \otimes \cdots$, which is the BBS with boxes having (possibly inhomogeneous) general capacities [15]. The case $\cdots \otimes B^{k,1} \otimes B^{k,1} \otimes B^{k,1} \otimes \cdots$ has also been investigated in [20]. The closest model to our cBBS is $\cdots \otimes (B^{1,1} \otimes B^{n-1,1}) \otimes (B^{1,1} \otimes B^{n-1,1}) \otimes (B^{1,1} \otimes B^{n-1,1}) \otimes \cdots$, which yields, with an elaborate decoration at the boundary, the BBS with reflecting end [21]. Although the time evolution is designed differently there, the state space of the bulk part coincides with that of cBBS for the smallest choice $n = 3$. In all these examples except the lowest rank situation, scatterings of solitons are non-diagonal, which testifies the novelty of cBBS. As for the recent progress on probabilistic and statistical aspects of the randomized BBS (non-complete BBS except $n = 2$), see also [17, 22–27].

The outline of the paper is as follows. In Sec. 2 we introduce cBBS and explain its basic properties such as the commuting family of time evolutions, complete set of conserved quantities, solitons and their scattering rule, and the inverse scattering formalism. A key role is played by the Bethe ansatz structure which is realized as *soliton/string correspondence*. In Sec.

3 we proceed to the randomized version of cBBS. Explicit solutions are presented for the TBA equation (3.19) in (3.24), and string/hole densities in (3.52)-(3.53). They are expressed by the special combination (3.36) of the Schur function (3.6). These results can also be viewed as the solution to the variant of the limit shape problem of rigged configurations [17] adapted to cBBS. In Sec. 4 we apply GHD to the randomized cBBS. We present the speed equation of solitons for any time evolution (4.1) and its explicit solution in (4.7). The former coincides with [23, eq.(11.7)] for $n = 2$ and $T_\infty^{(1)}$ dynamics. The formulation of the GHD equations in matrix forms in Sec. 4.2 is useful to recognize the characteristic Bethe ansatz structure. In Sec. 5 we study the density plateaux generated from domain wall initial conditions by GHD. Each plateau is formed by particular subsets of solitons from two sides of the domain wall. These subsets, which we call soliton contents, show some intriguing patterns. Sec. 6 is devoted to summary and conclusions.

Appendix A recalls the algorithm for obtaining the image of the combinatorial $R$ in the most general case. Appendix B includes the explicit piecewise linear formulas for the combinatorial $R$ for $n = 2$ and 3. Appendix C is a brief exposition of the KSS bijection in the situation used in this paper.

## 2 Complete box-ball system

Throughout the paper we use the notation

$$[a, b] = \{a, a+1, \ldots, b\} \quad (a \leq b \in \mathbb{Z}), \qquad \mathcal{I} = \{(a, i) \mid a \in [1, n-1], i \in \mathbb{Z}_{\geq 1}\}. \tag{2.1}$$

### 2.1 Preliminaries

Consider the classical simple Lie algebra $sl_n$ ($n \geq 2$). We denote its Cartan matrix by $(C_{ab})_{a,b=1}^{n-1}$, where

$$C_{ab} = 2\delta_{a,b} - \delta_{|a-b|,1}. \tag{2.2}$$

Let $\varpi_1, \ldots, \varpi_n$ be the fundamental weights and $\alpha_1, \ldots, \alpha_n$ be the simple roots. They are related by $\alpha_a = \sum_{b=1}^{n} C_{ab}\varpi_b$. Let $\widehat{sl}_n$ be the non-twisted affinization of $sl_n$ [28] and $U_q = U_q(\widehat{sl}_n)$ be the quantum affine algebra (without derivation operator) [29,30]. There is a family of irreducible finite-dimensional representations $\{W_l^{(k)} \mid (k, l) \in [1, n-1] \times \mathbb{Z}_{\geq 0}\}$ of $U_q$ called Kirillov-Reshetikhin (KR) module[2] named after the related work on the Yangian [31]. As a representation of $U_q(sl_n)$, $W_l^{(k)}$ is isomorphic to the type 1 irreducible highest weight module $V(l\varpi_k)$ with highest weight $l\varpi_k$. $W_l^{(k)}$ is known to have a crystal base $B^{k,l}$ [4,32]. Roughly speaking, it is a set of basis vectors of the $W_l^{(k)}$ at $q = 0$. Practically in this paper we only need its combinatorial definition as a set and the operation called combinatorial $R$. They are described in the next subsection.

### 2.2 Combinatorial $R$

For $(k, l) \in \mathcal{I}$, let $B^{k,l}$ be the set of semistandard tableaux of $k \times l$ rectangular shape over the alphabet $\{1, 2, \ldots, n\}$. Let $b = (t_{ij})$, where $t_{ij}$ is the entry at the $i$ th row from the top and the $j$ th column from the left. By the definition, $t_{1j} < t_{2j} < \cdots < t_{kj}$ and $t_{i1} \leq t_{i2} \leq \cdots \leq t_{il}$ hold for any $i \in [1, k]$ and $j \in [1, l]$. The array row$(b) = t_k \ldots t_2 t_1$ with $t_i = t_{i1} t_{i2} \ldots t_{il}$ is

---

[2]The actual KR modules carries a spectral parameter. In this paper it is irrelevant and hence suppressed.

called the *row word* of $b$. For instance, we have

$$b = \begin{array}{|c|c|c|}\hline 1 & 2 & 3 \\\hline 2 & 4 & 5 \\\hline\end{array} \in B^{2,3}, \quad \mathrm{row}(b) = 245123, \qquad c = \begin{array}{|c|c|}\hline 1 & 2 \\\hline 2 & 3 \\\hline 4 & 5 \\\hline\end{array} \in B^{3,2}, \quad \mathrm{row}(c) = 452312. \quad (2.3)$$

For tableaux $S$ and $T$, their product is defined in the following two ways which are known to be equivalent:

$$S \cdot T = (\cdots((S \leftarrow u_1) \leftarrow u_2) \leftarrow \cdots) \leftarrow u_l \qquad (\mathrm{row}(T) = u_1 u_2 \ldots u_l) \qquad (2.4)$$

$$= v_1 \rightarrow (v_2 \rightarrow (\cdots(v_m \rightarrow T)\cdots)) \qquad (\mathrm{row}(S) = v_1 v_2 \ldots v_m). \qquad (2.5)$$

Here $\leftarrow$ denotes the *row insertion* and $\rightarrow$ does the *column insertion* [12].

*Row insertion $S \leftarrow x$:*

1. Start at the top row of $S$.

2. If $x$ is larger than or equal to the rightmost number in the current row, add $x$ to the right of the row, which is the end of the insertion.

3. Otherwise, replace the leftmost element $y$ of the row such that $x < y$ by $x$ and go to step (1) starting at the next row, now with $y$ to be inserted.

*Column insertion $x \rightarrow S$:*

1. Start at the leftmost column of $S$.

2. If $x$ is larger than the bottom number in the current column, add $x$ to the bottom of the column, which is the end of the insertion.

3. Otherwise, replace the smallest element $y$ of the column such that $x \leq y$ by $x$ and go to step (1) starting at the next column, now with $y$ to be inserted.

In the above example one has

$$b \cdot c = \begin{array}{|c|c|c|c|c|}\hline 1 & 1 & 2 & 2 & 5 \\\hline 2 & 2 & 3 \\\cline{1-3} 3 & 4 \\\cline{1-2} 4 & 5 \\\cline{1-2}\end{array} \quad , \qquad c \cdot b = \begin{array}{|c|c|c|c|c|}\hline 1 & 1 & 2 & 2 & 3 \\\hline 2 & 2 & 4 & 5 \\\cline{1-4} 3 & 5 \\\cline{1-2} 4 \\\cline{1-1}\end{array} \quad . \qquad (2.6)$$

Now we are ready to explain the *combinatorial $R$* and the *local energy $H$*, which are essential ingredients in our cBBS. These notions were introduced in the crystal base theory [33] as the proper analogue of the quantum $R$ matrices at $q = 0$, motivated by the corner transfer matrix method [2]. The concrete description given below is due to [34]. The combinatorial $R$ is the bijection

$$R: \ B^{k,l} \otimes B^{k',l'} \rightarrow B^{k',l'} \otimes B^{k,l}; \quad b \otimes c \mapsto \tilde{c} \otimes \tilde{b}, \qquad (2.7)$$

whose image $\tilde{c} \otimes \tilde{b} = R(b \otimes c)$ is characterized by the condition $c \cdot b = \tilde{b} \cdot \tilde{c}$. Here and in what follows, $\otimes$ is used to mean just the product of sets. It should not be confused with the product $\cdot$ of tableaux. Note that the dependence on $k, k', l, l'$ has been suppressed in $R$. For example, from

$$\begin{array}{|c|c|c|}\hline 1 & 2 & 2 \\\hline 2 & 4 & 5 \\\hline\end{array} \cdot \begin{array}{|c|c|}\hline 1 & 3 \\\hline 2 & 4 \\\hline 3 & 5 \\\hline\end{array} = \begin{array}{|c|c|c|c|c|}\hline 1 & 1 & 2 & 2 & 3 \\\hline 2 & 2 & 4 & 5 \\\cline{1-4} 3 & 5 \\\cline{1-2} 4 \\\cline{1-1}\end{array} \quad , \qquad (2.8)$$

which agrees with the right tableau in (2.6), we find

$$R: \quad \boxed{\begin{array}{ccc}1&2&3\\2&4&5\end{array}} \otimes \boxed{\begin{array}{cc}1&2\\2&3\\4&5\end{array}} \mapsto \boxed{\begin{array}{cc}1&3\\2&4\\3&5\end{array}} \otimes \boxed{\begin{array}{ccc}1&2&2\\2&4&5\end{array}}. \tag{2.9}$$

This is a particular case of $R : B^{2,3} \otimes B^{3,2} \to B^{3,2} \otimes B^{2,3}$. The algorithm to find $\tilde{b}$ and $\tilde{c}$ satisfying the condition $\tilde{b} \cdot \tilde{c} = c \cdot b$ is described in Appendix A following [35, p55].

The local energy $H$ is the function defined by

$$H: \ B^{k,l} \otimes B^{k',l'} \to \mathbb{Z}_{\geq 0}; \quad b \otimes c \mapsto H(b \otimes c), \tag{2.10}$$

$H(b \otimes c) =$ number of boxes strictly below the $\max(k, k')$-th row of the tableau $c \cdot b$.
$$\tag{2.11}$$

In our working example (2.3), $\max(k, k') = \max(2, 3) = 3$, hence from (2.6) we have $H(b \otimes c) = 1$.

Let $u_{k,l} \in B^{k,l}$ be the particular tableau whose entries in the $i$ th row from the top are all $i$. It corresponds to the highest weight element in the representation of $U_q(sl_n)$. Some small examples are

$$u_{1,1} = \boxed{1}, \quad u_{2,1} = \boxed{\begin{array}{c}1\\2\end{array}}, \quad u_{3,1} = \boxed{\begin{array}{c}1\\2\\3\end{array}}, \quad u_{1,2} = \boxed{\begin{array}{cc}1&1\end{array}}, \quad u_{2,2} = \boxed{\begin{array}{cc}1&1\\2&2\end{array}}, \quad u_{3,2} = \boxed{\begin{array}{cc}1&1\\2&2\\3&3\end{array}}. \tag{2.12}$$

It is easy to see

$$R(u_{k,l} \otimes u_{k',l'}) = u_{k',l'} \otimes u_{k,l}, \tag{2.13}$$
$$H(b \otimes u_{k',l'}) = 0 \qquad (\forall b \in B^{k,l}) \tag{2.14}$$

for any $k, k', l, l'$.

If $R$ in (2.7) and $H$ in (2.10) are denoted by $R_{B^{k,l} \otimes B^{k',l'}}$ and $H_{B^{k,l} \otimes B^{k',l'}}$ respectively, they satisfy

$$R_{B^{k,l} \otimes B^{k,l}} = \mathrm{id}_{B^{k,l} \otimes B^{k,l}}, \tag{2.15}$$
$$R_{B^{k,l} \otimes B^{k',l'}} R_{B^{k',l'} \otimes B^{k,l}} = \mathrm{id}_{B^{k',l'} \otimes B^{k,l}}, \tag{2.16}$$
$$H_{B^{k',l'} \otimes B^{k,l}} \circ R_{B^{k,l} \otimes B^{k',l'}} = H_{B^{k,l} \otimes B^{k',l'}} \tag{2.17}$$

by the definition. In spite of the simplification (2.15), the local energy $H_{B^{k,l} \otimes B^{k,l}}(b \otimes c)$ still depends on $b \otimes c$ nontrivially . The relation in (2.16) is called the inversion relation, which implies that $R(b \otimes c) = \tilde{c} \otimes \tilde{b}$ is equivalent to $R(\tilde{c} \otimes \tilde{b}) = b \otimes c$. In view of this we write these relations also as $b \otimes c \simeq \tilde{c} \otimes \tilde{b}$.

The most important property of the combinatorial $R$ is the Yang-Baxter equation [2]. It includes $R$ as the classical part and $H$ as the affine part. To unify them into a single equation, we introduce an infinite set $\mathrm{Aff}(B^{k,l}) = \{b[\alpha] \mid b \in B^{k,l}, \alpha \in \mathbb{Z}\}$ and extend $R$ to the map $\hat{R}$ defined by

$$\begin{aligned}\hat{R}: \ \mathrm{Aff}(B^{k,l}) \otimes \mathrm{Aff}(B^{k',l'}) &\to \quad \mathrm{Aff}(B^{k',l'}) \otimes \mathrm{Aff}(B^{k,l}) \\ b[\alpha] \otimes c[\beta] \quad &\mapsto \quad \tilde{c}[\beta + H(b \otimes c)] \otimes \tilde{b}[\alpha - H(b \otimes c)],\end{aligned} \tag{2.18}$$

where $\tilde{c} \otimes \tilde{b} = R(b \otimes c)$ as in (2.7). The integer $\alpha$ attached to $b$ in $b[\alpha]$ is called a *mode*. It is a reminiscent of the spectral parameter in quantum $R$ matrices. Note that the shift of the mode $H(b \otimes c)$ in (2.18) is equally presented as $H(\tilde{c} \otimes \tilde{b})$ since they coincide owing to

(2.17). This guarantees that $\hat{R}$ also satisfies the inversion relation. Thus we also write (2.18) as $b[\alpha] \otimes c[\beta] \simeq \tilde{c}[\beta + H(b \otimes c)] \otimes \tilde{b}[\alpha - H(b \otimes c)]$ putting the two sides on a more equal footing.

Now the Yang-Baxter equation is presented as

$$(\hat{R} \otimes \mathrm{id})(\mathrm{id} \otimes \hat{R})(\hat{R} \otimes \mathrm{id}) = (\mathrm{id} \otimes \hat{R})(\hat{R} \otimes \mathrm{id})(\mathrm{id} \otimes \hat{R}), \tag{2.19}$$

which is an equality of the maps $\mathrm{Aff}(B^{k,l}) \otimes \mathrm{Aff}(B^{k',l'}) \otimes \mathrm{Aff}(B^{k'',l''}) \to \mathrm{Aff}(B^{k'',l''}) \otimes \mathrm{Aff}(B^{k',l'}) \otimes \mathrm{Aff}(B^{k,l})$ for arbitrary $(k,l), (k',l'), (k'',l'') \in \mathcal{I}$. Here is an example for $(k,k',k'',l,l',l'') = (1, 2, 3, 3, 2, 1)$, where the modes are indicated as the indices in the bottom right of the tableaux.

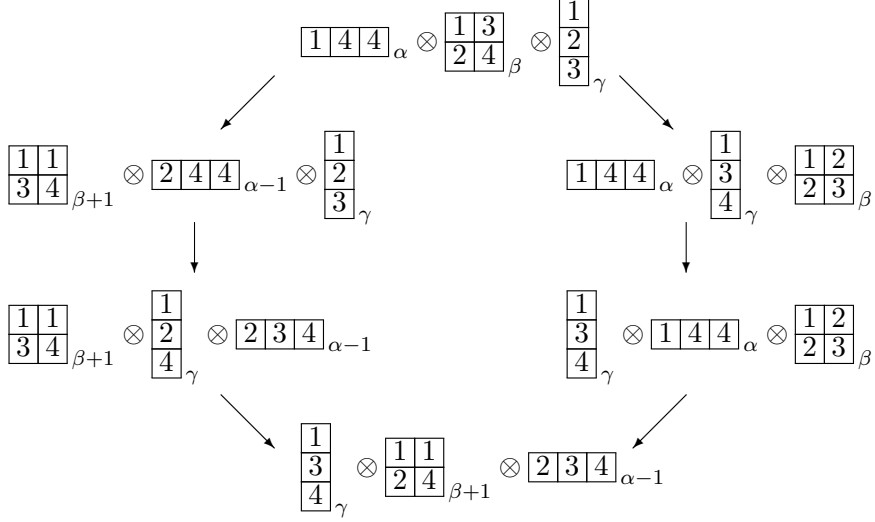

In general, we will write $b_1 \otimes \cdots \otimes b_L \simeq b_1' \otimes \cdots \otimes b_L'$ if the elements $b_1 \otimes \cdots \otimes b_L$ and $b_1' \otimes \cdots \otimes b_L'$ are transformed to each other by successively applying the combinatorial $R$ to the neighboring components as above.

The relation $b[\alpha] \otimes c[\beta] \simeq \tilde{c}[\beta + h] \otimes \tilde{b}[\alpha - h]$ (2.18) or its classical part (2.7) without the mode will be depicted as ($h = H(b \otimes c)$)

$$
\begin{array}{ccc}
& c[\beta] & \\
b[\alpha] \xrightarrow{\phantom{xxxx}} & \tilde{b}[\alpha - h] & \qquad b \xrightarrow{\phantom{xx}c\phantom{xx}} \tilde{b} \\
& \tilde{c}[\beta + h] & \qquad\quad \tilde{c}
\end{array}
\tag{2.20}
$$

As the examples shown so far indicate, one can always forget the modes without destroying the relations on the classical parts. For readers convenience, we include explicit formulas of $R$ and $H$ for $n = 2, 3$ cases in Appendix B.

## 2.3 Vacuum state and stability

Collecting all the $l = 1$ cases in (2.12) for a given $n$, we set

$$\mathrm{vac} = u_{1,1} \otimes u_{2,1} \otimes \cdots \otimes u_{n-1,1} \in B, \qquad B = B^{1,1} \otimes B^{2,1} \otimes \cdots \otimes B^{n-1,1}. \tag{2.21}$$

For instance, vac looks as

$$\boxed{1} \;\; (n=2), \qquad \boxed{1} \otimes \begin{array}{c}\boxed{1}\\\boxed{2}\end{array} \;\; (n=3), \qquad \boxed{1} \otimes \begin{array}{c}\boxed{1}\\\boxed{2}\end{array} \otimes \begin{array}{c}\boxed{1}\\\boxed{2}\\\boxed{3}\end{array} \;\; (n=4). \tag{2.22}$$

The special element vac is referred to as *vacuum*, which will be used in the background configuration in our complete BBS. The set $B$ is the tensor product of (crystals of) the complete list of fundamental representations of $U_q(sl_n)$. Using the KSS bijection in Appendix C, one can show a stability

$$B^{k,l} \otimes B^{\otimes L} \ni b \otimes \mathrm{vac}^{\otimes L} \simeq v_1 \otimes \cdots \otimes v_m \otimes \mathrm{vac}^{\otimes L-m} \otimes u_{k,l} \in B^{\otimes L} \otimes B^{k,l} \qquad (\forall b \in B^{k,l}), \tag{2.23}$$

when $L$ gets sufficiently large for some $m$ and $v_1, \ldots, v_m \in B$.

As we will discuss in detail in Sec. 2.4, in the equation above we may interpret $b$ as the initial state of a 'carrier'. The system contains $L$ boxes that are initially in the vaccum state. The equality describes the change of the state of the system after the carrier has passed across the system, from left to right. It indicates that for large enough $L$ only a finite number (at most $m$) of boxes on the left side get modified, while the others remain in the vacuum state vac. The output state of the carrier is also the vacuum $u_{k,l}$. Here is an example $(k,l) = (1,3)$ with $n = 3$:

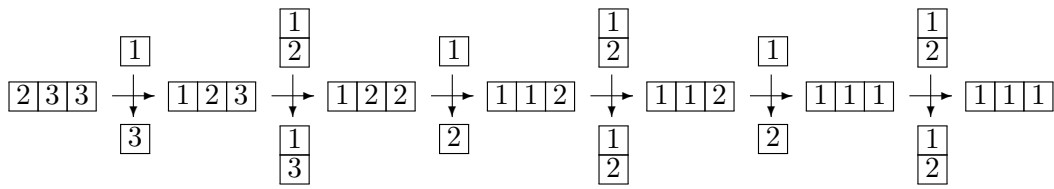

Here $b = \boxed{2\,3\,3}$ and $L = 3$. The diagram is constructed by concatenating (2.20). Another example for $(k,l) = (2,3)$ with $n = 3$ is

In the (right) distance, all the combinatorial $R$'s tend to the situation (2.13).

## 2.4 Time evolutions and conserved quantities

Now we define the complete BBS (cBBS). It is a dynamical system on $B^{\otimes L}$. An element $b_1 \otimes \cdots \otimes b_L \in B^{\otimes L}$ is called a *state* where each $b_i \in B$ is a *local state* at site $i$. Thus there are $\#B = \binom{n}{1}\binom{n}{2}\cdots\binom{n}{n-1}$ local states which have an internal structure as in (2.21).

We assume that $L$ is sufficiently large and impose the boundary condition that the "distant" local states are all vac in (2.21). This is compatible with the dynamics that we are going to introduce below. For any $(r,l) \in \mathcal{I}$ in (2.1), we define the time evolution $T_l^{(r)} : B^{\otimes L} \to B^{\otimes L}$ as follows:

$$T_l^{(r)}(b_1 \otimes \cdots \otimes b_L) = b_1' \otimes \cdots \otimes b_L' \qquad (b_i \in B), \tag{2.24}$$

$$u_{r,l} \otimes b_1 \otimes \cdots \otimes b_L \simeq b_1' \otimes \cdots \otimes b_L' \otimes u_{r,l}. \tag{2.25}$$

The relation (2.25) is obtained by successively applying the combinatorial $R$ sending $B^{r,l}$ through $B^{\otimes L}$ to the right. Like the previous examples, the procedure can be shown graphically as

$$\begin{array}{c} b_1 \quad b_2 \qquad\qquad b_L \\ u_{r,l} \;\rule[0.5ex]{4cm}{0.4pt}\; u_{r,l} \\ b_1' \quad b_2' \qquad\qquad b_L' \end{array} \tag{2.26}$$

Here the boundary condition implies $b_j = b_{j+1} = \cdots = b_L = $ vac for $L \gg L - j \gg 1$. Then, thanks to the stability (2.23), we always end up with $u_{r,l}$ in the right. In short, $T_l^{(r)}$ is obtained by going from NW to SE in (2.26). Thanks to the inversion relation (2.16), one can reverse the procedure to define the inverse $(T_l^{(r)})^{-1} : b_1' \otimes \cdots \otimes b_L' \mapsto b_1 \otimes \cdots \otimes b_L$ by going from SE to NW under the boundary condition $b_1 = b_2 = \cdots = b_m = $ vac for $1 \ll m \ll L$.

The key to the above construction is $B^{r,l}$ attached to the horizontal arrow in the diagram (2.26). It induces the time evolution $T_l^{(r)}$ via the interactions with the local states by the combinatorial $R$. This degrees of freedom is called *carrier* [36], and especially $u_{r,l}$ is referred to as the vacuum carrier.

A local state may be labeled with its deviation from vac. For instance when $n = 3$, the 9 local states in $B = B^{1,1} \otimes B^{2,1}$ can be displayed as

$$
\begin{array}{llll}
\text{vac} = \boxed{1} \otimes \boxed{\genfrac{}{}{0pt}{}{1}{2}} = \ , &
\boxed{1} \otimes \boxed{\genfrac{}{}{0pt}{}{1}{3}} = \ , &
\boxed{1} \otimes \boxed{\genfrac{}{}{0pt}{}{2}{3}} = \ , \\[6pt]
\boxed{2} \otimes \boxed{\genfrac{}{}{0pt}{}{1}{2}} = \ , &
\boxed{2} \otimes \boxed{\genfrac{}{}{0pt}{}{1}{3}} = \ , &
\boxed{2} \otimes \boxed{\genfrac{}{}{0pt}{}{2}{3}} = \ , \\[6pt]
\boxed{3} \otimes \boxed{\genfrac{}{}{0pt}{}{1}{2}} = \ , &
\boxed{3} \otimes \boxed{\genfrac{}{}{0pt}{}{1}{3}} = \ , &
\boxed{3} \otimes \boxed{\genfrac{}{}{0pt}{}{2}{3}} = \ ,
\end{array} \tag{2.27}
$$

where the entries common with vac are colored white. In general one may regard a local state as an arrangement of $n - 1$ kinds of balls in $n - 1$ columns each obeying the semistandard condition. Then the combinatorial $R$ specifies the rule under which the balls are exchanged between the carrier and a box. The complete BBS is a nomenclature emphasizing that the complete list of the possible column length $1, 2, \ldots, n - 1$ have been built in each local state as in (2.21). See also a remark after (2.22). We will see that this will lead to a remarkable simplification of solitons and their scattering.

The time evolution $T_l^{(r)}$ is associated with the *energy* $E_l^{(r)} : B^{\otimes L} \to \mathbb{Z}_{\geq 0}$. Its quickest definition is to declare that the mode of the carrier in (2.26) is shifted as follows:

$$
u_{r,l}[\alpha] \ \begin{array}{c} b_1 \quad b_2 \qquad\qquad\qquad b_L \\ \xrightarrow{\hspace{5cm}} \\ b_1' \quad b_2' \qquad\qquad\qquad b_L' \end{array} \ u_{r,l}[\alpha - E_l^{(r)}(b_1 \otimes \cdots \otimes b_L)] \tag{2.28}
$$

To be more detailed, suppose the local states have the form $b_i = c_{(n-1)(i-1)+1} \otimes \cdots \otimes c_{(n-1)i} \in B = B^{1,1} \otimes \cdots \otimes B^{n-1,1}$ so that $b_1 \otimes \cdots \otimes b_L = c_1 \otimes \cdots \otimes c_{(n-1)L}$. The carriers $u_1, \ldots, u_{(n-1)L} \in B^{r,l}$ in the intermediate steps are determined from the initial condition $u_0 = u_{r,l}$ and the recursion

$$
u_{j-1} \otimes c_j \simeq c_j' \otimes u_j \qquad (1 \leq j \leq (n-1)L), \tag{2.29}
$$

for some $c_j'$. From the defining behavior of the mode in (2.18), the energy $E_l^{(r)}$ is expressed as the sum of local energies attached to (2.29) as

$$
E_l^{(r)}(b_1 \otimes \cdots \otimes b_L) = \sum_{j=1}^{(n-1)L} H(u_{j-1} \otimes c_j) \in \mathbb{Z}_{\geq 0}. \tag{2.30}
$$

As $j$ gets large, $u_j$ tends to $u_{r,l}$ by the stability (2.23) under the boundary condition. Then (2.14) assures that the sum (2.30) converges to a finite value which is independent of $L$ if it gets large enough.

The time evolution and the energy enjoy the properties

$$
\text{commutativity}: \ T_l^{(k)} T_{l'}^{(k')} = T_{l'}^{(k')} T_l^{(k)}, \tag{2.31}
$$

$$
\text{conservation}: \ E_l^{(k)} T_{l'}^{(k')} = E_l^{(k)} \tag{2.32}
$$

on any state and for arbitrary $(k, l), (k', l') \in \mathcal{I}$. This is a simple consequence of the Yang-Baxter equation. In fact, write $\mathfrak{s} = b_1 \otimes \cdots \otimes b_L$ for short and consider

$$
\begin{aligned}
u_{k,l}[\alpha] \otimes u_{k',l'}[\beta] \otimes \mathfrak{s} &\simeq u_{k,l}[\alpha] \otimes T_{l'}^{(k')}(\mathfrak{s}) \otimes u_{k',l'}[\beta - E_{l'}^{(k')}(\mathfrak{s})] \\
&\simeq T_l^{(k)} T_{l'}^{(k')}(\mathfrak{s}) \otimes u_{k,l}[\alpha - E_l^{(k)}(T_{l'}^{(k')}(\mathfrak{s}))] \otimes u_{k',l'}[\beta - E_{l'}^{(k')}(\mathfrak{s})]
\end{aligned} \tag{2.33}
$$

by using (2.28) successively. On the other hand, one may first apply $u_{k,l}[\alpha] \otimes u_{k',l'}[\beta] \simeq u_{k',l'}[\beta] \otimes u_{k,l}[\alpha]$ which is a consequence of (2.13) and (2.14). It tells that (2.33) is also equal to

$$
\begin{aligned}
&T_{l'}^{(k')} T_l^{(k)}(\mathfrak{s}) \otimes u_{k',l'}[\beta - E_{l'}^{(k')}(T_l^{(k)}(\mathfrak{s}))] \otimes u_{k,l}[\alpha - E_l^{(k)}(\mathfrak{s})] \\
&\simeq T_{l'}^{(k')} T_l^{(k)}(\mathfrak{s}) \otimes u_{k,l}[\alpha - E_l^{(k)}(\mathfrak{s})] \otimes u_{k',l'}[\beta - E_{l'}^{(k')}(T_l^{(k)}(\mathfrak{s}))].
\end{aligned} \tag{2.34}
$$

Comparing (2.33) and (2.34), one obtains (2.31) and (2.32).

**Remark 2.1.** The composition of the combinatorial $R$'s achieving $B \otimes B \mapsto B \otimes B$ is the identity. Therefore if vac is put in place of $u_{r,l}$ at the left in the diagram (2.26), we have $b_1' = \text{vac}, b_2' = b_1,$
$b_3' = b_2, \ldots$. It follows that the composition $T_1^{(1)} T_1^{(2)} \cdots T_1^{(n-1)}$ is the *translation* to the right by one lattice unit.

**Remark 2.2.** Both the time evolution $T_l^{(k)}$ and the associated energy $E_l^{(k)}$ have the well-defined limit as $l \to \infty$. A simple explanation of this fact is provided by the inverse scattering scheme (2.48), which translates $T_l^{(k)}$ as in (2.49) and $E_l^{(k)}$ as in (2.50) and (C.10). In particular, if $\gamma_k$ denotes the maximal amplitude of color $k$ solitons that will be defined in the next subsection, one has $T_l^{(k)} = T_{\gamma_k}^{(k)}$ and $E_l^{(k)} = E_{\gamma_k}^{(k)}$ for all $l \geq \gamma_k$.

## 2.5 Solitons and their scattering

This subsection is the place where things start to differ significantly from the conventional (non-complete) BBS. The claims can be proved by invoking the inverse scattering method explained in Sec. 2.6.

We observe the cBBS in terms of the deviation of the states from the background configuration $\text{vac}^{\otimes L}$. The first question is to find the "collective modes" or "quasi particles" which are stable localized patterns having a constant speed under any time evolution when isolated from the other patterns different from vac. They deserve to be called *solitons* if the stability under *multi-body* collisions, a much more stringent postulate, is further obeyed. This turn out to be the case for the cBBS reflecting the existence of the $n-1$ families of the conserved quantities $E_l^{(1)}, \ldots, E_l^{(n-1)}$ ($l \in \mathbb{Z}_{\geq 1}$). In what follows we present a complete list of solitons together with their scattering rule.

First we introduce the elementary excitations $s_1, s_2, \ldots, s_{n-1} \in B$ by

$$
n = 2 : \quad s_1 = \boxed{2}, \tag{2.35}
$$

$$
n = 3 : \quad s_1 = \boxed{2} \otimes \begin{smallmatrix}\boxed{1}\\\boxed{2}\end{smallmatrix}, \qquad s_2 = \boxed{1} \otimes \begin{smallmatrix}\boxed{1}\\\boxed{3}\end{smallmatrix}, \tag{2.36}
$$

$$
n = 4 : \quad s_1 = \boxed{2} \otimes \begin{smallmatrix}\boxed{1}\\\boxed{2}\end{smallmatrix} \otimes \begin{smallmatrix}\boxed{1}\\\boxed{2}\\\boxed{3}\end{smallmatrix}, \qquad s_2 = \boxed{1} \otimes \begin{smallmatrix}\boxed{1}\\\boxed{3}\end{smallmatrix} \otimes \begin{smallmatrix}\boxed{1}\\\boxed{2}\\\boxed{3}\end{smallmatrix}, \qquad s_3 = \boxed{1} \otimes \begin{smallmatrix}\boxed{1}\\\boxed{2}\end{smallmatrix} \otimes \begin{smallmatrix}\boxed{1}\\\boxed{2}\\\boxed{4}\end{smallmatrix}. \tag{2.37}
$$

They are different from $\text{vac} \in B$ in (2.22) by only one entry at the bottom of one tableau.

General definition is this:

$$s_a = u_{1,1} \otimes \cdots u_{a-1,1} \otimes \boxed{\begin{array}{c} 1 \\ 2 \\ \vdots \\ a-1 \\ a+1 \end{array}} \otimes u_{a+1,1} \otimes \cdots \otimes u_{n-1,1} \in B \qquad (a \in [1, n-1]). \tag{2.38}$$

Using $s_a$ as the building block we next introduce

$$S_i^{(a)} = \overbrace{s_a \otimes \cdots \otimes s_a}^{i} \in B^{\otimes i} \qquad ((a,i) \in \mathcal{I}) \tag{2.39}$$

and call it soliton of *color* $a$ and *length* (or amplitude or size) $i$, or *type* $(a,i)$ for short. They have the following properties.

(I) When isolated, a soliton $S_i^{(a)}$ proceeds to the right with the velocity $\delta_{ar} \min(i,l)$ under $T_l^{(r)}$.

(II) Solitons are stable under collisions. A collision of solitons of type $(a,i)$ and $(b,j)$ induces the displacement $\Delta$ in their asymptotic trajectories which is common in the magnitude and opposite in the direction.

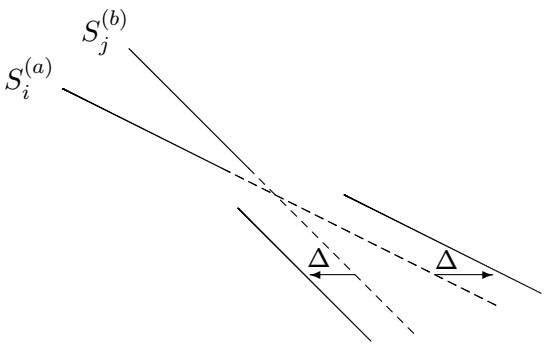

Here time grows from the top to the bottom. The quantity $\Delta$, we call it *phase shift*, is given by

$$\Delta = C_{ab} \min(i,j), \tag{2.40}$$

where $(C_{ab})_{1 \le a,b \le n-1}$ is the Cartan matrix (2.2). Note that $\Delta$ can be either positive or negative or even zero.

(III) By applying time evolutions sufficiently many times, *any* state can be decomposed into isolated solitons. Such asymptotic states can be taken, for example, as

$$\ldots (\text{color 1 solitons}) \ldots \ldots (\text{color 2 solitons}) \ldots \quad \ldots \quad \ldots (\text{color } n-1 \text{ solitons}) \ldots, \tag{2.41}$$

where $\ldots$ denotes vac's. Distance of the neighboring islands can be made as large as one wishes. Each (color $a$ solitons) has the form

$$\ldots (S_1^{(a)}\text{'s}) \ldots \ldots (S_2^{(a)}\text{'s}) \ldots \quad \ldots \quad \ldots (S_{\gamma_a}^{(a)}\text{'s}) \ldots, \tag{2.42}$$

where $\gamma_a$ is the maximal length of the color $a$ solitons as mentioned in Remark 2.2. Again the distance of $(S_k^{(a)}$'s) and $(S_{k+1}^{(a)}$'s) can be made arbitrarily large. Each $(S_i^{(a)}$'s) can be brought into the form

$$\ldots S_i^{(a)} \ldots S_i^{(a)} \ldots \quad \ldots \quad \ldots S_i^{(a)} \ldots. \tag{2.43}$$

In contrast to (2.41) and (2.42), the separation $\ldots$ of the neighboring $S_i^{(a)}$'s remains constant under time evolutions, but it is at least $i$ everywhere, namely, vac$^{\otimes d}$ with $d \geq i$.

(IV) The energy $E_l^{(r)}$ of the asymptotic states (hence all those connected to it by time evolutions) described in (2.41) – (2.43) is given by

$$E_l^{(r)} = \sum_{i=1}^{\gamma_r} \min(i,l) m_i^{(r)}, \tag{2.44}$$

where $m_i^{(a)}$ is the number (multiplicity) of type $(a,i)$ solitons in (2.43).

By inverting (2.44) one can obtain the $m_i^{(r)}$, that is the soliton content of the state, from the conserved energies $E_l^{(r)}$. In addition, by using the local energies (2.30), one gets the local contribution to $m_i^{(r)}$, which is therefore a soliton density. This is how the soliton densities are computed in the simulations in Sec. 5.3.

**Remark 2.3.** Denote the Cartan matrix (2.2) by $C_n$ exhibiting the dependence on $n$ temporarily. In view of the inverse $(C_n^{-1})_{ab} = \min(a,b) - \frac{ab}{n}$, the phase shift $C_{ab} \min(i,j)$ in (2.40) is the matrix element of $C_n \otimes C_\infty^{-1}$. This originates in the $x = 0$ Fourier component of the TBA kernel $\hat{\mathcal{A}}_{ab}^{ij}(x)\hat{\mathcal{M}}_{ab}(x)$ for the level $\ell$ $U_q(\widehat{sl}_n)$ RSOS model in [37, eq.(14.14)] in the limit $\ell \to \infty$.

Reflecting the commutativity (2.31), the phase shift $\Delta$ is independent of the choice of the time evolution $T_l^{(r)}$ as long as the two solitons have different bare speeds in (I) under it so that they eventually collide.

Let us present a few examples of (II) concerning collisions. We take $n = 4$.

**Example 2.4.** Collision of $S_3^{(a)}$ and $S_1^{(a)}$ under the time evolution by successive applications of $T_l^{(a)}$ with any $a = 1, 2, 3$ and $l \geq 3$.

$$
\begin{array}{cccccccccccccccccc}
, & , s_a, s_a, s_a, & , & , & , s_a, & , & , & , & , & , & , & , & , & , \\
, & , & , & , & , s_a, s_a, s_a, & , s_a, & , & , & , & , & , & , & , & , \\
, & , & , & , & , & , & , s_a, & , s_a, s_a, s_a, & , & , & , & , & , & , \\
, & , & , & , & , & , & , & , s_a, & , & , & , s_a, s_a, s_a, & , & , & , \\
\end{array}
$$

A blank , , signifies the vacuum vac. See (2.22). The larger (resp. smaller) soliton $S_3^{(a)}$ (resp. $S_1^{(a)}$) is pushed forward (resp. pulled backward) by 2 sites compared from its free motion. This agrees with the phase shift $C_{aa} \min(3,1) = 2$ in (2.40). Note that we count $B$ having the internal structure as one lattice unit. In this example, no local state other than vac and $s_a$ is generated by the collision. In general if solitons of only one color are present, cBBS is equivalent to the very original BBS [1] corresponding to $n = 2$

**Example 2.5.** Successive collisions of $S_3^{(3)}$ with $S_1^{(1)}$ and $S_2^{(2)}$ under the time evolution $T_l^{(3)}$ with $l \geq 3$. We write $s_2$ in (2.37) for example as $\begin{smallmatrix} 1 \\ 12 \\ 133 \end{smallmatrix}$ to save the space.

$$
\begin{array}{ccccccccccccccccccc}
, & \genfrac{}{}{0pt}{}{\genfrac{}{}{0pt}{}{1}{12}}{124}, & \genfrac{}{}{0pt}{}{\genfrac{}{}{0pt}{}{1}{12}}{124}, & \genfrac{}{}{0pt}{}{\genfrac{}{}{0pt}{}{1}{12}}{124}, & , & , & , & \genfrac{}{}{0pt}{}{\genfrac{}{}{0pt}{}{1}{12}}{223}, & , & \genfrac{}{}{0pt}{}{\genfrac{}{}{0pt}{}{1}{12}}{133}, & \genfrac{}{}{0pt}{}{\genfrac{}{}{0pt}{}{1}{12}}{133}, & , & , & , & , & , & , & ,
\end{array}
$$

$$
\begin{array}{ccccccccccccccccccc}
, & , & , & , & \genfrac{}{}{0pt}{}{\genfrac{}{}{0pt}{}{1}{12}}{124}, & \genfrac{}{}{0pt}{}{\genfrac{}{}{0pt}{}{1}{12}}{124}, & \genfrac{}{}{0pt}{}{\genfrac{}{}{0pt}{}{1}{12}}{124}, & \genfrac{}{}{0pt}{}{\genfrac{}{}{0pt}{}{1}{12}}{223}, & , & \genfrac{}{}{0pt}{}{\genfrac{}{}{0pt}{}{1}{12}}{133}, & \genfrac{}{}{0pt}{}{\genfrac{}{}{0pt}{}{1}{12}}{133}, & , & , & , & , & , & , & ,
\end{array}
$$

$$
\begin{array}{ccccccccccccccccccc}
, & , & , & , & , & , & , & \genfrac{}{}{0pt}{}{\genfrac{}{}{0pt}{}{1}{12}}{124}, & \genfrac{}{}{0pt}{}{\genfrac{}{}{0pt}{}{1}{12}}{224}, & \genfrac{}{}{0pt}{}{\genfrac{}{}{0pt}{}{1}{12}}{124}, & \genfrac{}{}{0pt}{}{\genfrac{}{}{0pt}{}{1}{12}}{133}, & \genfrac{}{}{0pt}{}{\genfrac{}{}{0pt}{}{1}{12}}{133}, & , & , & , & , & , & ,
\end{array}
$$

$$
\begin{array}{ccccccccccccccccccc}
, & , & , & , & , & , & , & , & \genfrac{}{}{0pt}{}{\genfrac{}{}{0pt}{}{1}{12}}{223}, & , & \genfrac{}{}{0pt}{}{\genfrac{}{}{0pt}{}{1}{12}}{124}, & \genfrac{}{}{0pt}{}{\genfrac{}{}{0pt}{}{1}{12}}{144}, & \genfrac{}{}{0pt}{}{\genfrac{}{}{0pt}{}{1}{12}}{133}, & , & , & , & , & ,
\end{array}
$$

$$
\begin{array}{ccccccccccccccccccc}
, & , & , & , & , & , & , & , & \genfrac{}{}{0pt}{}{\genfrac{}{}{0pt}{}{1}{12}}{223}, & , & , & \genfrac{}{}{0pt}{}{\genfrac{}{}{0pt}{}{1}{13}}{144}, & \genfrac{}{}{0pt}{}{\genfrac{}{}{0pt}{}{1}{12}}{124}, & , & , & , & , & ,
\end{array}
$$

$$
\begin{array}{ccccccccccccccccccc}
, & , & , & , & , & , & , & , & \genfrac{}{}{0pt}{}{\genfrac{}{}{0pt}{}{1}{12}}{223}, & , & , & , & \genfrac{}{}{0pt}{}{\genfrac{}{}{0pt}{}{1}{12}}{133}, & \genfrac{}{}{0pt}{}{\genfrac{}{}{0pt}{}{1}{12}}{134}, & \genfrac{}{}{0pt}{}{\genfrac{}{}{0pt}{}{1}{12}}{124}, & \genfrac{}{}{0pt}{}{\genfrac{}{}{0pt}{}{1}{12}}{124}, & , & ,
\end{array}
$$

$$
\begin{array}{ccccccccccccccccccc}
, & , & , & , & , & , & , & , & \genfrac{}{}{0pt}{}{\genfrac{}{}{0pt}{}{1}{12}}{223}, & , & , & , & \genfrac{}{}{0pt}{}{\genfrac{}{}{0pt}{}{1}{12}}{133}, & \genfrac{}{}{0pt}{}{\genfrac{}{}{0pt}{}{1}{12}}{133}, & , & \genfrac{}{}{0pt}{}{\genfrac{}{}{0pt}{}{1}{12}}{124}, & \genfrac{}{}{0pt}{}{\genfrac{}{}{0pt}{}{1}{12}}{124}, & \genfrac{}{}{0pt}{}{\genfrac{}{}{0pt}{}{1}{12}}{124},
\end{array}
$$

The solitons $S_1^{(1)}$ and $S_2^{(2)}$ do not move by themselves as their bare speed is zero under $T_l^{(3)}$. See (I). One observes that the phase shift of $S_3^{(3)}$ vs $S_1^{(1)}$ collision is $C_{31}\min(3,1)=0$, whereas the one for $S_3^{(3)}$ vs $S_2^{(2)}$ collision is $C_{32}\min(3,2)=-2$ in agreement with (2.40). The intermediate states contain the non-vacuum local states $\genfrac{}{}{0pt}{}{\genfrac{}{}{0pt}{}{1}{12}}{224}, \genfrac{}{}{0pt}{}{\genfrac{}{}{0pt}{}{1}{12}}{144}, \genfrac{}{}{0pt}{}{\genfrac{}{}{0pt}{}{1}{13}}{144}, \genfrac{}{}{0pt}{}{\genfrac{}{}{0pt}{}{1}{12}}{134}$ which are not included in the elementary excitations $s_1, s_2, s_3$ in (2.37).

**Example 2.6.** Let us demonstrate how a generic state is decomposed into solitons as indicated in (III). We pick the state on the top line and first apply $T_l^{(3)}$ with $l \geq 1$ repeatedly to get

$$
\begin{array}{ccccccccc}
, & \genfrac{}{}{0pt}{}{\genfrac{}{}{0pt}{}{1}{13}}{134}, & \genfrac{}{}{0pt}{}{\genfrac{}{}{0pt}{}{1}{23}}{134}, & , & , & , & , & , & ,
\end{array}
$$

$$
\begin{array}{ccccccccc}
, & \genfrac{}{}{0pt}{}{\genfrac{}{}{0pt}{}{1}{12}}{133}, & \genfrac{}{}{0pt}{}{\genfrac{}{}{0pt}{}{1}{32}}{143}, & \genfrac{}{}{0pt}{}{\genfrac{}{}{0pt}{}{1}{12}}{134}, & , & , & , & , & ,
\end{array}
$$

$$
\begin{array}{ccccccccc}
, & \genfrac{}{}{0pt}{}{\genfrac{}{}{0pt}{}{1}{12}}{133}, & \genfrac{}{}{0pt}{}{\genfrac{}{}{0pt}{}{2}{13}}{134}, & \genfrac{}{}{0pt}{}{\genfrac{}{}{0pt}{}{1}{12}}{133}, & \genfrac{}{}{0pt}{}{\genfrac{}{}{0pt}{}{1}{12}}{124}, & , & , & , & ,
\end{array}
$$

$$
\begin{array}{ccccccccc}
, & \genfrac{}{}{0pt}{}{\genfrac{}{}{0pt}{}{1}{12}}{133}, & \genfrac{}{}{0pt}{}{\genfrac{}{}{0pt}{}{1}{12}}{133}, & \genfrac{}{}{0pt}{}{\genfrac{}{}{0pt}{}{1}{13}}{234}, & , & \genfrac{}{}{0pt}{}{\genfrac{}{}{0pt}{}{1}{12}}{124}, & , & , & ,
\end{array}
$$

$$
\begin{array}{ccccccccc}
, & \genfrac{}{}{0pt}{}{\genfrac{}{}{0pt}{}{1}{12}}{133}, & \genfrac{}{}{0pt}{}{\genfrac{}{}{0pt}{}{1}{12}}{133}, & \genfrac{}{}{0pt}{}{\genfrac{}{}{0pt}{}{1}{12}}{233}, & \genfrac{}{}{0pt}{}{\genfrac{}{}{0pt}{}{1}{12}}{134}, & , & \genfrac{}{}{0pt}{}{\genfrac{}{}{0pt}{}{1}{12}}{124}, & , & ,
\end{array}
$$

$$
\begin{array}{ccccccccc}
, & \genfrac{}{}{0pt}{}{\genfrac{}{}{0pt}{}{1}{12}}{133}, & \genfrac{}{}{0pt}{}{\genfrac{}{}{0pt}{}{1}{12}}{133}, & \genfrac{}{}{0pt}{}{\genfrac{}{}{0pt}{}{1}{12}}{233}, & \genfrac{}{}{0pt}{}{\genfrac{}{}{0pt}{}{1}{12}}{133}, & \genfrac{}{}{0pt}{}{\genfrac{}{}{0pt}{}{1}{12}}{124}, & , & \genfrac{}{}{0pt}{}{\genfrac{}{}{0pt}{}{1}{12}}{124}, & ,
\end{array}
$$

$$
\begin{array}{ccccccccc}
, & \genfrac{}{}{0pt}{}{\genfrac{}{}{0pt}{}{1}{12}}{133}, & \genfrac{}{}{0pt}{}{\genfrac{}{}{0pt}{}{1}{12}}{133}, & \genfrac{}{}{0pt}{}{\genfrac{}{}{0pt}{}{1}{12}}{233}, & \genfrac{}{}{0pt}{}{\genfrac{}{}{0pt}{}{1}{12}}{133}, & , & \genfrac{}{}{0pt}{}{\genfrac{}{}{0pt}{}{1}{12}}{124}, & , & \genfrac{}{}{0pt}{}{\genfrac{}{}{0pt}{}{1}{12}}{124}, &
\end{array}
$$

The two color 3 solitons $S_1^{(3)}$ are separated. Their distance is 1 which is indeed not less than their common length in agreement with the last claim in (III). The two solitons can be taken away to the right by further applying $T_l^{(3)}$ without changing their mutual distance. It allows us to focus on the remaining left part of the state. Applying $T_l^{(2)}$ with $l \geq 3$ to it successively, we find

$$
\begin{array}{ccccccccccccccccccc}
, & \genfrac{}{}{0pt}{}{\genfrac{}{}{0pt}{}{1}{12}}{133}, & \genfrac{}{}{0pt}{}{\genfrac{}{}{0pt}{}{1}{12}}{133}, & \genfrac{}{}{0pt}{}{\genfrac{}{}{0pt}{}{1}{12}}{233}, & \genfrac{}{}{0pt}{}{\genfrac{}{}{0pt}{}{1}{12}}{133}, & , & , & , & , & , & , & , & , & , & , & , & , & ,
\end{array}
$$

$$
\begin{array}{ccccccccccccccccccc}
, & , & , & \genfrac{}{}{0pt}{}{\genfrac{}{}{0pt}{}{1}{12}}{323}, & , & \genfrac{}{}{0pt}{}{\genfrac{}{}{0pt}{}{1}{12}}{133}, & \genfrac{}{}{0pt}{}{\genfrac{}{}{0pt}{}{1}{12}}{133}, & \genfrac{}{}{0pt}{}{\genfrac{}{}{0pt}{}{1}{12}}{133}, & , & , & , & , & , & , & , & , & , & ,
\end{array}
$$

$$
\begin{array}{ccccccccccccccccccc}
, & , & , & \genfrac{}{}{0pt}{}{\genfrac{}{}{0pt}{}{1}{22}}{133}, & , & , & \genfrac{}{}{0pt}{}{\genfrac{}{}{0pt}{}{1}{12}}{133}, & \genfrac{}{}{0pt}{}{\genfrac{}{}{0pt}{}{1}{12}}{133}, & \genfrac{}{}{0pt}{}{\genfrac{}{}{0pt}{}{1}{12}}{133}, & , & , & , & , & , & , & , & , & ,
\end{array}
$$

$$
\begin{array}{ccccccccccccccccccc}
, & , & , & \genfrac{}{}{0pt}{}{\genfrac{}{}{0pt}{}{1}{12}}{233}, & , & , & , & \genfrac{}{}{0pt}{}{\genfrac{}{}{0pt}{}{1}{12}}{133}, & \genfrac{}{}{0pt}{}{\genfrac{}{}{0pt}{}{1}{12}}{133}, & \genfrac{}{}{0pt}{}{\genfrac{}{}{0pt}{}{1}{12}}{133}, & , & , & , & , & , & , & , & ,
\end{array}
$$

$$
\begin{array}{ccccccccccccccccccc}
, & , & , & \genfrac{}{}{0pt}{}{\genfrac{}{}{0pt}{}{1}{12}}{223}, & \genfrac{}{}{0pt}{}{\genfrac{}{}{0pt}{}{1}{12}}{133}, & , & , & , & , & , & \genfrac{}{}{0pt}{}{\genfrac{}{}{0pt}{}{1}{12}}{133}, & \genfrac{}{}{0pt}{}{\genfrac{}{}{0pt}{}{1}{12}}{133}, & \genfrac{}{}{0pt}{}{\genfrac{}{}{0pt}{}{1}{12}}{133}, & , & , & , & , & ,
\end{array}
$$

$$
\begin{array}{ccccccccccccccccccc}
, & , & , & \genfrac{}{}{0pt}{}{\genfrac{}{}{0pt}{}{1}{12}}{223}, & \genfrac{}{}{0pt}{}{\genfrac{}{}{0pt}{}{1}{12}}{133}, & , & , & , & , & , & , & , & \genfrac{}{}{0pt}{}{\genfrac{}{}{0pt}{}{1}{12}}{133}, & \genfrac{}{}{0pt}{}{\genfrac{}{}{0pt}{}{1}{12}}{133}, & \genfrac{}{}{0pt}{}{\genfrac{}{}{0pt}{}{1}{12}}{133}, & , & ,
\end{array}
$$

Thus the color 2 solitons $S_1^{(2)}$ and $S_3^{(2)}$ are separated to the right leaving the color 1 soliton $S_1^{(1)}$ in the left. These procedures achieve (2.41)–(2.43).

**Remark 2.7.** Instead of (2.41) one can achieve the asymptotic decomposition

$$
\ldots (\text{color } a_1 \text{ solitons}) \ldots \ldots (\text{color } a_2 \text{ solitons}) \ldots \quad \ldots \quad \ldots (\text{color } a_{n-1} \text{ solitons}) \ldots \tag{2.45}
$$

for arbitrary permutation $a_1, a_2, \ldots, a_{n-1}$ of $1, 2, \ldots, n-1$. As indicated by Example 2.6, it is done by applying $(T_\infty^{(a_1)})^{M_1}(T_\infty^{(a_2)})^{M_2} \cdots (T_\infty^{(a_{n-1})})^{M_{n-1}}$ with $M_1, M_2, \ldots, M_{n-1} \gg 1$.

**Remark 2.8.** The tableaux letter $a \in [1, n]$ carries the $sl_n$ weight $\varpi_1 - \alpha_1 - \cdots - \alpha_{a-1}$. In this counting, the elementary excitation $s_a$ in (2.38) has the extra weight $-\alpha_a$ compared from the background vac (2.21). Thus the soliton $S_i^{(a)}$ (2.39) carries the weight $-i\alpha_a$.

In terms of letters in the boxes of semistandard tableaux, the presence of a soliton $S_i^{(a)}$ changes the letter $a$ into $a + 1$ for $i$ times compared with the vacuum background. On the other hand, vac in (2.21) contains a letter $a$ for $(n - a)$ times. Therefore the number $\lambda_a$ of the letter $a$ in the $L$ site system is

$$\lambda_a = L(n - a) - \sum_{i=1}^{\infty} i m_i^{(a)} + \sum_{i=1}^{\infty} i m_i^{(a-1)} \quad (a \in [1, n]), \tag{2.46}$$

where $m_i^{(a)}$ the number of solitons $S_i^{(a)}$ and $m_i^{(0)} = 0$ by convention.[3] This formula is nothing but (C.14)[4] with (C.10) under the special choice $N = (n - 1)L$ and $k_i \equiv i \in \mathbb{Z}_{n-1}$ with $i \in [1, n-1]$. In terms of $\rho_i^{(a)}$ and $\varepsilon_i^{(a)}$ that will be introduced in (3.16) and (3.17), the density $\varrho_a := \lambda_a/L$ of the tableau letter $a$ is given by

$$\varrho_a = n - a - \sum_{i=1}^{\infty} i \rho_i^{(a)} + \sum_{i=1}^{\infty} i \rho_i^{(a-1)} = n - a - \varepsilon_\infty^{(a)} + \varepsilon_\infty^{(a-1)} \quad (a \in [1, n]), \tag{2.47}$$

where $\rho_i^{(0)} = \varepsilon_\infty^{(0)} = 0$.

## 2.6 Inverse scattering method

In Appendix C a brief exposition is given on the KSS bijection $\Phi$ between the set of *highest paths* $\mathcal{P}_+(\mathcal{B}, \lambda)$ and *rigged configurations* $\mathrm{RC}(\mathcal{B}, \lambda)$. States of our cBBS satisfying the boundary condition are examples of the former with $\mathcal{B} = B^{\otimes L}$. See the remark in the end of Appendix C. The array $\lambda = (\lambda_1, \ldots, \lambda_n)$ specifies that the number of letter $a$ in a state is $\lambda_a$. On the other hand, a rigged configuration is a combinatorial object like (C.12) visualizing the collection of *strings* which are triplets (color, length, rigging) as in $S_0$ (C.11).

The time evolution $T_l^{(k)}$ of the BBS induces that of rigged configurations via the commutative diagram:[5]

$$
\begin{array}{ccc}
\mathcal{P}_+(\mathcal{B}, \lambda) & \xrightarrow{\Phi} & \mathrm{RC}(\mathcal{B}, \lambda) \\
T_l^{(k)} \downarrow & & \downarrow T_l^{(k)} \\
\mathcal{P}_+(\mathcal{B}, \lambda) & \xrightarrow{\Phi} & \mathrm{RC}(\mathcal{B}, \lambda)
\end{array} \tag{2.48}
$$

A remarkable feature of this scheme is that it achieves the *linearization* [38]:

$$T_l^{(k)} : \{(a_i, j_i, r_i)\} \mapsto \{(a_i, j_i, r_i + \delta_{k,a_i} \min(l, j_i))\}, \tag{2.49}$$

---

[3]The formula (2.46) is obvious for asymptotic states. General case follows from it and the fact that $m_i^{(a)}$'s are conserved quantities.

[4]The formula (C.14) is given for rigged configurations, but it is known to agree with (C.1) under the KSS bijection.

[5]We use the same notation $T_l^{(k)}$ to also represent the time evolution of the rigged configurations since they are to be identified via the bijection $\Phi$.

where rigged configurations are represented as multisets as in (C.7). It implies that the partial data $\{(a_i, j_i)\}$ forms the complete set of conserved quantities (action variables) and the riggings undergo a straight motion (angle variables). In short, rigged configurations are action-angle variables of our cBBS, and the maps $\Phi$ and $\Phi^{-1}$ are direct and inverse scattering transformations. The scheme (2.48) achieves the solution of the initial value problem of cBBS by the inverse scattering method as $T_l^{(k)} = \Phi^{-1} \circ T_l^{(k)} \circ \Phi$.

Take a state $b_1 \otimes \cdots \otimes b_L$ of cBBS and let $m_j^{(a)}$ be the number of color $a$ length $j$ strings in the rigged configuration $\Phi(b_1 \otimes \cdots \otimes b_L)$. We set $\mathcal{E}_j^{(a)} = \mathcal{E}_j^{(a)}(b_1 \otimes \cdots \otimes b_L) = \sum_{k \geq 1} \min(j, k) m_k^{(a)}$ according to (C.10). Since $m_j^{(a)}$ is determined only by the action variables $\{(a_i, j_i)\}$, the quantity $\mathcal{E}_j^{(a)}$ is conserved under the time evolution. On the other hand recall the conserved energy of a state $E_l^{(k)} = E_l^{(k)}(b_1 \otimes \cdots \otimes b_L)$ introduced in (2.30). The two are known to coincide:

$$E_l^{(k)} = \mathcal{E}_l^{(k)}. \tag{2.50}$$

Due to the invariance under the time evolution, the proof reduces to the doable case of asymptotic states where solitons are all far apart. See for example [39].

Recall that $m_j^{(a)}$ in $E_l^{(k)}$ (2.44) was the number of color $a$ length $j$ *solitons* contained in a state, whereas the one in $\mathcal{E}_l^{(k)}$ is the number of color $a$ length $j$ *strings*. Therefore (2.50) implies the *soliton/string correspondence* [38]

$$\text{soliton} = \text{string}.$$

It has played a key role in the Bethe ansatz study of the randomized BBS [11, 17].

To illustrate, consider the state on the top line of Example 2.6 in the previous subsection:

$$\mathfrak{s} = \begin{matrix} 1 \\ 12 \\ 123 \end{matrix} \otimes \begin{matrix} 1 \\ 12 \\ 123 \end{matrix} \otimes \begin{matrix} 1 \\ 12 \\ 123 \end{matrix} \otimes \begin{matrix} 1 \\ 13 \\ 134 \end{matrix} \otimes \begin{matrix} 1 \\ 23 \\ 134 \end{matrix} \otimes \begin{matrix} 1 \\ 12 \\ 123 \end{matrix} \otimes \begin{matrix} 1 \\ 12 \\ 123 \end{matrix} \otimes \begin{matrix} 1 \\ 12 \\ 123 \end{matrix} \in \mathcal{P}_+(B^{\otimes 8}, (23, 13, 10, 2)), \tag{2.51}$$

where a few $\text{vac} = \begin{smallmatrix} 1 \\ 12 \\ 123 \end{smallmatrix}$ are supplied in the front and in the tail and displayed explicitly. Such operators do not influence the relevant rigged configurations essentially. See the explanation in the end of Appendix C. From the demonstration in the previous subsection, we know that $\mathfrak{s}$ "consists of" solitons $S_1^{(3)}, S_1^{(3)}$ of color 3, solitons $S_3^{(2)}, S_1^{(2)}$ of color 2 and a soliton $S_1^{(1)}$ of color 1 including multiplicity. This indeed matches the image of the map $\Phi$:

$$\Phi(\mathfrak{s}) = \quad \square\, 5 \qquad \begin{matrix}\square\square\square\square\, 0 \\ \square\square\,3\end{matrix} \qquad \begin{matrix}\square\,3 \\ \square\,3\end{matrix}\;, \tag{2.52}$$

where the precise meaning of the above graphical representation for the rigged configuration is detailed in Appendix C.2.

# 3 Randomized cBBS

## 3.1 Generalized Gibbs ensemble

Now we introduce the randomized version of cBBS which corresponds to the distribution of the initial states according to the generalized Gibbs ensemble (GGE). The GGE partition function for size $L$ system $B^{\otimes L}$ is the sum of the Boltzmann factor $\exp(-\sum_{r,l} \beta_l^{(r)} E_l^{(r)})$ over all the

states of the cBBS, where $E_l^{(k)}$ is the conserved energy. See (2.30), (2.44), (2.50) and (C.10) for the definition and the properties. Especially from (2.44), the system can really be regarded as a gas of interacting solitons with various colors and lengths. The parameter $\beta_l^{(r)}$ is the inverse temperature associated with $E_l^{(r)}$. We are interested in the asymptotic behavior as $L \to \infty$.

At this stage a nontrivial question arises as how to incorporate properly the boundary condition that all the distant local states are vac (2.20). Here we propose that it is done by restricting the state sum to the *highest* ones that are defined in Appendix C.1. Similar treatment has been done in [11,17,27], where more detailed justification was made especially in the first reference. Thus the GGE partition function of our concern is

$$Z_L(\{\beta_l^{(r)}\}) = \sum_{\text{highest states} \in B^{\otimes L}} \exp(-\sum_{(r,l)\in\mathcal{I}} \beta_l^{(r)} E_l^{(r)}). \tag{3.1}$$

By synthesizing various results it can be rewritten as follows:

$$Z_L(\{\beta_l^{(r)}\}) = \sum_{\lambda} \sum_{\mathfrak{s}\in\mathcal{P}_+(B^{\otimes L},\lambda)} \exp(-\sum_{(r,l)\in\mathcal{I}} \beta_l^{(r)} E_l^{(r)}(\mathfrak{s})) \qquad \text{(definition (C.5))}$$

$$= \sum_{\lambda} \sum_{S\in\mathrm{RC}(B^{\otimes L},\lambda)} \exp(-\sum_{(r,l)\in\mathcal{I}} \beta_l^{(r)} \mathcal{E}_l^{(r)}(S))$$

$$\qquad\qquad\qquad ((2.50) \text{ and 1:1 correspondence (C.17)})$$

$$= \sum_{\lambda} \sum_{\{m_j^{(a)}\}}^{(\lambda)} \exp(-\sum_{(r,l)\in\mathcal{I}} \beta_l^{(r)} \sum_{j\geq 1} \min(j,l) m_j^{(r)}) \prod_{(a,j)\in\mathcal{I}} \binom{p_j^{(a)} + m_j^{(a)}}{m_j^{(a)}}$$

$$\qquad\qquad\qquad\qquad \text{(from (C.10), (C.16))}$$

$$= \sum_{\{m_j^{(a)}\}} \exp(-\sum_{(r,l)\in\mathcal{I}} \beta_l^{(r)} \sum_{j\geq 1} \min(j,l) m_j^{(r)}) \prod_{(a,j)\in\mathcal{I}} \binom{p_j^{(a)} + m_j^{(a)}}{m_j^{(a)}}. \tag{3.2}$$

Here the vacancy $p_i^{(a)}$ defined in general by (C.10) takes the form

$$p_i^{(a)} = L - \sum_{(b,j)\in\mathcal{I}} C_{ab} \min(i,j) m_j^{(b)}, \tag{3.3}$$

for our cBBS reflecting the fact that the states belong to $B^{\otimes L}$ with $B$ given by (2.21). The constraint (C.8) is taken into account by setting $\binom{K}{J} = 0$ unless $J \in [0,K]$. In (3.2) we are summing over the number $m_j^{(a)}$ of solitons $S_j^{(a)}$. In this sense $Z_L(\{\beta_l^{(r)}\})$ is a GGE partition function for the grand canonical ensemble of solitons.

**Remark 3.1.** The appearence of the phase shift $\Delta = C_{ab}\min(i,j)$ (defined in (2.40)) in (3.3) is not a coincidence. It is indeed a general property in Bethe ansatz integrable systems that such a shift appears in the relation between the hole and particle densities and is related to the TBA kernel (see also Remark 2.3).

## 3.2 I.I.D. randomness

From here we concentrate on the situation where local states $b \in B = B^{1,1} \otimes \cdots \otimes B^{n-1,1}$ obey an i.i.d. randomness. More specifically we consider an i.i.d. measure $\mathbb{P}(b \in B)$ including

the parameters $z_1 > z_2 > \cdots > z_n > 0$ as follows:

$$\mathbb{P}(b = c_1 \otimes \cdots \otimes c_{n-1} \in B) = \mathbb{P}^{(1)}(c_1) \cdots \mathbb{P}^{(n-1)}(c_{n-1}), \tag{3.4}$$

$$\mathbb{P}^{(k)}\big(c = (a_1, \ldots, a_k) \in B^{k,1}\big) = \frac{z_{a_1} \cdots z_{a_k}}{\sum_{c \in B^{k,1}} z_{a_1} \cdots z_{a_k}}, \tag{3.5}$$

where $a_i$ denotes the entry of the tableau $c \in B^{k,1}$ in the $i$ th box from the top. The formula (3.5) implies that the probability of occurrence of $a \in [1, n]$ is proportional to $z_a$. In this sense, $z_1, \ldots, z_n$ are *tableau variables* representing the fugacities of the letters $1, \ldots, n$. Their ambiguity due to the invariance of (3.5) under the change $z_a \to u z_a$ for a constant $u$ will be fixed in the rightmost condition in (3.7) below. The denominator of (3.5) is the $k$ th elementary symmetric function which is $Q_1^{(k)}$ in the notation (3.10).

For a partition $\nu = (\nu_1, \nu_2, \ldots, \nu_n)$, let $s_\nu = s_\nu(z_1, \ldots, z_n)$ denote the Schur polynomial[6] [13]:

$$s_\nu(z_1, \ldots, z_n) = \frac{\det(z_j^{\nu_i + n - i})_{i,j=1}^n}{\det(z_j^{n-i})_{i,j=1}^n}. \tag{3.6}$$

Partitions are identified with Young diagrams as usual. Schur polynomials are irreducible characters of finite dimensional $gl_n$ modules which can be restricted to those for $sl_n$ modules. Reflecting this fact we will use further sets of variables $w_a, x_a$ related to $z_a$ as follows:

$$z_a = w_{a-1}^{-1} w_a \qquad (1 \le a \le n), \qquad w_0 = w_n = 1, \qquad z_1 z_2 \cdots z_n = 1, \tag{3.7}$$

$$w_a = e^{\varpi_a} = z_1 z_2 \cdots z_a \qquad (0 \le a \le n), \qquad \varpi_0 = \varpi_n = 0, \tag{3.8}$$

$$x_a = e^{-\alpha_a} = z_a^{-1} z_{a+1} = \prod_{b=1}^{n-1} w_b^{-C_{ab}} \qquad (1 \le a \le n - 1). \tag{3.9}$$

The variables $w_a$ and $x_a$ are formal exponential of the fundamental weight $\varpi_a$ and the negated simple root $-\alpha_a$ regarded as parameters. See Sec. 2.1. The last relation in (3.7) leads to $s_\nu = s_{\tilde\nu}$ where $\tilde\nu = (\nu_1 - \nu_n, \ldots, \nu_{n-1} - \nu_n, 0)$. We use a special notation for rectangle $\nu$'s as

$$Q_i^{(a)} = s_{(i^a)}(z_1, \ldots, z_n). \tag{3.10}$$

It satisfies the Q-system [31, 37]

$$(Q_i^{(a)})^2 = Q_{i-1}^{(a)} Q_{i+1}^{(a)} + Q_i^{(a-1)} Q_i^{(a+1)}, \tag{3.11}$$

for $(a, i) \in \mathcal{I}$ with $Q_i^{(0)} = Q_i^{(n)} = 1$. To validate (3.11) also at $i = 0$ with $Q_0^{(a)} = 1$, we employ the convention $Q_{-1}^{(a)} = 0$.

Let us show that the i.i.d. measure (3.4) is the special $(n-1)$-parameter reduction of the GGE in Sec. 3.1. Denote the weight of a state $\mathfrak{s} \in B^{\otimes L}$ in the sense of (C.1) by $\text{wt}(\mathfrak{s}) = (\lambda_1, \ldots, \lambda_n)$. Then the formulas (3.4) and (3.5) mean that $\mathfrak{s}$ is realized with the probability proportional to $z_1^{\lambda_1} \cdots z_n^{\lambda_n}$. Since the KKS bijection is weight preserving (see (C.17)), the RHS of (C.1) and (C.14) may be identified. Thus the factor $\prod_{a=1}^n z_a^{\lambda_a}$ is proportional to $\prod_{a=1}^n z_a^{-\mathcal{E}_\infty^{(a)} + \mathcal{E}_\infty^{(a-1)}}$, where we have dropped $\sum_{i=1}^N \theta(k_i \ge a)$ in (C.14) since it is the constant $(n-a)L$ for our states in $B^{\otimes L}$. (Note $\mathcal{E}_j^{(0)} = \mathcal{E}_j^{(n)} = 0$ as mentioned there.) We can further replace $\mathcal{E}_\infty^{(a)}$ by $E_\infty^{(a)}$ thanks to (2.50). Therefore the probability of the state $\mathfrak{s}$ is proportional to

$$z_1^{-E_\infty^{(1)}} z_2^{-E_\infty^{(2)} + E_\infty^{(1)}} \cdots z_{n-1}^{-E_\infty^{(n-1)} + E_\infty^{(n-2)}} z_n^{E_\infty^{(n-1)}} = \prod_{a=1}^{n-1} \Big(\frac{z_{a+1}}{z_a}\Big)^{E_\infty^{(a)}} = \exp\Big(-\sum_{a=1}^{n-1} \alpha_a E_\infty^{(a)}\Big). \tag{3.12}$$

---

[6]The Schur polynomial $s_\nu$ should not be confused with $s_a$ in (2.38).

Comparing this with (3.1) we see that the i.i.d. measure (3.4) corresponds to the special case of GGE in which the inverse temperatures $\beta_l^{(a)}$ is zero except

$$\beta_\infty^{(a)} = \alpha_a \qquad (a \in [1, n-1]). \tag{3.13}$$

Thus the inverse temperatures (with index $\infty$) can be identified with the simple roots. We will nonetheless allow co-existence of the symbols $\beta_\infty^{(a)}$ and $\alpha_a$ in what follows. The identification (3.13) will also be reconfirmed after (3.25). From (3.9), the regime of parameters we consider is also specified in the variables $z_a, \alpha_a, x_a$ as

$$z_1 > \cdots > z_n > 0 \ (z_1 \cdots z_n = 1), \qquad \alpha_1, \ldots, \alpha_{n-1} > 0, \qquad x_1, \ldots, x_{n-1} < 1. \tag{3.14}$$

In particular $x_a \to 0$ and $x_a \to 1$ correspond to the zero and infinite limit of the temperature $1/\beta_\infty^{(a)}$, respectively.

**Remark 3.2.** The *dilute* limit where no soliton is allowed is given by $z_1 \gg z_2 \gg \cdots \gg z_n$ or equivalently $x_1, \ldots, x_n \to 0$. In fact, $\mathbb{P}^{(k)}(c) \to \delta_{c, u_{k,1}}$ in (3.5) hence $\mathbb{P}(b) \to \delta_{b, \text{vac}}$ in (3.4) hold in this limit. One also has $Q_i^{(a)}/z_a^i \to 1$. See (2.12) and (2.21) for the definition of $u_{k,1}$ and vac.

Now the partition function (3.2) is reduced to

$$Z_L(\{\beta_\infty^{(r)}\}) = \sum_{\{m_j^{(a)}\}} \prod_{(a,j) \in \mathcal{I}} \exp(-\beta_\infty^{(a)} j m_j^{(a)}) \binom{p_j^{(a)} + m_j^{(a)}}{m_j^{(a)}}. \tag{3.15}$$

## 3.3 Thermodynamic Bethe ansatz

In the large $L$ limit, the dominant contribution in the sum (3.15) comes from those $\{m_j^{(a)}\}$ having the $L$-linear asymptotic behavior

$$m_i^{(a)} \simeq L\rho_i^{(a)}, \qquad p_i^{(a)} \simeq L\sigma_i^{(a)}, \qquad \mathcal{E}_i^{(a)} \simeq L\varepsilon_i^{(a)}, \tag{3.16}$$

$$\sigma_i^{(a)} = 1 - \sum_{b=1}^{n-1} C_{ab}\varepsilon_i^{(b)}, \qquad \varepsilon_i^{(a)} = \sum_{j \geq 1} \min(i,j)\rho_j^{(a)}, \tag{3.17}$$

where the second line follows from the first by (3.3). The relation (3.17) is a spectral parameter free version of the Bethe equation in terms of the string density $\rho_i^{(a)}$ and the hole density $\sigma_i^{(a)}$. One should seek $\rho = (\rho_i^{(a)})$ that minimizes the free energy per site

$$F[\rho] = \sum_{a=1}^{n-1} \beta_\infty^{(a)} \sum_{i=1}^{s} i\rho_i^{(a)} - \sum_{a=1}^{n-1} \sum_{i=1}^{s} \Big( (\rho_i^{(a)} + \sigma_i^{(a)}) \log(\rho_i^{(a)} + \sigma_i^{(a)}) - \rho_i^{(a)} \log \rho_i^{(a)} - \sigma_i^{(a)} \log \sigma_i^{(a)} \Big). \tag{3.18}$$

This is $(-1/L)$ times logarithm of the summand in (3.15) to which Stirling's formula has been applied. The scaling (3.16) is consistent with the extensive property of the free energy, which has enabled us to remove the system size $L$ as a common overall factor. We have introduced a temporary cut-off $s$ for the length $i$ of solitons. It will be taken to infinity properly later. Accordingly the latter relation in (3.17) should be understood as $\varepsilon_i^{(a)} = \sum_{j=1}^{s} \min(i,j)\rho_j^{(a)}$.

From $\frac{\partial \sigma_j^{(b)}}{\partial \rho_i^{(a)}} = -C_{ab}\min(i,j)$, one finds that the condition $\frac{\partial F[\rho]}{\partial \rho_i^{(a)}} = 0$ is expressed as a TBA equation

$$-i\beta_\infty^{(a)} + \log(1 + (y_i^{(a)})^{-1}) = \sum_{b=1}^{n-1} C_{ab} \sum_{j=1}^{s} \min(i,j)\log(1 + y_j^{(b)}), \qquad (3.19)$$

in terms of the ratio[7]

$$y_i^{(a)} = \frac{\rho_i^{(a)}}{\sigma_i^{(a)}}. \qquad (3.20)$$

The corresponding maximal value $\mathcal{F}$ of the free energy per site $F[\rho]$ (3.18) is obtained by using (3.17), (3.19) and (3.20). The result reads

$$\mathcal{F} = -\sum_{a=1}^{n-1}\sum_{i=1}^{s}\log(1 + y_i^{(a)}). \qquad (3.21)$$

The TBA equation (3.19) is equivalent to the constant Y-system

$$\frac{(1 + (y_i^{(a)})^{-1})^2}{(1 + (y_{i-1}^{(a)})^{-1})(1 + (y_{i+1}^{(a)})^{-1})} = \prod_{b=1}^{n-1}(1 + y_i^{(b)})^{C_{ab}} \qquad (1 \le i \le s), \qquad (3.22)$$

with the boundary condition

$$(y_0^{(a)})^{-1} = 0, \quad 1 + (y_{s+1}^{(a)})^{-1} = e^{\beta_\infty^{(a)}}(1 + (y_s^{(a)})^{-1}). \qquad (3.23)$$

The Y-system (3.22) follows from the Q-system (3.11) by the substitution (cf. [37, Prop. 14.1])

$$y_i^{(a)} = \frac{Q_i^{(a-1)}Q_i^{(a+1)}}{Q_{i-1}^{(a)}Q_{i+1}^{(a)}}, \qquad 1 + (y_i^{(a)})^{-1} = \prod_{b=1}^{n-1}(Q_i^{(b)})^{C_{ab}}, \qquad 1 + y_i^{(a)} = \frac{(Q_i^{(a)})^2}{Q_{i-1}^{(a)}Q_{i+1}^{(a)}}. \qquad (3.24)$$

As for the boundary condition (3.23), the first relation is valid due to the convention $Q_{-1}^{(a)} = 0$ mentioned after (3.11). On the other hand the second relation is translated into

$$e^{\beta_\infty^{(a)}} = \prod_{b=1}^{n-1}\left(\frac{Q_{s+1}^{(b)}}{Q_s^{(b)}}\right)^{C_{ab}}. \qquad (3.25)$$

At this point we let $s$ tend to infinity. The result [40, Th. 7.1 (C)] tells that $\lim_{s\to\infty}(Q_{s+1}^{(a)}/Q_s^{(a)}) = e^{\varpi_a}$ in the regime $e^{\alpha_1},\ldots,e^{\alpha_{n-1}} > 1$ under consideration. Thus the large $s$ limit of (3.25) leads to $e^{\beta_\infty^{(a)}} = \prod_{b=1}^{n-1}(e^{\varpi_b})^{C_{ab}} = e^{\alpha_a}$ by (3.9). This is consistent with (3.13). We also remark that $\lim_{s\to\infty}(Q_{s+1}^{(a)}/Q_s^{(a)}) = e^{\varpi_a}$ and the last relation in (3.24) tells that $\lim_{i\to\infty} y_i^{(a)} = 0$, therefore (3.20) indicates $\lim_{i\to\infty}\rho_i^{(a)} = 0$ which is indeed necessary for the convergence of $\varepsilon_\infty^{(a)} = \sum_{i=1}^\infty i\rho_i^{(a)}$.

Substituting the last expression in (3.24) into the free energy density (3.21), we get

$$\mathcal{F} = -\sum_{a=1}^{n-1}\log\left(\frac{Q_1^{(a)}Q_s^{(a)}}{Q_{s+1}^{(a)}}\right) \xrightarrow{s\to\infty} -\sum_{a=1}^{n-1}\log\overline{Q}_1^{(a)}, \qquad (3.26)$$

$$\overline{Q}_i^{(a)} = e^{-i\varpi_a}Q_i^{(a)} = 1 + \cdots \in \mathbb{Z}_{\ge 0}[x_1,\ldots,x_{n-1}]. \qquad (3.27)$$

---

[7]This $y_i^{(a)}$ is the inverse of $Y_i^{(a)}$ in [17, eq.(52)].

The quantity $\overline{Q}_i^{(a)}$ is the *normalized* character of the irreducible $sl_n$ module with highest weight $i\varpi_a$.

From (3.17) one sees that $\varepsilon_\infty^{(a)} = \sum_{i \geq 1} i\rho_i^{(a)}$ is the total density of color $a$ solitons. Let $\epsilon_1, \ldots, \epsilon_{n-1}$ be the *expectation values* of $\varepsilon_\infty^{(1)}, \ldots, \varepsilon_\infty^{(n-1)}$. From (3.1), they are to be derived from the free energy density by the standard prescription $\epsilon_a = \frac{\partial \mathcal{F}}{\partial \beta_\infty^{(a)}}$. Further substitution of (3.26) into this leads to the *equation of state* of our cBBS as follows:

$$\epsilon_a = -\sum_{b=1}^{n-1} \frac{\partial \log \overline{Q}_1^{(b)}}{\partial \alpha_a} = x_a \frac{\partial}{\partial x_a} \log\big(\overline{Q}_1^{(1)} \cdots \overline{Q}_1^{(n-1)}\big) \qquad (a \in [1, n-1]), \qquad (3.28)$$

where we have written $\beta_\infty^{(a)}$ as $\alpha_a$ by (3.13) to express the RHS purely in terms of representation theoretical data. An analogous result on "non-complete BBS" was given in [17, eq. (66)].

**Example 3.3.** In the simplest case $n = 2$, $z_2 = z_1^{-1}$, $x_1 = z_1^{-2}$. Thus we have

$$Q_1^{(1)} = z_1 + z_2, \qquad \overline{Q}_1^{(1)} = 1 + x_1, \qquad \epsilon_1 = \frac{x_1}{1 + x_1} = \frac{z_2}{z_1 + z_2}, \qquad (3.29)$$

which agrees with the density of total number of "balls" 2. Similarly for $n = 3$, one has

$$Q_1^{(1)} = z_1 + z_2 + z_3, \qquad \overline{Q}_1^{(1)} = 1 + x_1 + x_1 x_2, \quad \epsilon_1 = \frac{x_1(1 + x_2)}{1 + x_1 + x_1 x_2} + \frac{x_1 x_2}{1 + x_2 + x_1 x_2}, \qquad (3.30)$$

$$Q_1^{(2)} = z_1 z_2 + z_1 z_3 + z_2 z_3, \quad \overline{Q}_1^{(2)} = 1 + x_2 + x_1 x_2, \quad \epsilon_2 = \frac{x_2(1 + x_1)}{1 + x_2 + x_1 x_2} + \frac{x_1 x_2}{1 + x_1 + x_1 x_2}. \qquad (3.31)$$

In this way one can relate the densities and the inverse temperatures.

To summarize so far, we have obtained the solution $\{y_i^{(a)}\}$ to the TBA equation $(3.19)|_{s=\infty}$ in (3.24), (3.10) and (3.6). They depend on $n-1$ independent parameters which may be taken either as the inverse temperatures $\beta_\infty^{(a)} = \alpha_a\,(1 \leq a < n)$ or the fugacities $z_a\,(1 \leq a \leq n)$ as in (3.7)–(3.9). They can further be related to the prescribed expectation values $\epsilon_a$ of $\varepsilon_\infty^{(a)} = \sum_{i \geq 1} i\rho_i^{(a)}\,(1 \leq a < n)$ by the equation of state (3.28).

The remaining task is to express the string and hole densities $\rho_i^{(a)}$ and $\sigma_i^{(a)}$ in terms of these variables. From (3.20) and (3.17), it suffices to determine $\varepsilon_i^{(a)}$. It is characterized by the second order difference equation and the boundary condition

$$(y_i^{(a)})^{-1}(-\varepsilon_{i-1}^{(a)} + 2\varepsilon_i^{(a)} - \varepsilon_{i+1}^{(a)}) + \sum_{b=1}^{n-1} C_{ab}\,\varepsilon_i^{(b)} = 1, \qquad (3.32)$$

$$\varepsilon_0^{(a)} = 0, \qquad \varepsilon_\infty^{(a)} = x_a \frac{\partial}{\partial x_a} \log\big(\overline{Q}_1^{(1)} \cdots \overline{Q}_1^{(n-1)}\big). \qquad (3.33)$$

The equation (3.32) is just the first relation in (3.17), where the first term is a disguised form of $\sigma_i^{(a)}$. The last relation in (3.33) is the postulate from the equation of state (3.28).

At this point we invoke [17, Th. 5.1]. In the setting of this paper, it states that for any $(r, l) \in \mathcal{I}$, the unique solution $h_i^{(a)} = h_i^{(a)}(r, l)$ to

$$(y_i^{(a)})^{-1}(-h_{i-1}^{(a)} + 2h_i^{(a)} - h_{i+1}^{(a)}) + \sum_{b=1}^{n-1} C_{ab}\,h_i^{(b)} = \delta_{a,r} \min(i, l), \qquad (3.34)$$

$$h_0^{(a)} = 0, \qquad h_\infty^{(a)} = x_a \frac{\partial}{\partial x_a} \log \overline{Q}_l^{(r)} \qquad (3.35)$$

is given by

$$h_i^{(a)}(r,l) = \frac{\sum_{\nu=(\nu_1,\ldots,\nu_n)}(\sum_{j=\max(a,r)+1}^n \nu_j)s_\nu}{s_{(i^a)}s_{(l^r)}} \quad (= h_l^{(r)}(a,i)), \tag{3.36}$$

where the outer sum in (3.36) runs over those Young diagrams $\nu$ labeling the irreducible $gl_n$ modules appearing in the decomposition of the tensor product of those corresponding to the rectangles $(i^a)$ and $(l^r)$. We note that the irreducible decomposition of two rectangles is multiplicity free.

**Example 3.4.** For generic $n$, the Littlewood-Richardson rule [13] leads to

$$h_i^{(1)}(1,l) = \frac{1}{s_{(i)}s_{(l)}} \sum_{j=1}^{\min(i,l)} j s_{(i+l-j,j)}, \tag{3.37}$$

$$h_i^{(2)}(1,l) = \frac{1}{s_{(i,i)}s_{(l)}} \sum_{j=1}^{\min(i,l)} j s_{(i+l-j,i,j)}, \tag{3.38}$$

$$h_i^{(2)}(2,l) = \frac{1}{s_{(i,i)}s_{(l,l)}} \sum_{j,j'} (2l-2j-j') s_{(i+j+j',i+j,l-j,l-j-j')}, \tag{3.39}$$

$$h_i^{(a)}(r,1) = \frac{1}{s_{(i^a)}s_{(1^r)}} \sum_{j=1}^{\min(r,n-a)} j s_{((i+1)^{r-j},i^{a-r+j},1^j)} \quad (1 \le r \le a), \tag{3.40}$$

$$= \frac{1}{s_{(i^a)}s_{(1^r)}} \sum_{j=1}^{\min(a,n-r)} j s_{(i+1)^{a-j},i^j,1^{r-a+j}} \quad (a \le r \le n). \tag{3.41}$$

The sum (3.39) is taken over $j,j' \ge 0$ such that $j+j' \le l$ and $\max(l-i,0) \le j \le l$. Given $n$, the sums (3.37)–(3.39) should be truncated so that the length of the partitions appearing in the indices of the Schur functions does not exceed $n$. For instance when $n=3$, (3.39) is reduced to $(s_{(i,i)}s_{(l,l)})^{-1}\sum_{0\le j<l}(l-j)s_{(i+l,i+j,l-j)}$. For $n=4$ (3.40) and (3.41) read

$$h_i^{(1)}(1,1) = \frac{s_{(i,1)}}{s_{(i)}s_{(1)}}, \qquad h_i^{(1)}(2,1) = \frac{s_{(i,1,1)}}{s_{(i)}s_{(1,1)}}, \qquad h_i^{(1)}(3,1) = \frac{s_{(i,1,1,1)}}{s_{(i)}s_{(1,1,1)}}, \tag{3.42}$$

$$h_i^{(2)}(1,1) = \frac{s_{(i,i,1)}}{s_{(i,i)}s_{(1)}}, \qquad h_i^{(2)}(2,1) = \frac{s_{(i+1,i,1)}+2s_{(i,i,1,1)}}{s_{(i,i)}s_{(1,1)}}, \quad h_i^{(2)}(3,1) = \frac{s_{(i+1,i,1,1)}}{s_{(i,i)}s_{(1,1,1)}}, \tag{3.43}$$

$$h_i^{(3)}(1,1) = \frac{s_{(i,i,i,1)}}{s_{(i,i,i)}s_{(1)}}, \qquad h_i^{(3)}(2,1) = \frac{s_{(i+1,i,i,1)}}{s_{(i,i,i)}s_{(1,1)}}, \qquad h_i^{(3)}(3,1) = \frac{s_{(i+1,i+1,i,1)}}{s_{(i,i,i)}s_{(1,1,1)}}. \tag{3.44}$$

Comparing the two systems (3.32), (3.33) and (3.34), (3.35), we find that the solution to the former is constructed as the superposition of the latter as

$$\varepsilon_i^{(a)} = \sum_{r=1}^{n-1} h_i^{(a)}(r,1). \tag{3.45}$$

From (3.40)-(3.41) we obtain the solution to (3.32), (3.33) as

$$\varepsilon_i^{(a)} = \frac{1}{s_{(i^a)}} \sum_{r=1}^{n-1} \frac{1}{s_{(1^r)}} \sum_{k=1}^{\min(a,r,n-a,n-r)} k\, s_{((i+1)^{\min(a,r)-k},i^{a+k-\min(a,r)},1^{r+k-\min(a,r)})}, \tag{3.46}$$

where $z_1 \cdots z_n = 1$ (3.7) is imposed on the Schur funciton (3.6).

**Example 3.5.** For small $n$, (3.46) reads as

$$n = 2; \quad \varepsilon_i^{(1)} = \frac{s_{(i,1)}}{s_{(i)}s_{(1)}} = \frac{x_1(1 - x_1^i)}{(1 + x_1)(1 - x_1^{i+1})}, \tag{3.47}$$

$$n = 3; \quad \varepsilon_i^{(1)} = \frac{1}{s_{(i)}}\left(\frac{s_{(i,1)}}{s_{(1)}} + \frac{s_{(i,1,1)}}{s_{(1,1)}}\right), \qquad \varepsilon_i^{(2)} = \frac{1}{s_{(i,i)}}\left(\frac{s_{(i,i,1)}}{s_{(1)}} + \frac{s_{(i+1,i,1)}}{s_{(1,1)}}\right), \tag{3.48}$$

$$n = 4; \quad \varepsilon_i^{(1)} = \frac{1}{s_{(i)}}\left(\frac{s_{(i,1)}}{s_{(1)}} + \frac{s_{(i,1,1)}}{s_{(1,1)}} + \frac{s_{(i,1,1,1)}}{s_{(1,1,1)}}\right), \tag{3.49}$$

$$\varepsilon_i^{(2)} = \frac{1}{s_{(i,i)}}\left(\frac{s_{(i,i,1)}}{s_{(1)}} + \frac{s_{(i+1,i,1)}}{s_{(1,1)}} + \frac{2s_{(i,i,1,1)}}{s_{(1,1)}} + \frac{s_{(i+1,i,1,1)}}{s_{(1,1,1)}}\right), \tag{3.50}$$

$$\varepsilon_i^{(3)} = \frac{1}{s_{(i,i,i)}}\left(\frac{s_{(i,i,i,1)}}{s_{(1)}} + \frac{s_{(i+1,i,i,1)}}{s_{(1,1)}} + \frac{s_{(i+1,i+1,i,1)}}{s_{(1,1,1)}}\right). \tag{3.51}$$

The result (3.47) reproduces [11, eq. (3.24)] with $a = z = x_1$.

Finally the densities $\rho_i^{(a)}$ and $\sigma_i^{(a)}$ are obtained from (3.17), (3.20) and (3.45) as

$$\rho_i^{(a)} = \sum_{r=1}^{n-1}(-h_{i-1}^{(a)}(r,1) + 2h_i^{(a)}(r,1) - h_{i+1}^{(a)}(r,1)) = (y_i^{(a)})^{-1}\sigma_i^{(a)}, \tag{3.52}$$

$$\sigma_i^{(a)} = 1 - \sum_{b,r=1}^{n-1} C_{ab}h_i^{(b)}(r,1). \tag{3.53}$$

## 4 Generalized hydrodynamics

### 4.1 Effective speed in homogenous system

According to Sec. 2.5, the configurations $S_i^{(a)}$ defined by (2.39) and (2.38) with $(a,i) \in \mathcal{I}$ provide the complete list of solitons. Under the time evolution $T_l^{(r)}$, a soliton $S_i^{(a)}$ possesses the bare speed $\delta_{ar}\min(i,l)$ and acquires the phase shift (2.40) in a collision with $S_j^{(b)}$. Let $v_i^{(a)} = v_i^{(a)}(r,l)$ denote the effective speed of $S_i^{(a)}$ under $T_l^{(r)}$. Then the speed equation in the sense of [41–43] takes the form

$$v_i^{(a)} = \delta_{ar}\min(i,l) + \sum_{(b,j)\in\mathcal{I}} C_{ab}\min(i,j)(v_i^{(a)} - v_j^{(b)})\rho_j^{(b)}. \tag{4.1}$$

Here $\rho_i^{(a)}$ is density of $S_i^{(a)}$ given either by $\rho_i^{(a)} = -\varepsilon_{i-1}^{(a)} + 2\varepsilon_i^{(a)} - \varepsilon_{i+1}^{(a)}$ or $\rho_i^{(a)} = y_i^{(a)}(1 - \sum_{b=1}^{n-1} C_{ab}\varepsilon_i^{(b)})$ due to (3.17) and (3.20). It is independent of $T_l^{(r)}$ and the coincidence of the two expressions is assured by (3.32). For $n = 2$ and $T_\infty^{(1)}$ dynamics, the equation (4.1) reduces to [23, eq.(11.7)].

From (3.17), the first term in the sum in (4.1) is equal to $v_i^{(a)}\sum_{b=1}^{n-1} C_{ab}\varepsilon_i^{(b)} = v_i^{(a)}(1 - \sigma_i^{(a)})$. Substitution of this into the speed equation (4.1) causes a cancellation of $v_i^{(a)}$, after which the result is expressed neatly in terms of the combination $\nu_i^{(a)}$ as follows:

$$\nu_i^{(a)} + \sum_{(b,j)\in\mathcal{I}} C_{ab}\min(i,j)y_j^{(b)}\nu_j^{(b)} = \delta_{ar}\min(i,l), \tag{4.2}$$

$$\nu_i^{(a)} = \nu_i^{(a)}(r,l) = \sigma_i^{(a)}v_i^{(a)}(r,l). \tag{4.3}$$

The equation (4.2) coincides with (3.34) under the identification $h_i^{(a)} = \sum_{j \geq 1} \min(i,j) y_j^{(a)} \nu_j^{(a)}$. Therefore it has a solution

$$\nu_i^{(a)}(r,l) = \delta_{ar} \min(i,l) - \sum_{b=1}^{n-1} C_{ab} h_i^{(b)}(r,l) \tag{4.4}$$

$$= (y_i^{(a)})^{-1}(-h_{i-1}^{(a)}(r,l) + 2h_i^{(a)}(r,l) - h_{i+1}^{(a)}(r,l)). \tag{4.5}$$

From the first formula (4.4) with $l = 1$ and (3.53) we have

$$\sigma_i^{(a)} = \sum_{k=1}^{n-1} \nu_i^{(a)}(k,1). \tag{4.6}$$

Combining (4.3) and (4.6) we obtain the effective speed

$$v_i^{(a)}(r,l) = \frac{\nu_i^{(a)}(r,l)}{\sum_{k=1}^{n-1} \nu_i^{(a)}(k,1)}, \tag{4.7}$$

in terms of $\nu_i^{(a)}(r,l)$ in (4.4)–(4.5) with $h_i^{(a)}(r,l)$ given by (3.36). The result leads to the relation

$$\sum_{r=1}^{n-1} v_i^{(a)}(r,1) = 1, \tag{4.8}$$

which is consistent with Remark 2.1.

Among the family of time evolutions $T_l^{(r)}$, typical ones are $T_\infty^{(1)}, \ldots, T_\infty^{(n-1)}$. The corresponding limit of the numerator $\lim_{l \to \infty} \nu_i^{(a)}(r,l)$ in (4.7) can be evaluated compactly as

$$\nu_i^{(a)}(r,\infty) = \delta_{ar} i - \sum_{b=1}^{n-1} C_{ab} h_i^{(b)}(r,\infty) = \delta_{ar} i - \sum_{b=1}^{n-1} C_{ab} h_\infty^{(r)}(b,i)$$

$$\stackrel{(3.35)}{=} \delta_{ar} i - \sum_{b=1}^{n-1} C_{ab} x_r \frac{\partial}{\partial x_r} \log(e^{-i\varpi_b} Q_i^{(b)}) = x_r \frac{\partial}{\partial x_r} \log\left(\frac{Q_i^{(a-1)} Q_i^{(a+1)}}{(Q_i^{(a)})^2}\right). \tag{4.9}$$

**Example 4.1.** Consider $n = 2$ case. The variables in (3.7)–(3.9) are related as $z_1 = z_2^{-1} = w_1 = x_1^{-\frac{1}{2}}$. The Schur function is given by $s_{(\nu_1,\nu_2)}(z_1, z_2) = (z_1^{d+1} - z_1^{-d-1})/(z_1 - z_1^{-1})$ with $d = \nu_1 - \nu_2$. From (4.4) and (3.37) we have

$$\nu_i^{(1)}(1,l) = i - \frac{2}{s_{(i)} s_{(l)}} \sum_{k=1}^{i} k s_{(i+l-k,k)} = i \frac{(1 + x_1^{1+i})(1 + x_1^{1+l})}{(1 - x_1^{1+i})(1 - x_1^{1+l})} - \frac{2x_1(1 - x_1^i)(1 + x_1^{1+l})}{(1 - x_1)(1 - x_1^{1+i})(1 - x_1^{1+l})}$$

$$\tag{4.10}$$

for $i \leq l$. In view of the symmetry $\nu_i^{(1)}(1,l) = \nu_l^{(1)}(1,i)$ (see (3.36)), general case is obtained by replacing $i$ with $j := \min(i,l)$ and $l$ with $m := \max(i,l)$ in this formula. In particular,

$$\nu_i^{(1)}(1,1) = \frac{(1 - x_1)(1 + x_1^{1+i})}{(1 + x_1)(1 - x_1^{1+i})} \tag{4.11}$$

holds for $i \geq 1$. Now the effective speed (4.7) is evaluated as

$$v_i^{(1)}(1,l) = \frac{\nu_i^{(1)}(1,l)}{\nu_i^{(1)}(1,1)} = \frac{1 + x_1^{1+m}}{1 - x_1^{1+m}} \left( j \frac{1+x_1}{1-x_1} - \frac{2x_1(1+x_1)(1-x^j)}{(1-x_1)^2(1+x_1^{1+j})} \right). \tag{4.12}$$

This reproduces [11, eq.(D.2)] with $a = z = x_1$. Let us also check (4.9) which says

$$\nu_i^{(1)}(1,\infty) = -2x_1 \frac{\partial}{\partial x_1} \log \frac{x_1^{\frac{i+1}{2}} - x_1^{-\frac{i+1}{2}}}{x_1^{\frac{1}{2}} - x_1^{-\frac{1}{2}}} = i \frac{1 + x_1^{i+1}}{1 - x_1^{i+1}} - \frac{2x_1(1-x_1^i)}{(1-x_1)(1-x_1^{i+1})}. \tag{4.13}$$

In the regime (3.14) under consideration, this indeed coincides $\lim_{l\to\infty} \nu_i^{(1)}(1,l)$ derived from (4.10).

**Example 4.2.** Consider $n = 3$ case. Explicit formulas for the effective speed $v_i^{(a)}(r,l)$ are messy for generic $z_1, z_2, z_3$ and $l$. Here we treat $v_i^{(a)}(r,\infty)$ for the one parameter specialization $(z_1, z_2, z_3) = (q^{-1}, 1, q)$ with $0 < q < 1$, which satisfies (3.14). This is the so called *principal specialization* reducing the Schur function to the $q$-dimension as $s_{(\lambda_1,\lambda_2,\lambda_3)} = \frac{q^{\lambda_3-\lambda_1}(1-q^{\lambda_1-\lambda_2+1})(1-q^{\lambda_1-\lambda_3+2})(1-q^{\lambda_2-\lambda_3+1})}{(1-q)^2(1-q^2)}$. To calculate the denominator of (4.7) we substitute

$$h_i^{(1)}(1,1) = \frac{s_{(i,1)}}{s_{(i)}s_{(1)}}, \quad h_i^{(2)}(1,1) = \frac{s_{(i-1,i-1)}}{s_{(i,i)}s_{(1)}}, \quad h_i^{(1)}(2,1) = \frac{s_{(i-1)}}{s_{(1,1)}s_{(i)}},$$

$$h_i^{(2)}(2,1) = \frac{s_{(i,i-1)}}{s_{(i,i)}s_{(1,1)}} \tag{4.14}$$

into (4.4), i.e., $\nu_i^{(1)}(1,1) = 1 - 2h_i^{(1)}(1,1) + h_i^{(2)}(1,1)$, $\nu_i^{(2)}(1,1) = h_i^{(1)}(1,1) - 2h_i^{(1)}(1,1)$, $\nu_i^{(1)}(2,1) = -2h_i^{(1)}(2,1) + h_i^{(2)}(2,1)$, $\nu_i^{(2)}(2,1) = 1 + h_i^{(1)}(2,1) - 2h_i^{(1)}(2,1)$ to find

$$\nu_i^{(1)}(1,1) = \nu_i^{(2)}(2,1) = \frac{(1-q)^2(1+q^{i+1})(1-q^{i+3})}{(1-q^3)(1-q^{i+1})(1-q^{i+2})}, \tag{4.15}$$

$$\nu_i^{(1)}(2,1) = \nu_i^{(2)}(1,1) = \frac{q(1-q)^2(1-q^i)(1+q^{i+2})}{(1-q^3)(1-q^{i+1})(1-q^{i+2})}. \tag{4.16}$$

Thus the denominator of (4.7) is factorized as

$$\nu_i^{(1)}(1,1) + \nu_i^{(1)}(2,1) = \nu_i^{(2)}(1,1) + \nu_i^{(2)}(2,1) = \frac{(1-q)(1-q^2)(1-q^{2i+3})}{(1-q^3)(1-q^{i+1})(1-q^{i+2})}. \tag{4.17}$$

Similar calculations of (4.4)–(4.5) lead to

$$\nu_i^{(1)}(1,\infty) = \nu_i^{(2)}(2,\infty) = i \frac{1 + q^{i+1} + q^{i+2}}{(1-q^{i+1})(1-q^{i+2})} - \frac{q(1-q^i)(2+3q+q^{i+3})}{(1-q^2)(1-q^{i+1})(1-q^{i+2})}, \tag{4.18}$$

$$\nu_i^{(1)}(2,\infty) = \nu_i^{(2)}(1,\infty) = -i \frac{q^{i+1}(1+q+q^{i+2})}{(1-q^{i+1})(1-q^{i+2})} + \frac{q(1-q^i)(1+3q^{i+2}+2q^{i+3})}{(1-q^2)(1-q^{i+1})(1-q^{i+2})}. \tag{4.19}$$

These are related by $\nu_i^{(1)}(2,\infty) = -\nu_i^{(1)}(1,\infty)\big|_{q\to q^{-1}}$. Finally the effective speed is given by

$$v_i^{(1)}(1,\infty) = v_i^{(2)}(2,\infty) = i \frac{(1-q^3)(1+q^{i+1}+q^{i+2})}{(1-q)(1-q^2)(1-q^{2i+3})} - \frac{q(1-q^3)(1-q^i)(2+3q+q^{i+3})}{(1-q)(1-q^2)^2(1-q^{2i+3})}, \tag{4.20}$$

$$v_i^{(1)}(2,\infty) = v_i^{(2)}(1,\infty) = -i \frac{q^{i+1}(1-q^3)(1+q+q^{i+2})}{(1-q)(1-q^2)(1-q^{2i+3})}$$
$$+ \frac{q(1-q^3)(1-q^i)(1+3q^{i+2}+2q^{i+3})}{(1-q)(1-q^2)^2(1-q^{2i+3})}. \tag{4.21}$$

Again they are related by $v_i^{(1)}(2,\infty) = -v_i^{(1)}(1,\infty)\big|_{q\to q^{-1}}$. The speeds (4.20) and (4.21) are positive for $0 < q < 1$.

## 4.2  Matrix form

It is convenient to formulate the results in the previous subsection in a matrix form whose indices range over $\mathcal{I}$ in (2.1). We introduce the matrices

$$I = (\delta_{ab}\delta_{ij})_{(a,i),(b,j)\in\mathcal{I}}, \qquad M = (C_{ab}\min(i,j))_{(a,i),(b,j)\in\mathcal{I}}, \tag{4.22}$$

$$\mathbf{y} = (\delta_{ab}\delta_{ij}y_i^{(a)})_{(a,i),(b,j)\in\mathcal{I}}, \qquad \mathbf{v} = \mathbf{v}(r,l) = (\delta_{ab}\delta_{ij}v_i^{(a)}(r,l))_{(a,i),(b,j)\in\mathcal{I}} \tag{4.23}$$

and the vectors

$$\mathbb{I} = (1)_{(a,i)\in\mathcal{I}}, \qquad \kappa = \kappa(r,l) = (\delta_{ar}\min(i,l))_{(a,i)\in\mathcal{I}}, \tag{4.24}$$

$$\rho = (\rho_i^{(a)})_{(a,i)\in\mathcal{I}}, \quad \sigma = (\sigma_i^{(a)})_{(a,i)\in\mathcal{I}}, \quad y = (y_i^{(a)})_{(a,i)\in\mathcal{I}}, \tag{4.25}$$

$$v = v(r,l) = \big(v_i^{(a)}(r,l)\big)_{(a,i)\in\mathcal{I}}, \quad \nu = \nu(r,l) = \sigma * v := \big(\sigma_i^{(a)}v_i^{(a)}(r,l)\big)_{(a,i)\in\mathcal{I}}, \tag{4.26}$$

where the component-wise (or Hadamard) product $*$ is commutative and will be used also for other vectors. For example we have $\rho = y * \sigma$.[8] In view of (4.1), the vector $\kappa(r,l)$ is the bare speed of solitons under the time evolution $T_l^{(r)}$. Note that $I$ and $\mathbb{I}$ are different.

Now the equations (3.17) and (4.2) regarded as the ones for $\sigma$ and $\nu$ are presented as follows:

$$\sigma + M\rho = (I + M\mathbf{y})\sigma = \mathbb{I}, \tag{4.27}$$

$$\sigma * v(r,l) + M(\rho * v(r,l)) = (I + M\mathbf{y})\nu(r,l) = \kappa(r,l). \tag{4.28}$$

In general we define the "dressing" $\eta^{\mathrm{dr}}$ of a vector $\eta$ by

$$\eta^{\mathrm{dr}} = \eta - M\mathbf{y}\eta^{\mathrm{dr}}, \tag{4.29}$$

namely, $\eta^{\mathrm{dr}} = (I + M\mathbf{y})^{-1}\eta$. Then (4.27) and (4.28) are rephrased as

$$\sigma = \mathbb{I}^{\mathrm{dr}}, \qquad \nu(r,l) = \mathbb{I}^{\mathrm{dr}} * v(r,l) = \kappa(r,l)^{\mathrm{dr}}. \tag{4.30}$$

In particular the effective speed is given by

$$v(r,l) = \frac{\kappa(r,l)^{\mathrm{dr}}}{\mathbb{I}^{\mathrm{dr}}} = \frac{(I + M\mathbf{y})^{-1}\kappa(r,l)}{(I + M\mathbf{y})^{-1}\mathbb{I}}, \tag{4.31}$$

where the division is component-wise. From $\rho = y * \sigma$ we also have

$$\rho = (\mathbf{y}^{-1} + M)^{-1}\mathbb{I}, \qquad \rho * v(r,l) = (\mathbf{y}^{-1} + M)^{-1}\kappa(r,l). \tag{4.32}$$

**Remark 4.3.** Regard the effective speed (4.31) as a function of $\{y_i^{(a)} \mid (a,i) \in \mathcal{I}\}$. Then from Cramer's formula and the fact that $\mathbf{y}$ (4.23) is diagonal, it follows that $v_s^{(p)}(r,l)$ does not depend on the variable $y_s^{(p)}$ for any $(p,s) \in \mathcal{I}$. This property will be used to ensure (5.5).

**Remark 4.4.** In the dilute limit explained in Remark 3.2, one has $y_i^{(a)} = 0$ and $\rho_i^{(a)} = 0$ from (3.24) and (3.20) for all $(a,i) \in \mathcal{I}$. Therefore the dressing (4.29) trivializes into $\eta^{\mathrm{dr}} = \eta$. This brings the effective speed back to the bare speed, i.e., $v(r,l) = \kappa(r,l)$ in (4.31).

---

[8]The $y$ in (4.25) is a vector which should be distinguished from the diagonal matrix $\mathbf{y}$ in (4.23).

### 4.3 Stationary currents

Let us define the (density of) color $a$ stationary current under the time evolution $T_l^{(r)}$ as

$$J^{(a)}(r,l) = \sum_{i \geq 1} i J_i^{(a)}(r,l), \qquad J_i^{(a)}(r,l) = \rho_i^{(a)} v_i^{(a)}(r,l). \tag{4.33}$$

In view of Remark 2.8, this is a component of the *weight* current $-\sum_{a=1}^{n-1} J^{(a)}(r,l)\alpha_a$ transported by solitons. It is expressed as a quadratic form of the bare velocities as

$$J^{(a)}(r,l) = \kappa(a,\infty)^{\mathrm{T}} \rho * v(r,l) = \kappa(a,\infty)^{\mathrm{T}} (\mathbf{y}^{-1} + M)^{-1} \kappa(r,l) \tag{4.34}$$

due to (4.32), where $\eta^{\mathrm{T}}$ signifies a row vector obtained by the transpose of a column vector $\eta$.

As an additional remark, one can also express $J^{(a)}(r,l)$ in a form that generalizes [11, eq.(4.4)] naturally. Let $m_i^{(a)}$ be the number of soliton $S_i^{(a)}$ (2.39) in the system. We assume $m_i^{(a)} = 0$ for $i$ sufficiently large. Using (3.3) one can rewrite the speed equation (4.2) as

$$p_i^{(a)} v_i^{(a)} + \sum_{(b,j) \in \mathcal{I}} C_{ab} \min(i,j) m_j^{(b)} v_j^{(b)} = L\delta_{ar} \min(i,l). \tag{4.35}$$

Let $\hat{\mathcal{I}} = \{(a,i,\alpha) \mid (a,i) \in \mathcal{I}, \alpha \in [1, m_i^{(a)}]\}$ be an extension of the index set $\mathcal{I}$ in (2.1). The extra component $\alpha$ may be viewed as labeling individual solitons $S_i^{(a)}$ of color $a$ and length $i$. The cardinality of $\hat{\mathcal{I}}$ is $\sum_{(a,i) \in \mathcal{I}} m_i^{(a)}$ which is finite by the assumption. With a vector $\eta = (\eta_i^{(a)})_{(a,i) \in \mathcal{I}}$ we associate a similar extension $\hat{\eta} = (\eta_i^{(a)})_{(a,i,\alpha) \in \hat{\mathcal{I}}}$. It repeats the same element $\eta_i^{(a)}$ for $m_i^{(a)}$ times which is the size of the $(a,i)$ block. Introduce the matrix[9]

$$B = \left(p_i^{(a)} \delta_{ab} \delta_{ij} \delta_{\alpha\beta} + C_{ab} \min(i,j)\right)_{(a,i,\alpha),(b,j,\beta) \in \hat{\mathcal{I}}}. \tag{4.36}$$

These definitions enable us to present the speed equation (4.35) as

$$B\hat{v}(r,l) = L\hat{\kappa}(r,l). \tag{4.37}$$

Similarly the current (4.33) is expressed as a quadratic form:

$$J^{(a)}(r,l) = \frac{1}{L} \sum_{i \geq 1} i m_i^{(a)} v_i^{(a)}(r,l) = \frac{1}{L} \hat{\kappa}(a,\infty)^{\mathrm{T}} \hat{v}(r,l) = \hat{\kappa}(a,\infty)^{\mathrm{T}} B^{-1} \hat{\kappa}(r,l). \tag{4.38}$$

This formula is a natural generalization of the $n = 2$ case in [11, eq.(4.4)], where a further connection to the period matrix of the tropical Riemann theta function was explored. At present, time averaged current of the $n$-color cBBS on a circle is yet to be obtained. However we expect the result should coincide with (4.38) supported by the $n = 2$ case established in [11].

### 4.4 Dynamics in an inhomogeneous system

Let us study an inhomogeneous system, with the hypothesis that it can be locally described by using the above formalism, but with the densities that have acquired a dependence on space ($x = j/L$, $j$ = lattice site number) and time ($t = j/L$, $j$ = time step). The main assumption is

---

[9]$B$ appeared in [44, eq.(4.6)] for a non-complete multi-color BBS with a periodic boundary condition with the vacancy $p_i^{(a)}$ different from (3.3) reflecting the "non-completeness". Of course this $B$ should not be confused with $B$ in (2.21).

that the *soliton* current $J_i^{(a)}(r,l)$ in (4.33) associated with $S_i^{(a)}$ obeys the continuity equation $\partial_t \rho_i^{(a)} + \partial_x J_i^{(a)}(r,l) = 0$ for any time evolution $T_l^{(r)}$.[10] In the matrix notation it reads

$$\partial_t \rho + \partial_x(\rho * v(r,l)) = 0. \tag{4.39}$$

Substituting (4.32) and using $\partial_\alpha(\mathbf{y}^{-1}+M)^{-1} = \beta\partial_\alpha\mathbf{y}(\mathbf{y}^{-1}+M)^{-1}$ with $\beta = (\mathbf{y}^{-1}+M)^{-1}\mathbf{y}^{-2}$ for $\alpha = t,x$, we get $\partial_t\mathbf{y}(\mathbf{y}^{-1}+M)^{-1}\mathbb{I} + \partial_x\mathbf{y}(\mathbf{y}^{-1}+M)^{-1}\kappa(r,l) = 0$. By utilizing (4.32) again and noting that $\mathbf{y}$ and $\mathbf{v}$ in (4.23) are diagonal, the result becomes

$$\partial_t y + \mathbf{v}(r,l)\partial_x y = 0. \tag{4.40}$$

This is a collection of separated equations for each $(a,i) \in \mathcal{I}$ meaning that $y_i^{(a)}$'s are the *normal modes* of the generalized hydrodynamics [10].

**Remark 4.5.** Replacing $\rho = (\rho_i^{(a)})_{(a,i)\in\mathcal{I}}$ in (4.39) by $(f_i^{(a)}(y_i^{(a)})\rho_i^{(a)})_{(a,i)\in\mathcal{I}}$ with any function $f_i^{(a)}$ leads to the same equation (4.40). In particular, taking $f_i^{(a)}(y_i^{(a)}) = y_i^{(a)}$ leads to the conservation of the hole current $\partial_t\sigma + \partial_x(\sigma * v(r,l)) = 0$.

## 5 Domain wall initial condition

Let us embark on the study of transient behaviors of the randomized cBBS started from the domain wall initial condition. This will give the possibility to check some of the previous results obtained by using TBA and GHD with direct numerical simulations. Recall that an i.i.d. distribution is specified by the fugacity $(z_1, \ldots, z_n)$ as explained in (3.4)–(3.9). We prepare the system initially in the i.i.d. distribution with fugacity $(z_{1L}, \ldots, z_{nL})$ in the left domain $x < 0$ and a different fugacity $(z_{1R}, \ldots, z_{nR})$ in the right domain $x > 0$. Then at time $t = 0$, we begin running a dynamics $T_l^{(r)}$ for some $(r,l) \in \mathcal{I}$, and observe the long time non-equilibrium behavior in the mixture of the soliton gas. This kind of setting is called partitioning protocol, which has been studied in the recent literature. See [10, 18, 19] and the references therein. We shall present our GHD analysis first in a ballistic picture (Sec. 5.1) and then its diffusive correction (Sec. 5.2) followed by numerical verifications using simulations (Sec. 5.3).

### 5.1 Ballistic picture

The first approximation we employ is the *ballistic* scaling which implies that the normal mode $y_i^{(a)}(x,t)$ (see Sec. 4.4) depends on $x,t$ only through the *ray variable* $\zeta = x/t$ as $y_i^{(a)}(\zeta)$. Then (4.40) is reduced to

$$\big(\zeta - v_i^{(a)}(r,l)\big)\partial_\zeta y_i^{(a)}(\zeta) = 0 \qquad ((a,i) \in \mathcal{I}). \tag{5.1}$$

Since the set of velocities $v_i^{(a)}(r,l)$ is discrete in our setting, it follows that $y_i^{(a)}(\zeta)$, hence the local state of the system itself, is piecewise constant and uniform when observed via the variable $\zeta$.[11][12] Thus the plot of any local quantity, typically density of solitons $S_i^{(a)}$, against $\zeta$

---

[10]The associated time variable could also be denoted by $t_l^{(r)}$, which is however avoided for notational simplicity.

[11]One can show that $y_i^{(a)}(\zeta)$ can change discontinuously across $\zeta = v_i^{(a)}(r,l) \pm 0$ without violating the current conservation. The proof is formally identical with [11, Sec. 5.3].

[12]A short comment is here in order about the word *state*. So far we have been using state to represent a single microscopic ball configuration. However, in the present GHD context, the (local) state of the system must be understood in a thermodynamic or probabilistic sense, as a measure over microscopic configurations.

collapses onto a single (and "static") curve as $t \to \infty$, which is a collection of *plateaux* having sharp edges as follows:

local quantity

$$\mathcal{P}$$
$$\mathcal{P}'$$
$$\zeta^*$$
$$\zeta = x/t$$

(5.2)

Obviously heights in such a plot depend on the local quantity to consider but the locations of the plateau edges are common for all of them.

In general the normal mode $y_i^{(a)}(\zeta)$ can take two values $y_{iL}^{(a)}$ or $y_{iR}^{(a)}$ which are determined from the fugacity $(z_{1L}, \ldots, z_{nL})$ or $(z_{1R}, \ldots, z_{nR})$ by (3.24) and (3.10). In the mixed inhomogeneous system under consideration, we have the boundary condition $y_i^{(a)}(-\infty) = y_{iL}^{(a)}$ and $y_i^{(a)}(\infty) = y_{iR}^{(a)}$. For the neighboring plateaux $\mathcal{P}$ and $\mathcal{P}'$ as in (5.2), the relation (5.1) indicates that there is a subset $\mathcal{J} \subset \mathcal{I}$ such that the following hold[13]:

$$y_i^{(a)}(\zeta^* - 0) = y_i^{(a)}(\zeta^* + 0) \;\; (= y_{iL}^{(a)} \text{ or } y_{iR}^{(a)}) \;\; \text{for } (a,i) \in \overline{\mathcal{J}}, \tag{5.3}$$

$$y_i^{(a)}(\zeta) = \begin{cases} y_{iL}^{(a)} & \zeta < \zeta^* \\ y_{iR}^{(a)} & \zeta > \zeta^* \end{cases} \;\; \text{for } (a,i) \in \mathcal{J}, \tag{5.4}$$

$$v_i^{(a)}(r,l)_{\mathcal{P}} = v_i^{(a)}(r,l)_{\mathcal{P}'} \;\; \text{for } (a,i) \in \mathcal{J}. \tag{5.5}$$

Here $v_i^{(a)}(r,l)_{\mathcal{P}}, v_i^{(a)}(r,l)_{\mathcal{P}'}$, denote the effective speed of $S_i^{(a)}$ in $\mathcal{P}, \mathcal{P}'$ which can be computed from the respective normal mode $\mathbf{y}$ by (4.31). The equality (5.5) can be derived from (5.3) by using Remark 4.3. Put plainly, $\mathcal{J}$ is the set such that (5.1) holds as $0 \times \infty = 0$ if $(a,i) \in \mathcal{J}$ and as (finite nonzero) $\times 0 = 0$ if $(a,i) \in \overline{\mathcal{J}}$. From the consistency with this picture, we expect that the value (5.5) coincides with $\zeta^*$ for all $(a,i) \in \mathcal{J}$.

As we will see, $\mathcal{J}$ is not necessarily a single element set. In fact the point $\zeta^* = 0$ in the figure 2 corresponds to the infinite set $\mathcal{J} = \{(2,i) \mid i \in \mathbb{Z}_{\geq 1}\}$.

Let us label a plateau $\mathcal{P}$ with a subset $\mathcal{P} \subseteq \mathcal{I}$ (denoted by the same symbol) such that

$$y_i^{(a)}(\zeta) = \begin{cases} y_{iL}^{(a)} & (a,i) \in \overline{\mathcal{P}}, \\ y_{iR}^{(a)} & (a,i) \in \mathcal{P}. \end{cases} \tag{5.6}$$

Then the transition rule (5.3)–(5.4) from $\mathcal{P}$ to $\mathcal{P}'$ across the edge $\zeta = \zeta^*$ is stated as

$$\mathcal{P} \sqcup \mathcal{J} = \mathcal{P}'. \tag{5.7}$$

The boundary condition means that $\mathcal{P} \to \emptyset$ as $\zeta \to -\infty$ and $\mathcal{P} \to \mathcal{I}$ as $\zeta \to \infty$. Thus a plateau configuration can be viewed as specifying a monotonic growth pattern of the label $\mathcal{P}$. One starts from $\mathcal{P} = \emptyset$ in the far left and proceeds to the right inductively to eventually get $\mathcal{P} = \mathcal{I}$ in the far right. In the inductive step, we assume that a plateau $\mathcal{P}$ and the associated data $\{v_i^{(a)}(r,l)_{\mathcal{P}} \mid (a,i) \in \mathcal{I}\}$ are at hand. Then its right edge $\zeta^*$ as in (5.2) and the set $\mathcal{J}$ which specifies the next plateau $\mathcal{P}'$ by (5.7) are determined as

$$\zeta^* = \min_{(a,i) \in \overline{\mathcal{P}}} \{v_i^{(a)}(r,l)_{\mathcal{P}}\}, \tag{5.8}$$

$$\mathcal{J} = \{(a,i) \in \overline{\mathcal{P}} \mid v_i^{(a)}(r,l)_{\mathcal{P}} = \zeta^*\}. \tag{5.9}$$

---

[13]For any subset $\mathcal{K} \subseteq \mathcal{I}$, we let $\overline{\mathcal{K}}$ denote the complement $\mathcal{I} \setminus \mathcal{K}$.

This inductive procedure to trace the plateau profile is convenient to arrange GHD based numerical calculations.

It is worth mentioning that the two alternatives of the normal mode variable $y_i^{(a)} = y_{iL}^{(a)}$ or $y_{iR}^{(a)}$ on a plateau can be interpreted as whether the solitons $S_i^{(a)}$ there originate in the domain $x < 0$ or $x > 0$. This is indicated in (3.20) which equates $y_i^{(a)}$ to the density $\rho_i^{(a)}$ of $S_i^{(a)}$ up to the factor $\sigma_i^{(a)}$ reflecting the interaction effect. Thus we may interpret physically a plateau label $\mathcal{P}$ as the *soliton content* meaning that it is the region of $\zeta$ where

solitons $\{S_i^{(a)} \mid (a, i) \in \mathcal{P}\}$ originating in the right domain have not gone away yet, (5.10)

solitons $\{S_i^{(a)} \mid (a, i) \in \overline{\mathcal{P}}\}$ originating in the left domain have already reached. (5.11)

In what follows we will signify the solitons in (5.10), (5.11) as $S_{iR}^{(a)}$, $S_{iL}^{(a)}$ and refer to $\mathcal{P}$ also as soliton content. If the initial domain is *empty*[14] for $x < 0$ or $x > 0$, then (5.11) or (5.10) becomes void, respectively.

So far our description of the plateaux is quite general. Now we propose a simplifying conjecture based on numerical experiments.

**Conjecture.** For any fugacity $(z_{1L}, \ldots, z_{nL})$ and $(z_{1R}, \ldots, z_{nR})$ satisfying (3.14), the following properties are valid:

1. Actually realized soliton contents $\mathcal{P} \subseteq \mathcal{I}$ are limited to the following form for some $\mathbf{d}$:
$$\mathcal{P}(\mathbf{d}) = \{(a, i) \mid 1 \le i \le d_a, \ a \in [1, n-1]\}, \qquad \mathbf{d} = (d_1, \ldots, d_{n-1}) \in (\mathbb{Z}_{\ge 0})^{n-1}. \tag{5.12}$$

2. The effective speeds are nonnegative for any time evolution and on any plateau, i.e.,
$$v_i^{(a)}(r, l)_{\mathcal{P}(\mathbf{d})} \ge 0 \qquad (\forall (a, i), (r, l) \in \mathcal{I}). \tag{5.13}$$

Nonnegativity of the velocities is not obvious since the solitons $S_i^{(a)}$ with $a \neq r$ have the vanishing bare speed under $T_l^{(r)}$ (see Sec. 2.5 (I)), and they may receive a "recoil effect" from the march of color $r$ solitons. For instance there is a minus sign in (4.21) although these velocities are indeed positive for $0 < q < 1$. Due to (5.13) all the solitons are right movers, therefore the domain $\zeta < 0$ is not influenced by the domain $\zeta > 0$. This feature allows us to replace one of the boundary condition $y_i^{(a)}(-\infty) = y_{iL}^{(a)}$ by $y_i^{(a)}(-0) = y_{iL}^{(a)}$. Consequently, the plateaux can be numbered as $0, 1, 2, \ldots$ from the left to the right, where the *leftmost* 0 covers the entire $\zeta < 0$ region at least.

Thanks to (5.12) a plateau can be labeled by a vector $\mathbf{d} \in (\mathbb{Z}_{\ge 0})^{n-1}$, which corresponds to the soliton content

$$
\begin{aligned}
& S_{1R}^{(1)}, \ \ldots, \ S_{d_1 R}^{(1)}, \ S_{d_1+1\,L}^{(1)}, \ S_{d_1+2\,L}^{(1)}, \ldots, \\
& S_{1R}^{(2)}, \ \ldots, \ S_{d_2 R}^{(2)}, \ S_{d_2+1\,L}^{(2)}, \ S_{d_2+2\,L}^{(2)}, \ldots, \\
& \qquad\qquad\qquad \cdots \\
& S_{1R}^{(n-1)}, \ldots, S_{d_{n-1} R}^{(n-1)}, \ S_{d_{n-1}+1\,L}^{(n-1)}, \ S_{d_{n-1}+2\,L}^{(n-1)}, \cdots
\end{aligned}
\tag{5.14}
$$

For each color $a$, it is like a Dirac sea of level $i = d_a$ without a hole. Obviously $\mathcal{P}(\mathbf{d}) \subseteq \mathcal{P}(\mathbf{d}')$ holds if and only if $\mathbf{d} \le \mathbf{d}'$, i.e. $\mathbf{d}' - \mathbf{d} \in (\mathbb{Z}_{\ge 0})^{n-1}$. Thus one can describe a plateau configuration just by an increasing sequence as $(0, \ldots, 0) = \mathbf{d}_0 \le \mathbf{d}_1 \le \mathbf{d}_2 \le \cdots \in (\mathbb{Z}_{\ge 0})^{n-1}$ which should tend to $(\infty, \ldots, \infty)$.

---

[14]Only vac (2.21) without a soliton, which corresponds to the dilute limit as mentioned in Remark 3.2 and Remark 4.4.

**Example 5.1.** Consider the cBBS with $n = 2$, which is nothing but the ordinary single color BBS studied in [11]. We have the time evolutions $T_l^{(r=1)}$ with $l = 1, 2, \ldots$, and the plateaux are associated with $0 = d_0 \leq d_1 \leq d_2 \leq \cdots$ specifying their solitons contents. A typical setting is to let the initial domain in $x > 0$ or $x < 0$ be empty. Examples of the ball density plot in such cases are available in [11, Fig. 4]. In either case, there are exactly $l + 1$ plateaux $\mathcal{P}(d_0), \mathcal{P}(d_1), \ldots, \mathcal{P}(d_l)$ where $d_j$ is given by

$$d_j = j \ \ (0 \leq j < l), \quad d_l = \infty. \tag{5.15}$$

Analytic formulas have also been obtained for many quantities in [11, Sec. 5.6-5.7], which are compatible with the above conjecture.

## 5.2 Diffusive correction

As we shall demonstrate in the next subsection, the actual plateau edges exhibit broadening for finite $t$ whose main cause is the diffusive effect [10, 11, 45, 46]. To describe it quantitatively, consider the adjacent plateaux $\mathcal{P}$ and $\mathcal{P}'$ as in (5.2). We assume that the soliton contents of $\mathcal{P}$ and $\mathcal{P}'$ differ only by one kind of solitons $S_s^{(p)}$ for some $(p, s) \in \mathcal{I}$, namely $\mathcal{J} = \{(p, s)\}$ and $\mathcal{P} \sqcup \{(p, s)\} = \mathcal{P}'$ in (5.7). In terms of (5.12) we have $\mathcal{P} = \mathcal{P}(\mathbf{d}), \mathcal{P}' = \mathcal{P}(\mathbf{d}')$ where $\mathbf{d} = (d_1, \ldots, d_{n-1})$ and $\mathbf{d}' = (d_1', \ldots, d_{n-1}')$ are only different in the $p$ th component as $d_p = s - 1, d_p' = s$.

Then the same argument as [11] (see also [46]) can be applied to obtain the envelope of an averaged local quantity $Q(x, t)$ including the diffusive correction. Here we just mention a few key formulas used in the derivation:

$$\frac{\partial^2 F[\rho]}{\partial \epsilon_\alpha \partial \epsilon_\beta} = \delta_{\alpha\beta} \frac{\rho_\alpha \sigma_\alpha}{\rho_\alpha + \sigma_\alpha}, \qquad \epsilon_\alpha := -\log y_\alpha, \tag{5.16}$$

$$\frac{\partial G_{\alpha\beta}}{\partial \epsilon_\gamma} = \delta_{\alpha\gamma} G_{\alpha\beta} - G_{\alpha\gamma} G_{\gamma\beta}, \qquad G := (1 + M\mathbf{y})^{-1}, \tag{5.17}$$

$$\frac{\partial v_\alpha}{\partial \epsilon_\beta} = M_{\alpha\beta}^{\mathrm{dr}} \frac{\rho_\beta}{\sigma_\alpha} (v_\beta - v_\alpha), \qquad M^{\mathrm{dr}} := (1 + M\mathbf{y})^{-1} M. \tag{5.18}$$

See (3.18), (3.20), (4.22), (4.23) and (4.29) for the relevant definitions. We have written $y_\alpha, \epsilon_\beta$ etc to mean $y_i^{(a)}, \epsilon_j^{(b)}$ for $\alpha = (a, i), \beta = (b, j) \in \mathcal{I}$ for simplicity.[15] The final result is expressed in terms of the complementary error function $\mathrm{erfc}(u) = \frac{2}{\sqrt{\pi}} \int_u^\infty \mathrm{e}^{-u^2} du$ as

$$\langle Q(x, t) \rangle = \frac{1}{2} \big( Q(\mathcal{P}) - Q(\mathcal{P}') \big) \mathrm{erfc} \left( \sqrt{\frac{t}{2}} \frac{x/t - \zeta}{\Sigma_{\mathcal{P}, \mathcal{P}'}} \right) + Q(\mathcal{P}'), \tag{5.19}$$

in the vicinity of $x/t = \zeta$. The symbols $Q(\mathcal{P})$ and $Q(\mathcal{P}')$ denote the (averaged) values of $Q$ on the plateau $\mathcal{P}$ and $\mathcal{P}'$, respectively. The most essential quantity in (5.19) is the constant $\Sigma_{\mathcal{P}, \mathcal{P}'} (> 0)$ which controls the width of the edge broadening. It is given by

$$(\Sigma_{\mathcal{P}, \mathcal{P}'})^2 = \frac{1}{(\sigma_s^{(p)})^2} \sum_{(b, j) \in \mathcal{P}} \big( M_{(p, s), (b, j)}^{\mathrm{dr}} \big)^2 |v_s^{(p)} - v_j^{(b)}| \sigma_j^{(b)} y_j^{(b)} (1 + y_j^{(b)}), \tag{5.20}$$

where $\sigma_i^{(a)}, v_i^{(a)}, y_i^{(a)}$ hence $M^{\mathrm{dr}}$ are those on the plateau $\mathcal{P}$.

The error function form (5.19) and the above quantity $\Sigma_{\mathcal{P}, \mathcal{P}'}$ will be compared quantitatively with the step broadening obtained in the simulations, at the end of the next section.

---

[15]The $\epsilon_\alpha$ in (5.16) should not be confused with the one in (3.28).

## 5.3 Simulations

In this section we present some numerical simulations of the domain wall problem in the $n = 3$ model, and we compare the results with the GHD predictions. The method we use to simulate the BBS dynamics is similar to that of Ref. [11]. We consider a large but finite system with $L$ sites, each having a $B^{1,1}$ tableau and a $B^{2,1}$ semistandard tableau. The system has open boundary conditions and is initially divided into three regions: the 'left' correspond to $x \in [-L/3, -1]$, the 'right' corresponds to $x \in [0, L/3 - 1]$, and the 'buffer' to $x \in [L/3, 2L/3 - 1]$.

As already mentioned at the beginning of Sec. 5, at time $t = 0$ the left part is initialized in some random i.i.d. state characterized by fugacities $z_{1,L}, z_{2,L}$ and $z_{3,L}$. The right part is initialized in some random i.i.d. state characterized by fugacities $z_{1,R}, z_{2,R}$ and $z_{3,R}$. As for the buffer, it is initially in the vacuum state. The role of this buffer region is to ensure that no soliton emanating from the left or the right has had enough time to reach the last site ($x = 2L/3 - 1$) at the end of the simulation at $t = t_{\max}$ (we typically take $t_{\max} = 1000$, but some simulations have been carried out up to $t = 3000$). It is also important to check that the fastest solitons starting from the leftmost sites of the system have not had enough time to reach the left/right boundary ($x = 0$) at the end of the simulation. These conditions ensure that the finite size does not influence the behavior of the system in the region $x \in [0, L/3 - 1]$ at least up to time $t_{\max}$. To match these conditions we need to scale the system size $L$ as $t_{\max}$ times the largest effective velocity.

The time evolution is computed by successive applications of a transfer matrix $T_l^{(a)}$ with some large $l$ (unless specified otherwise we used $l = 100$), and $a = 1$ or $a = 2$. We also report some results obtained using the product $T_l^{(1)} T_l^{(2)}$. The application of such a transfer matrix on a given configuration is done using the combinatorial $R$ defined in App. B. At a few specific times the local densities $\rho_k^{(a)}(x, t)$ of solitons of size $k$ and color $a$ (using the local versions of the energies defined in App. B) as well as the mean box occupancies ($\varrho_{i=1,2,3}$) are recorded. In practice these densities are estimated by some average over a large number $N_{\text{samples}}$ of initial conditions. We typically use $N_{\text{samples}} \geq 300000$.

The Figs. 1, 2, 3 and 4 display the six soliton densities $\rho_{k=1,2,3}^{(a=1,2)}(x, t)$ at time $t = 600$ and $t = 1000$ as a function of $\zeta = x/t$. In all cases the curves associated with different times are almost on top of each other, which indicates the ballistic nature of the transport. The above figures correspond to different initial states: the right part is empty in Fig. 1 and 4, the left part is empty in Fig. 2, and both sides are in a nontrivial state in Fig. 3. In all cases we observe that the soliton densities are in good agreement with the GHD prediction (Sec. 5.1). It can also be observed that in the vicinity of each transition between two consecutive plateaux the numerical curves do not form steep steps and have a finite slope. In addition the results are not exactly on top of each other, and the curves appear to become steeper when increasing $t$. This is a manifestation of the diffusive broadening discussed in Sec. 5.2. We will come back to this point below when discussing Fig. 10.

Fig. 1 offers an illustration of the evolution of the soliton content from one plateau to the next, as discussed in Sec. 5.1. In this particular example the right half is initially empty, and all the $y_{iR}^{(a)}$ therefore vanish. So, if a soliton type $(i, a)$ belongs to the set $\mathcal{P}$ of a given plateau, then this species is absent in this plateau, its density $\rho_i^{(a)}$ vanishes. In Fig. 1 the first step is located at $\zeta(1) \simeq 0.23815$, and beyond this value of $\zeta$ the solitons $S_1^{(2)}$ are absent. At this transition we have $\mathcal{J} = \{(a = 2, i = 1)\}$. The second transition ($\zeta(2) \simeq 0.30285$) is the place where the $S_1^{(1)}$ solitons disappear, hence $\mathcal{J} = \{(a = 1, i = 1)\}$ for this step. The third step is located at $\zeta(3) \simeq 0.53924$. Beyond this value of $\zeta$ the $S_2^{(2)}$ soliton are absent and this step is characterized by $\mathcal{J} = \{(a = 2, i = 2)\}$. At the fourth step, $\zeta(4) \simeq 0.80940$, the $S_3^{(2)}$ soliton

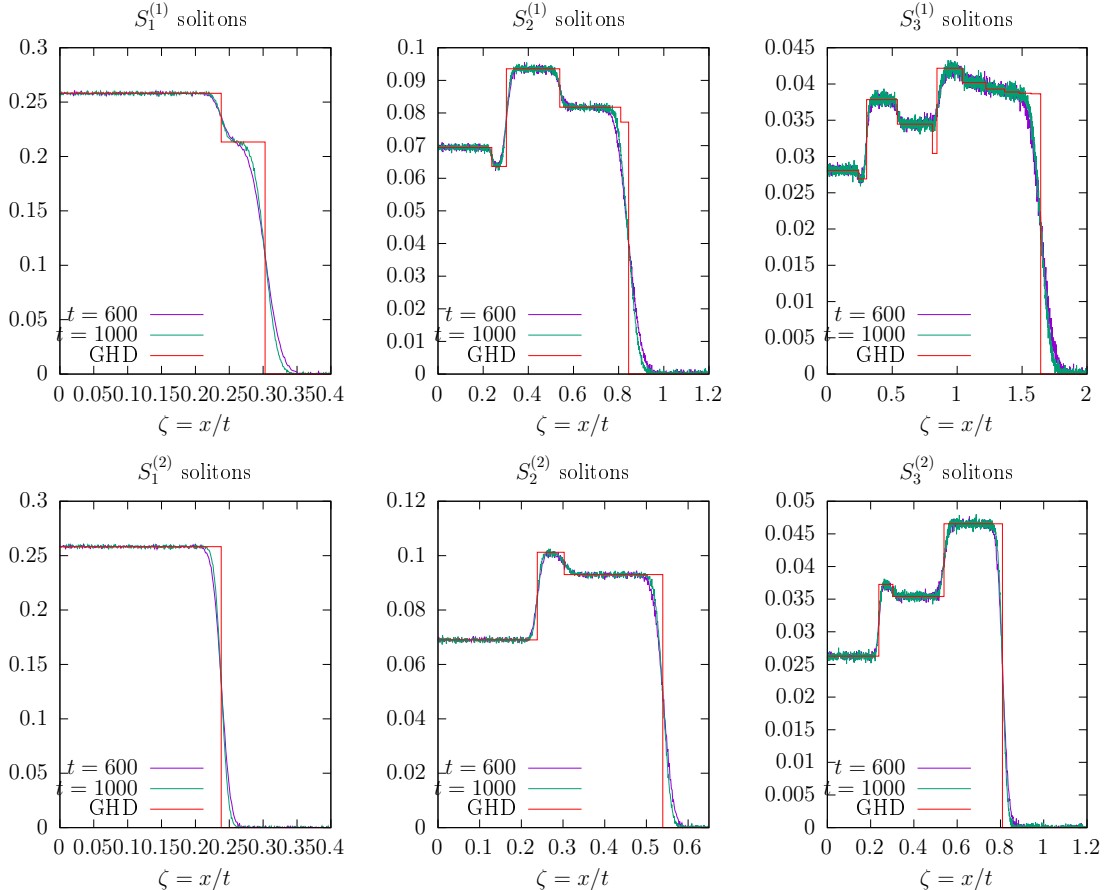

Figure 1: Numerical results for several $S_k^{(a)}$-soliton densities (with sizes $k = 1, 2, 3$ and 'color' $a = 1, 2$) as a function of $\zeta = x/t$ for $t = 600$ and $t = 1000$ and for the $T_{l=100}^{(1)}$ dynamics. The red line indicates the GHD prediction for $t = \infty$. The initial state is defined by the following fugacities in the left half: $z_{1,L} = 1$ $z_{2,L} = 0.7$ and $z_{3,L} = 0.3$. The right half is initially in the vacuum state. Parameters of the simulations: system size $L = 300000$ and number of random initial conditions $N_{\text{samples}} = 400000$.

density drops to zero and $\mathcal{J} = \{(a = 2, i = 3)\}$. Next we have $\zeta(5) \simeq 0.84452$, where the $S_2^{(1)}$ soliton disappear ($\mathcal{J} = \{(a = 1, i = 2)\}$). So far all this information can be read out from Fig. 1, by looking at the values of $\zeta$ where one soliton density drops to zero. Using the notations of (5.12) this translates into: $\mathbf{d}_0 = (0, 0)$, $\mathbf{d}_1 = (0, 1)$, $\mathbf{d}_2 = (1, 1)$, $\mathbf{d}_3 = (1, 2)$, $\mathbf{d}_4 = (1, 3)$, $\mathbf{d}_5 = (2, 3)$.

To go further in the description of the next plateaux one needs to look at the densities of solitons with larger ($> 3$) amplitude, and at larger values of $\zeta$ (data not shown). Such an analysis shows that for this particular initial condition the soliton content follows a somewhat irregular pattern up to the 12$^{\text{th}}$ plateau: $\mathbf{d}_6 = (2, 4)$, $\mathbf{d}_7 = (2, 5)$, $\mathbf{d}_8 = (2, 6)$, $\mathbf{d}_9 = (2, 7)$, $\mathbf{d}_{10} = (2, 8)$, $\mathbf{d}_{11} = (2, 9)$, $\mathbf{d}_{12} = (3, 9)$. And then the sequence becomes regular: $\mathbf{d}_{n \geq 12} = (3, n - 3)$. After this first infinite series of plateaux a second series begins. The plateaux of the second series are free of color-2 solitons and they are characterized by: $\mathbf{d}_{\infty+1} = (4, \infty)$, $\mathbf{d}_{\infty+2} = (5, \infty)$, $\cdots$, $\mathbf{d}_{\infty+n} = (n + 3, \infty)$.

The evolution of the soliton content is qualitatively different in the example of Fig. 2, where the initial state is the vacuum in the left half. There, the leftmost plateau $\mathcal{P}_0$ which starts at

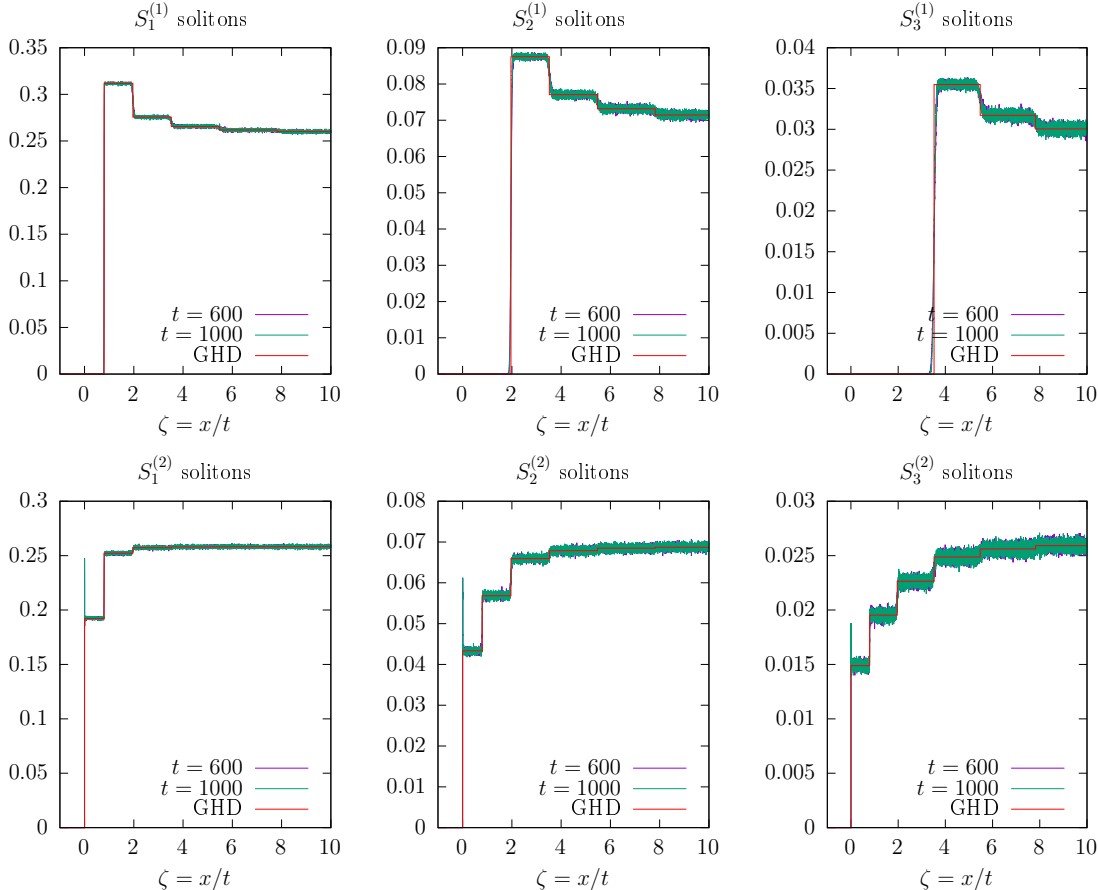

Figure 2: Numerical results for several $S_k^{(a)}$-soliton densities (with sizes $k = 1, 2, 3$ and 'color' $a = 1, 2$) as a function of $\zeta = x/t$ for $t = 600$ and $t = 1000$ and for the $T_{l=100}^{(1)}$ dynamics. The red line indicates the GHD prediction for $t = \infty$. The initial state is defined by the following fugacities in the right half: $z_{1,R} = 1$ $z_{2,R} = 0.7$ and $z_{3,R} = 0.3$, and the left half is initially in the vacuum state. Parameters of the simulations: system size $L = 300000$ and number of random initial conditions $N_{\text{samples}} = 400000$.

$\zeta = -\infty$ is empty, and in this region any test color-2 soliton would have a vanishing velocity for the $T^{(r=1)}$ dynamics considered in this example. We thus have a situation where several velocities are degenerate: $\forall i \, v_i^{(2)}(r = 1, l)_{\mathcal{P}_0} = 0$. The edge of $\mathcal{P}_0$ is thus at $\zeta = \zeta(1) = 0$ and all the color-2 soliton densities jump from 0 to some nonzero value at this point, as can be seen in the lower panels of Fig. 2. One can also notice some density spikes at $\zeta = 0$, where the all $S_k^{(2)}$ soliton densities jump from zero to some nonzero value. This phenomenon appears generically at the edge of a vacuum plateau and it also occurs in the $n = 2$ model [11].

Fig. 3 shows the soliton densities in a situation where both the left and the right are in a nontrivial i.i.d. state at $t = 0$ (the left and right fugacities for this example are given in the figure caption). Here again the comparison between the simulations and the densities obtained from GHD are in very good agreement.

In Fig. 4 we go back to a situation where the initial state is the vacuum for the right part, but the dynamics is generated by the product $T_{l=100}^{(1)} T_{l=100}^{(2)}$. In the GHD framework this means that the bare velocity $v_{i,\text{bare}}^{(a)}$ of the $S_i^{(a)}$ solitons is the sum of its value $\delta_{1,a} \min(i, l)$ (see (4.1)) for

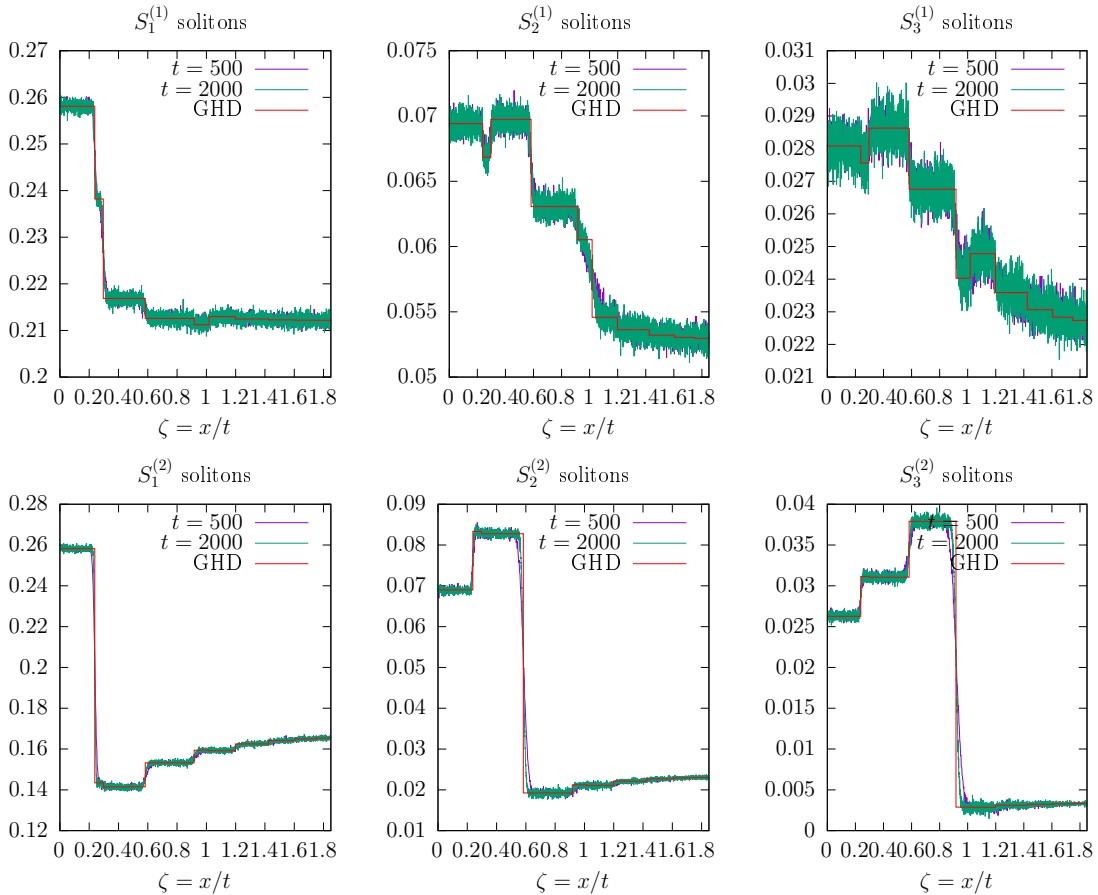

Figure 3: Numerical results for several $S_k^{(a)}$-soliton densities (with sizes $k = 1, 2, 3$ and 'color' $a = 1, 2$) as a function of $\zeta = x/t$ for $t = 500$ and $t = 2000$ and for the $T_{l=100}^{(1)}$ dynamics. The red line indicates the GHD prediction for $t = \infty$. The initial state is defined by the following fugacities in the left half: $z_{1,L} = 1$ $z_{2,L} = 0.7$ and $z_{3,L} = 0.3$, and the following fugacities in the right: $z_{1,R} = 1$ $z_{2,R} = 0.9$ and $z_{3,R} = 0.1$ Parameters of the simulations: system size $L = 640000$ and number of random initial conditions $N_{\text{samples}} = 200000$.

$T_l^{(r=1)}$ and its value $\delta_{2,a}\min(i, l)$ for $T_l^{(r=2)}$. We thus have $v_{i,\text{bare}}^{(a)} = \min(i, l)$ for $T_{l=100}^{(1)} T_{l=100}^{(2)}$. Contrary to the case where the dynamics is generated by $T_l^{(1)}$ or $T_l^{(2)}$, there is no infinite series of narrow plateaux which accumulate in the vicinity of some finite value of $\zeta$.

Fig. 5 represents the 'letter' densities of '1', '2' and '3', for the same parameters as Fig. 1. By specializing (2.47) to the case $n = 3$ we can express these letter densities using the soliton densities: $\varrho_1 = 2 - A$, $\varrho_2 = 1 + A - B$ and $\varrho_3 = B$ with $A = \sum_{i=1}^{\infty} i\rho_i^{(a=1)}$ and $B = \sum_{i=1}^{\infty} i\rho_i^{(a=2)}$. In Fig. 5 an infinite number of very narrow plateaux are observed to accumulate in the vicinity of $\zeta = \zeta_c \simeq 1.88708$. When approaching this limiting value $\zeta_c$ from below, color-2 solitons of increasing size gradually disappear. Beyond that limiting velocity only color-1 solitons are left. In terms of (5.12) the vectors describing the soliton content for $\zeta > \zeta_c$ take the form $\mathbf{d} = (k, \infty)$. A very similar infinite series of narrow plateaux are observed for the $T_{l=3}^{(2)}$ dynamic, as illustrated in Fig. 6. Fig. 7 illustrates this phenomenon for the same initial state but with a few other values of $l$. In other words, this phenomenon does not disappear when taking a small carrier capacity $l$. Such an infinite series is also present with the $T_{l=100}^{(2)}$ dynamics, as

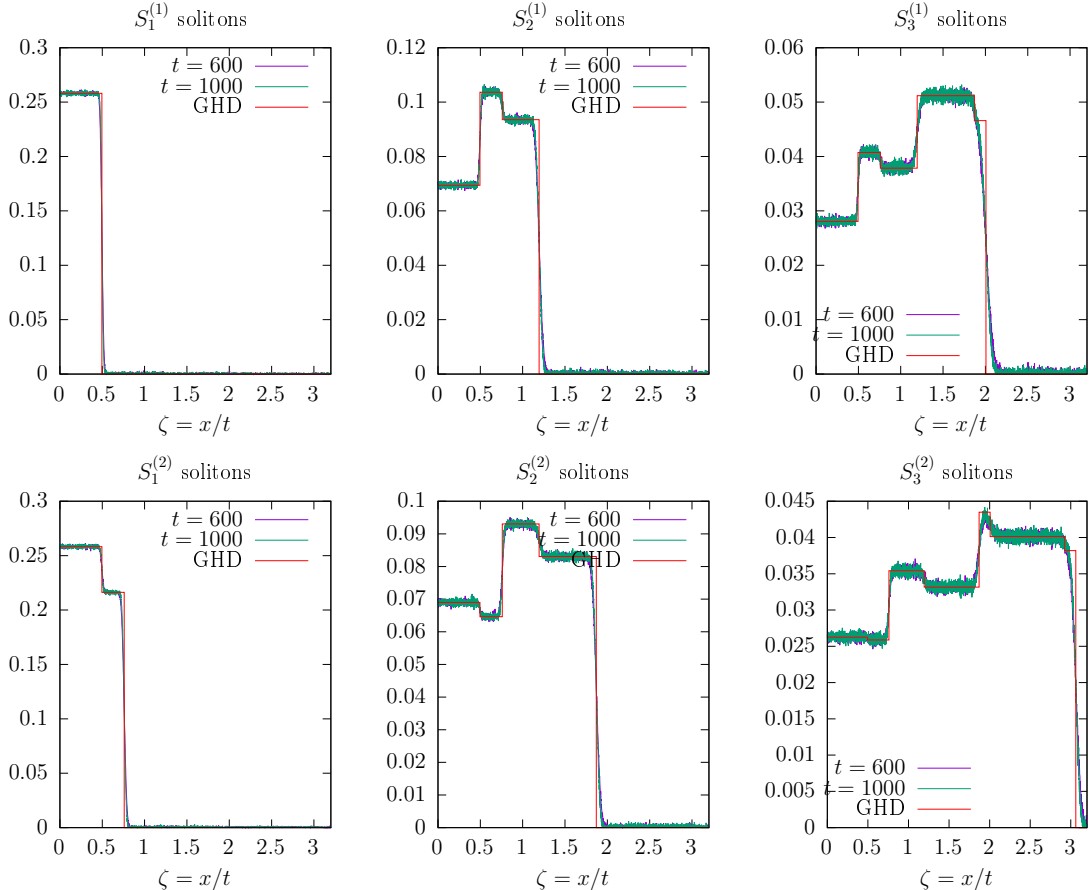

Figure 4: Numerical results for several $S_k^{(a)}$-soliton densities (with sizes $k = 1, 2, 3$ and 'color' $a = 1, 2$) as a function of $\zeta = x/t$ for $t = 600$ and $t = 1000$ and for the $T_{l=100}^{(1)} T_{l=100}^{(2)}$ dynamics. The dotted lines indicate the GHD prediction for the step positions. The initial state is defined by the following fugacities in the left half: $z_{1,L} = 1$ $z_{2,L} = 0.7$ and $z_{3,L} = 0.3$. The right half is initially in the vacuum state. Parameters of the simulations: system size $L = 300000$ and number of random initial conditions $N_{\text{samples}} = 300000$.

illustrated in Fig. 8. In that case the color-1 solitons are absent for $\zeta$ beyond the accumulation point (found at $\zeta_c \simeq 0.80019$).

It should finally be noted that such a phenomenon is absent in the simpler $n = 2$ model [11]. It is also absent if one considers the $n = 3$ complete BBS with the same initial state but with the $T_{l=100}^{(1)} T_{l=100}^{(2)}$ dynamics (Fig. 4).

Fig. 9 shows the velocities $v_{i=1\cdots10}^{(a=1,2)}$ and soliton densities $\rho_{i=1\cdots10}^{(a=1,2)}$ in the four first plateaux of the domain wall problem where the dynamics is generated by $T_{l=3}^{(r=1)}$, and the initial state is characterized by $z_{1,L} = 1$ $z_{2,L} = 0.7$ and $z_{3,L} = 0.3$ and is empty in the right (same as in Fig. 6). The upper panels illustrate the fact that the velocity is not necessarily a monotonic function of the soliton size. In this example the velocity of color-1 solitons is growing for $i = 1, 2, 3$ and then it decreases and approaches a finite limit when $i \to \infty$. As for the color-2 solitons, their velocities turn out to be increasing functions of the size. In the lower panels (densities) on can check that i) the density of $S_1^{(2)}$ solitons vanishes in the plateau $k \geq 1$ (since they are the slowest species in the plateau 0), ii) the density of $S_2^{(2)}$ solitons vanishes in the

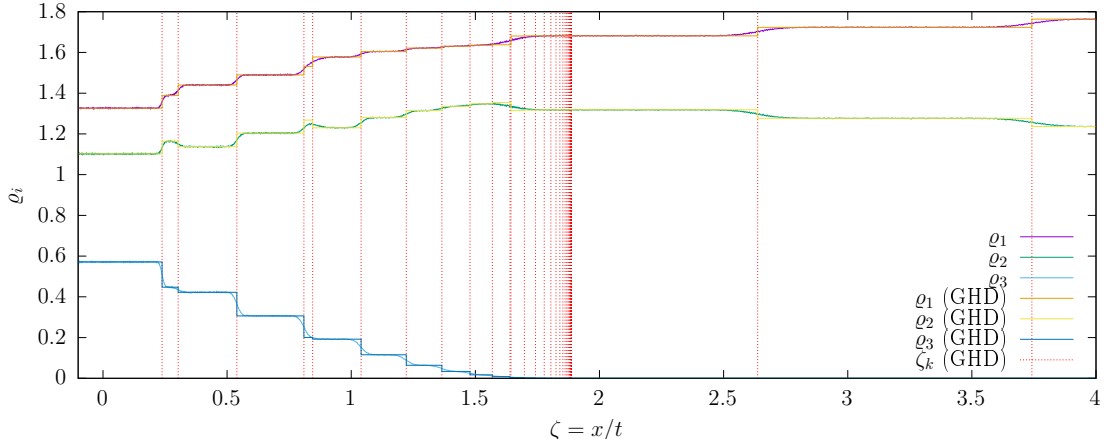

Figure 5: Densities $\varrho_{i=1,2,3}$ of boxes with the entry $i$, as a function of $\zeta = x/t$ for $t = 1000$ and for the $T_{l=100}^{(1)}$ dynamics. Note that by construction $\varrho_1 + \varrho_2 + \varrho_3 = 3$, since each site contains 3 boxes (one in the tableau $B^{1,1}$ and two in the tableau $B^{2,1}$). The vertical red dotted line indicate the locations $\zeta(k)$ of the steps between the different plateaux, as predicted by GHD. The orange, yellow and blue lines are $\varrho_1$, $\varrho_2$ and $\varrho_3$ from GHD (at $t = \infty$). The initial state is defined by the following fugacities in the left half: $z_{1,L} = 1$ $z_{2,L} = 0.7$ and $z_{3,L} = 0.3$. The right half is initially in the vacuum state. A large number of very narrow plateaux are observed to accumulate in the vicinity of $\zeta_c \simeq 1.88708$. Note that additional plateaux are present for $\zeta > 4$ (not shown) and the system is still not in the vacuum state at the right of the figure (contrary to Fig. 6). Parameters of the simulations: system size $L = 300000$ and number of random initial conditions $N_{\text{samples}} = 300000$.

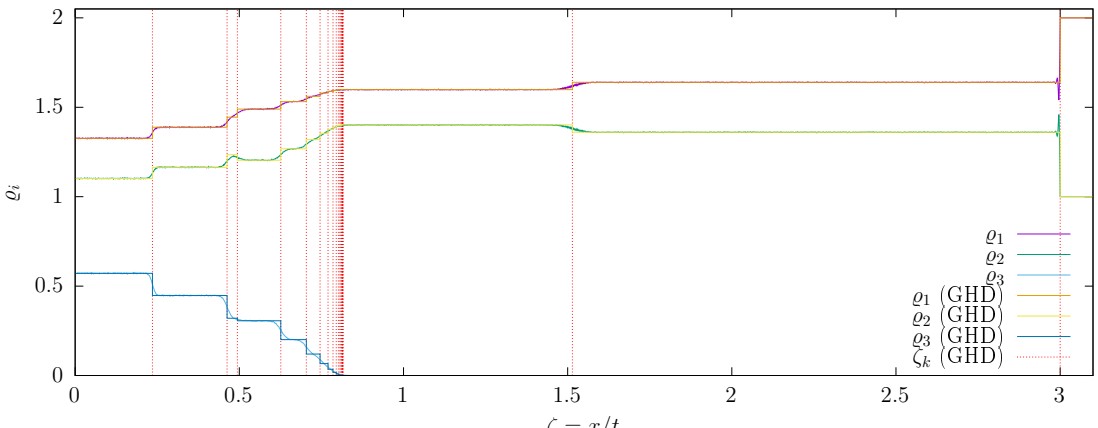

Figure 6: Same as Fig. 5 but for the $T_{l=3}^{(1)}$ dynamics. We again have an infinite series of narrow plateaux which accumulate in the vicinity of $\zeta \simeq 0.81586$. The last step is located at $\zeta = 3$, beyond this point the system is in the vacuum state (*i.e* $\varrho_1 = 2$, $\varrho_2 = 1$ and $\varrho_3 = 0$). Notice the density spike just before $\zeta = 3$. The soliton speeds and densities in the three four plateaux are displayed in Fig. 9.

plateau $k \geq 2$ (since they are the slowest species with non-zero density in the plateau 1), and iii) the density of $S_1^{(1)}$ solitons vanishes in the plateau $k \geq 3$ (since they are the slowest species

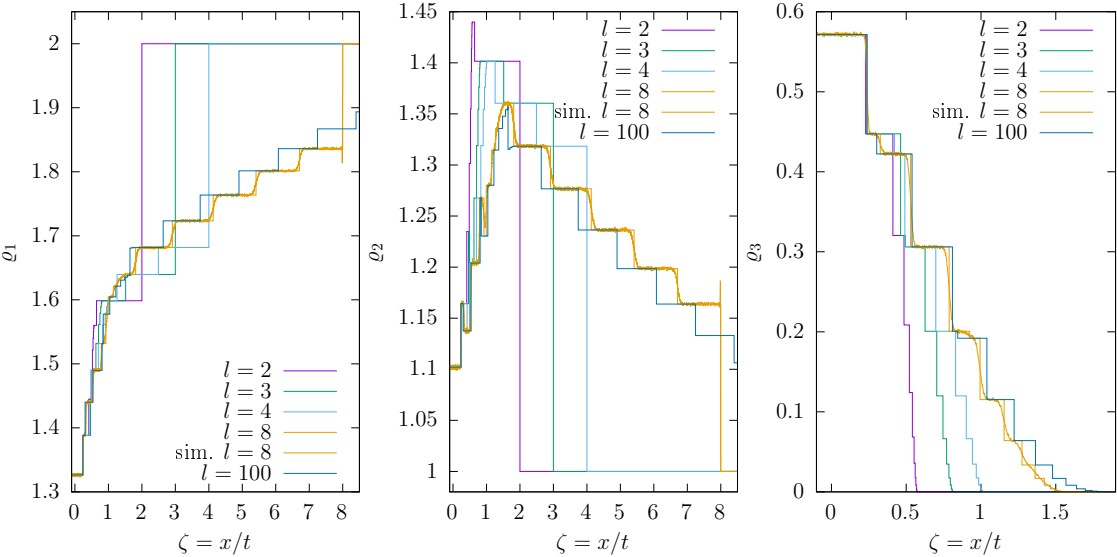

Figure 7: Ball densities $\varrho_{i=1,2,3}$ obtained from GHD calculations and plotted as a function of $\zeta$. The different colors correspond to different values of the parameter $l$ in $T_l^{(1)}$. The initial state is defined by the fugacities $z_{1,L} = 1$ $z_{2,L} = 0.7$, $z_{3,L} = 0.3$ and with the vacuum state in the right part. For $l = 8$ the results of simulations (at $t = 1000$ and $N_{\text{samples}} = 3.10^5$) are also shown. The sum of the three densities is equal to 3 by construction (notice the different vertical scales in the different panels). Three regions can be distinguished: i) In the region $\zeta < \zeta_c(l)$ some soliton of color 1 and color 2 are present. An infinity of narrow plateaux develop when $\zeta \to \zeta_c(l)$, where the color-2 solitons of increasing size gradually disappear. ii) For $\zeta_c(l) < \zeta < l$ only color-1 solitons are present. There is a finite number (of order $l$) of plateaux in this region. iii) For $l < \zeta$ the system is in the vacuum state. The critical values of $\zeta$ are: $\zeta_c(2) \simeq 0.57484$, $\zeta_c(3) \simeq 0.81586$, $\zeta_c(4) \simeq 1.02025$, $\zeta_c(8) \simeq 1.55262$, $\zeta_c(100) \simeq 1.88708$, and they can be identified in the right panel as the points where $\varrho_3$ vanishes.

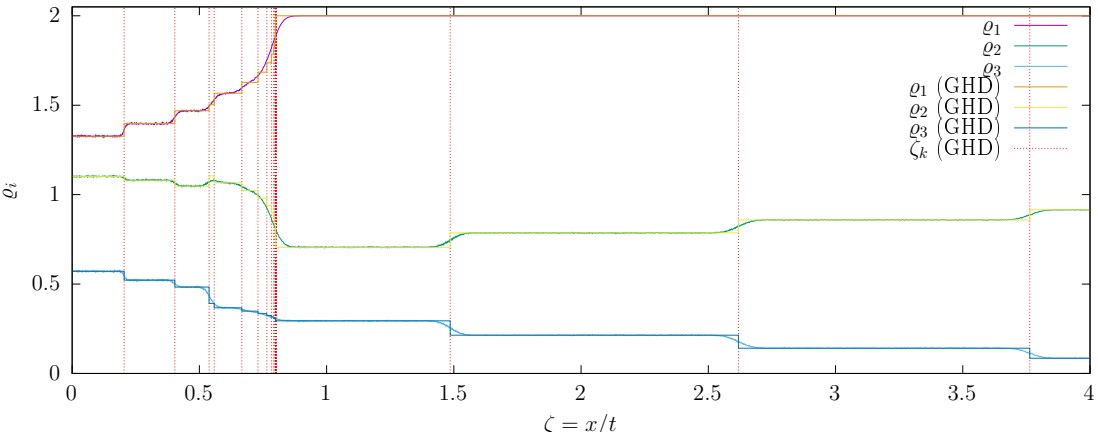

Figure 8: Same as Fig. 5 but for the $T_{l=100}^{(2)}$ dynamics. We again have an infinite series of narrow plateaux, they accumulate here in the vicinity of $\zeta_c \simeq 0.80019$.

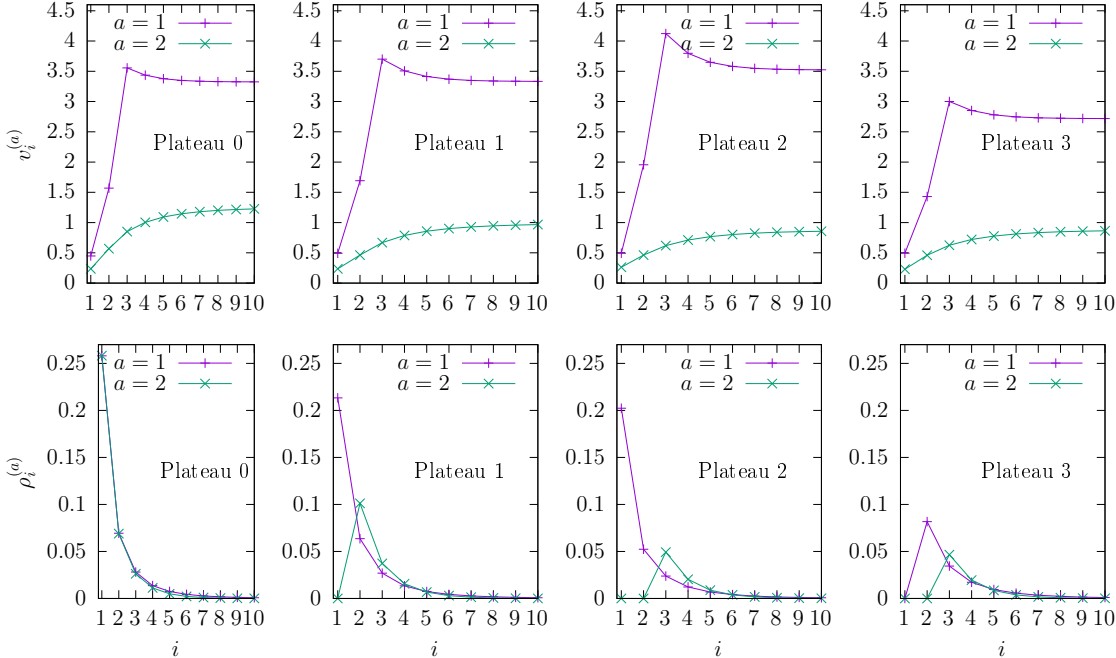

Figure 9: Soliton velocities $v_{i=1\cdots10}^{(a=1,2)}$ (top panels) and soliton densities $\rho_{i=1\cdots10}^{(a=1,2)}$ (bottom panels) plotted as a function of soliton size $i$ in the four first plateaux. These curves are obtained from GHD calculations and for the same parameters as in Fig. 6, that is with the $T_{l=3}^{(r=1)}$ dynamics, fugacities $z_{1,L}=1$ $z_{2,L}=0.7$, $z_{3,L}=0.3$ and with the vacuum state in the right part at $t=0$.

with non-zero density in the plateau 2). For this particular initial state we have checked for the $T_l^{(r=1)}$ dynamics with $l=2,3,\cdots,6$ that the fastest color-1 soliton has size $l$.

Finally Fig. 10 shows that the widths of the steps between consecutive plateaux scale like $t^{1/2}$, in agreement with the diffusive corrections calculated in Sec. 5.2. This figure represents some soliton densities in vicinity of the step $k=1,2,3$, plotted as a function of $(x/t-\zeta(k))t^{1/2}$. The fact that the data measured at different times almost fall onto the same error function curve demonstrates the diffusive nature of the step broadening. Moreover, the data are in good agreement with the GHD prediction (5.19) calculated in Sec. 5.2, including the quantitative value of the parameter $\Sigma_{\mathcal{P},\mathcal{P}'}$. It should in particular be noted that the red curves in Fig. 10 are the GHD predictions and contain no adjustable parameter. In the bottom right panel of Fig. 10 the numerical data appear to be slightly below the GHD curve. The reason for this mismatch is not clear to us but it should however be noticed that it tends to *decrease* when the time increases (compare $t=1000$, 2000 and 3000). It is therefore likely due to some finite-time effect.

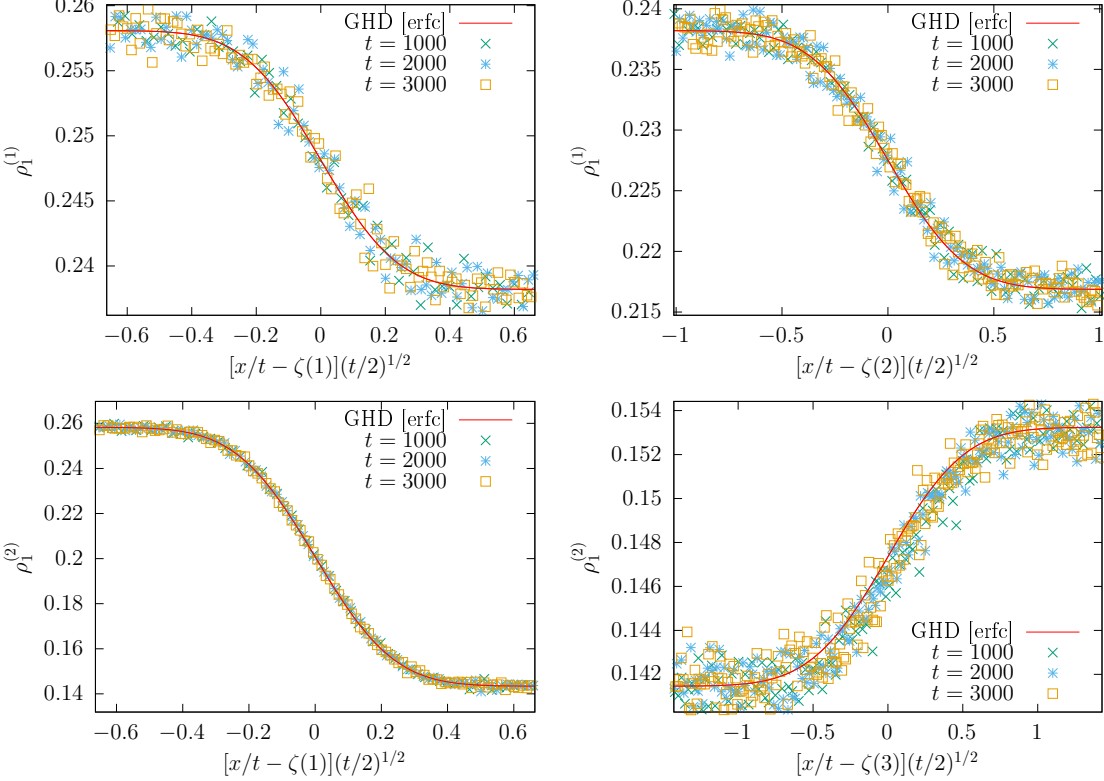

Figure 10: Densities $\rho_{i=1}^{(a)}$ of $S_1^{(a=1,2)}$ solitons in the vicinities of the three first steps (located at $x/t = \zeta(1)$, $x/t = \zeta(2)$ and $x/t = \zeta(3)$) and at three different times. With the choice $(x/t - \zeta(k))t^{1/2}$ for the horizontal scale the curves almost collapse onto each other. This illustrates the diffusive nature of the step broadening. The red curves have been obtained without any free parameter (no fit) and represent the GHD prediction. They take the form of a complementary error function (see Sec. 5.2 and (5.19)). Initial state fugacities: $z_{1,L} = 1$ $z_{2,L} = 0.7$ and $z_{3,L} = 0.3$, $z_{1,R} = 1$ $z_{2,R} = 0.9$ and $z_{3,R} = 0.1$. System size $L = 1.2 10^6$. Number of random initial conditions $N_{\text{samples}} = 400000$. This simulation represents $(2 \cdot L) \cdot N_{\text{samples}} \cdot t_{\text{max}} \sim 0.3 \cdot 10^{16}$ applications of the combinatorial $R$.

## 6 Summary and conclusions

In this paper we have introduced a new family of box ball systems, dubbed *complete* BBS. The models are indexed by a parameter $n \in \mathbb{Z}_{>1}$ and are associated to the quantum group $U_q(\widehat{sl}_n)$. The cBBS family generalizes the original single-color model [1], which is recovered for $n = 2$. Each 'site' of the model is made of the $n - 1$ column shape semistandard tableaux with length $1, 2, \cdots, n-1$. The 'balls' can have $n$ different 'colors' and occupy the boxes of the above tableaux, respecting the semistandard rules. These states are then equipped with some integrable dynamics, constructed from the so-called combinatorial $R$ (Sec. 2.2). As for the simplest BBS, the time evolutions can be viewed as generated by a 'carrier' which propagates through the system at each time step and which takes, moves and deposits 'balls' from one place to another. In the case of the time evolution $T_l^{(k)}$ (2.26), the load of the carrier is itself a rectangular tableau of shape $k \times l$ with $k \in \{1, \ldots, n-1\}$ and $l \in \mathbb{Z}_{>0}$ (Sec. 2.4). The integrability manifests itself from the fact that all these time evolutions commute (whatever $k$

and $l$) and from the existence of a family of conserved energies (2.30).[16] Another consequence of the integrability is the existence of solitons (Sec. 2.5). Each soliton is characterized by a 'color' $a$ and amplitude (or size) $i$. These solitons interact with each other in some nontrivial way during collisions. The local structure of each site of the cBBS may look complicated compared to the other versions of $n$-color BBS, but, as explained in Sec. 2, it offers a remarkable simplification: the soliton scattering is completely *diagonal*, namely each soliton regains its color and amplitude after collisions. The effect of the interactions is encoded in a simple phase shift. The end of Sec. 2 presents the inverse scattering scheme for this model, which is based on the KSS bijection (details in Appendix C), and which allows one to construct the action-angle variables.

The Sec. 3 discusses the randomized cBBS, where we have considered generalized Gibbs ensembles of cBBS configurations. We focused on a special family of GGE, called i.i.d., which are characterized by $n$ fugacities (one for each color) and where all the tableaux are independently distributed (Sec. 3.2). The properties of such i.i.d. GGE can be studied using TBA (Sec. 3.3) and we obtained the free energy, the mean value of the conserved energies and the associated soliton densities. In Sec. 4 we have presented the generalized hydrodynamics of the cBBS. The first ingredient is the effective speed of each soliton species in a homogenous state (see (4.1) and its solution in matrix form (4.31)). This could be obtained thanks to the TBA approach. Next, the currents carried by the soliton of each color are assumed to be given by the product of their effective speed and their density (4.33). For a system which is inhomogeneous but where spatial gradients are sufficiently small so that it can locally be approximated by a GGE, the continuity equations for the soliton currents give the GHD equations (4.39). These equations take a particularly simple form when re-written in terms of the $Y$-variables of the TBA (4.40), which shows that these variables are the normal modes of the GHD. As a concrete application of GHD we studied in Sec. 5 the evolution of the $n = 3$ model starting from a domain wall initial state with two different i.i.d. states in the left and in the right halves. In good quantitative agreement with the GHD prediction, the high-precision simulations showed that at sufficiently long times the state of the system becomes piecewise constant in the variable $\zeta = x/t$ (5.2). Inside each plateau, the soliton densities and the ball densities measured in the simulations are in good agreement with the GHD expectations. Compared to the simpler $n = 2$ model [11], the additional color degree of freedom leads to a complex evolution of the soliton content from one plateau to the next, and opens up the possibility to have an infinity of narrow plateaux which accumulate to some critical point $\zeta_c$ (Figs. 5, 6 and 8). Finally, we have computed in Sec. 5.2 the diffusion effect that is responsible for the broadening of the steps between consecutive plateaux. Here again we observed a good quantitative agreement between the simulations (Fig. 10) and the theoretical results. To our knowledge, this paper achieves the first systematic and successful application of GHD to an integrable system associated with a higher rank quantum algebra.

We wish to conclude this article by mentioning a few possible future directions. The equation (4.1) for the effective velocities is quite fundamental. It would be very interesting to have a proof of it, in the line of what was done in [11, Sec. 4.2] for $n = 2$. See also [25] for $n = 2$. It would also be appealing to invent and explore other protocols to set the system out of equilibrium. This might be done, for example, with a system that is open at its boundaries. One could also think of more general initial states, homogeneous or inhomogeneous, defined for instance by some nontrivial weights on the configurations, and look at the evolution toward equilibrium. This could be, for instance, a domain wall between two general (non-i.i.d.) GGE. In

---

[16]As a comparison, the classical hard rod model [19] (for which the GHD has also been worked out) has an infinite number of conserved quantities (the numbers of rods of each size) but it lacks the commuting dynamics. A similar remark also holds, for instance, for the classical cellular automata studied in Refs. [47,48], where domain-wall problems similar to those studied here have been solved exactly but where commuting transfer matrices are not known.

the spirit of what have been done for some simple one-dimensional cellular automata [47, 48] it would be interesting to compute some two-time correlation functions and to relate these correlations to transport coefficients. Another interesting question would be to explore the possibility of some anomalous transport in these models. Investigating numerically other values of $n$, possibly large ones, might also reveal some new phenomena. It also seems worth comparing the dynamics of the present cBBS with non-complete $n$-color BBS [14, 17, 20]. Finally, a possible direction would be to include some integrability-breaking perturbations in the model. We may expect the disappearance of the density plateaux, and it should be possible to observe numerically and to analyze some possible crossover from the integrable regime to a chaotic one where solitons are no longer stable but only short-lived. Such a setup could give rise to some prethermalization [49], possibly described by some GGE.

## A   Algorithm for the combinatorial $R$

We use the notation in Sec. 2.2. Given $b \otimes c \in B^{k,l} \otimes B^{k',l'}$, we present an algorithm for finding the image $\tilde{c} \otimes \tilde{b} = R(b \otimes c) \in B^{k',l'} \otimes B^{k,l}$ of the combinatorial $R$ following [35, p55]. As noted after (2.7), the task is to construct the solution to the equation $\tilde{b} \cdot \tilde{c} = c \cdot b$ which is unique when the tableaux are rectangular.

A *skew tableau* $\theta$ is a part of a semistandard Young tableau having a skew Young diagram shape. We call $\theta$ an *m-vertical strip* if it contains at most one box on each row and the total number of the boxes is $m$.

Let $P = c \cdot b$ be the product tableau [12]. Its shape Young diagram includes a $k \times l$ rectangle. Denote by $P'$ the NW part of $P$ corresponding to the $k \times l$ rectangle, and by $P/P'$ the skew tableau. The number of boxes of these tableaux (denoted by $|\cdot|$) are given by

$$|P| = kl + k'l', \qquad |P'| = kl, \qquad |P/P'| = k'l'. \tag{A.1}$$

*Step 1*. Label each box of $P/P'$ with $\{1, 2, \ldots, k'l'\}$.

Let $\theta_1$ be the rightmost vertical $k'$-strip in $P/P'$ as upper as possible. Remove $\theta_1$ from $P/P'$ and define the vertical $k'$-strip $\theta_2$ in a similar manner. This can be continued until $P/P'$ is decomposed into the disjoint union of the vertical $k'$-strips $\theta_1, \theta_2, \ldots, \theta_{l'}$. Now the label is obtained by assigning the boxes of $\theta_i$ with $k'(i-1) + 1, k'(i-1) + 2, \ldots, k'(i-1) + k'$ from the bottom to the top.

*Step 2*. Reverse row bumping from $P$ in the order of the label.

Find the semistandard Young tableaux $P_1, P_2, \ldots, P_{k'l'}$ and $w_1, w_2, \ldots, w_{k'l'} \in [1, n]$ such that

$$
\begin{aligned}
P &= (P_1 \leftarrow w_1); && \text{reverse bumping of the letter in the box } 1, \\
P_1 &= (P_2 \leftarrow w_2); && \text{reverse bumping of the letter in the box } 2, \\
&\cdots \\
P_{k'l'-1} &= (P_{k'l'} \leftarrow w_{k'l'}); && \text{reverse bumping of the letter in the box } k'l'.
\end{aligned}
\tag{A.2}
$$

Here $\leftarrow$ stands of the row insertion as in Sec. 2.2. The splitting of $P_{i-1}$ into $(P_i \leftarrow w_i)$ is done by the reverse row bumping. Starting from the box labeled $i$ in *Step 1* and the letter in it, the bumping goes upwards row by row as follows. Given a letter $\alpha$ in a row, find the box in the adjacent upper row filled with the largest $\beta$ such that $\beta < \alpha$ and the rightmost one in case more than one $\beta$ is contained. Then let $\alpha$ occupy the box by bumping out $\beta$. Repeating this procedure one eventually bumps out a letter $w_i$ from the top row of $P_{i-1}$ thereby getting also $P_i$ as the remnant tableau. Note that $|P_{k'l'}| = kl$.

*Step 3*. The image $\tilde{c} \otimes \tilde{b} \in B^{k,l} \otimes B^{k',l'}$ is constructed as

$$\tilde{b} = P_{k'l'}, \tag{A.3}$$

$$\tilde{c} = ((\cdots (\emptyset \leftarrow w_{k'l'}) \leftarrow \cdots) \leftarrow w_2) \leftarrow w_1. \tag{A.4}$$

The semistandard tableau (A.4) is known as the $P$-symbol of the word $w_{k'l'} \dots w_2 w_1$.

**Example A.1.** Consider $b \in B^{2,3}$ and $c \in B^{3,2}$ in (2.3). Their product $c \cdot b$ is the right tableau in (2.6). The labeling of the boxes of $P/P'$ with $\{1, 2, \dots, 6\}$ according to *Step 1* is shown by the indices in

$$\begin{array}{|c|c|c|c|c|} \hline 1 & 1 & 2 & 2_6 & 3_3 \\ \hline 2 & 2 & 4 & 5_2 \\ \cline{1-4} 3_5 & 5_1 \\ \cline{1-2} 4_4 \\ \cline{1-1} \end{array} \quad . \tag{A.5}$$

The reverse bumping in *Step 2* is done along the order of these indices as follows:

$$\begin{array}{|c|c|c|c|c|} \hline 1&1&2&2&3 \\ \hline 2&2&4&5 \\ \cline{1-4} 3&5 \\ \cline{1-2} 4 \\ \cline{1-1} \end{array} = \left( \begin{array}{|c|c|c|c|c|} \hline 1&1&2&2&4 \\ \hline 2&2&5&5 \\ \cline{1-4} 3 \\ \cline{1-1} 4 \\ \cline{1-1} \end{array} \leftarrow 3 \right), \tag{A.6}$$

$$\begin{array}{|c|c|c|c|c|} \hline 1&1&2&2&4 \\ \hline 2&2&5&5 \\ \cline{1-4} 3 \\ \cline{1-1} 4 \\ \cline{1-1} \end{array} = \left( \begin{array}{|c|c|c|c|c|} \hline 1&1&2&2&5 \\ \hline 2&2&5 \\ \cline{1-3} 3 \\ \cline{1-1} 4 \\ \cline{1-1} \end{array} \leftarrow 4 \right), \tag{A.7}$$

$$\begin{array}{|c|c|c|c|c|} \hline 1&1&2&2&5 \\ \hline 2&2&5 \\ \cline{1-3} 3 \\ \cline{1-1} 4 \\ \cline{1-1} \end{array} = \left( \begin{array}{|c|c|c|c|} \hline 1&1&2&2 \\ \hline 2&2&5 \\ \cline{1-3} 3 \\ \cline{1-1} 4 \\ \cline{1-1} \end{array} \leftarrow 5 \right), \tag{A.8}$$

$$\begin{array}{|c|c|c|c|} \hline 1&1&2&2 \\ \hline 2&2&5 \\ \cline{1-3} 3 \\ \cline{1-1} 4 \\ \cline{1-1} \end{array} = \left( \begin{array}{|c|c|c|c|} \hline 1&2&2&2 \\ \hline 2&3&5 \\ \cline{1-3} 4 \\ \cline{1-1} \end{array} \leftarrow 1 \right), \tag{A.9}$$

$$\begin{array}{|c|c|c|c|} \hline 1&2&2&2 \\ \hline 2&3&5 \\ \cline{1-3} 4 \\ \cline{1-1} \end{array} = \left( \begin{array}{|c|c|c|c|} \hline 1&2&2&3 \\ \hline 2&4&5 \\ \hline \end{array} \leftarrow 2 \right), \tag{A.10}$$

$$\begin{array}{|c|c|c|c|} \hline 1&2&2&3 \\ \hline 2&4&5 \\ \hline \end{array} = \left( \begin{array}{|c|c|c|} \hline 1&2&2 \\ \hline 2&4&5 \\ \hline \end{array} \leftarrow 3 \right). \tag{A.11}$$

Therefore from *Step 3* we obtain

$$\tilde{b} = \begin{array}{|c|c|c|} \hline 1&2&2 \\ \hline 2&4&5 \\ \hline \end{array}, \tag{A.12}$$

$$\tilde{c} = (((((\emptyset \leftarrow 3) \leftarrow 2) \leftarrow 1) \leftarrow 5) \leftarrow 4) \leftarrow 3 = \begin{array}{|c|c|} \hline 1&3 \\ \hline 2&4 \\ \hline 3&5 \\ \hline \end{array}. \tag{A.13}$$

This yields (2.9).

# B  Combinatorial $R$ and energy $H$ for $n = 2, 3$

We consider $n = 3$ case first. The sets $B^{1,l}, B^{2,l}$ of semistandard tableaux are parameterized as

$$B^{1,l} = \{x = (x_1, \dots, x_n) \in (\mathbb{Z}_{\geq 0})^n \mid x_1 + \cdots + x_n = l\}, \; x_i = \# \text{ of } i \text{ in tableau}, \tag{B.1}$$

$$B^{2,l} = \{x = (x_1, \dots, x_n) \in (\mathbb{Z}_{\geq 0})^n \mid x_1 + \cdots + x_n = l\}, \; x_i = \# \text{ of columns}$$
$$\text{without } i \text{ in tableau.} \tag{B.2}$$

We extend the indices to $\mathbb{Z}$ by $x_i = x_{i+n}$ and similarly also for $Q_i(x, y)$ and $P_i(x, y)$ introduced below. Then the combinatorial $R$ and the energy $H$ are given by the piecewise linear formulas as follows ($n = 3$) [21, eqs. (2.1)-(2.4)]:

$$R_{B^{1,l} \otimes B^{1,l'}} : \quad B^{1,l} \otimes B^{1,l'} \longrightarrow B^{1,l'} \otimes B^{1,l}; \quad x \otimes y \longmapsto \tilde{y} \otimes \tilde{x} \tag{B.3}$$

$$\tilde{x}_i = x_i + Q_i(x, y) - Q_{i-1}(x, y), \quad \tilde{y}_i = y_i + Q_{i-1}(x, y) - Q_i(x, y),$$

$$Q_i(x, y) = \min\{\sum_{j=1}^{k-1} x_{i+j} + \sum_{j=k+1}^{n} y_{i+j} \mid 1 \leq k \leq n\},$$

$$H_{B^{1,l} \otimes B^{1,l'}}(x \otimes y) = Q_0(x, y). \tag{B.4}$$

$$R_{B^{1,l} \otimes B^{2,l'}} : \quad B^{1,l} \otimes B^{2,l'} \longrightarrow B^{2,l'} \otimes B^{1,l}; \quad x \otimes y \longmapsto \tilde{y} \otimes \tilde{x} \tag{B.5}$$

$$\tilde{x}_i = x_i + P_{i+1}(x, y) - P_i(x, y), \quad \tilde{y}_i = y_i + P_{i+1}(x, y) - P_i(x, y),$$

$$P_i(x, y) = \min(x_i, y_i),$$

$$H_{B^{1,l} \otimes B^{2,l'}}(x \otimes y) = P_1(x, y). \tag{B.6}$$

$$R_{B^{2,l} \otimes B^{1,l'}} : \quad B^{2,l} \otimes B^{1,l'} \longrightarrow B^{1,l'} \otimes B^{2,l}; \quad x \otimes y \longmapsto \tilde{y} \otimes \tilde{x} \tag{B.7}$$

$$\tilde{x}_i = x_i + P_{i-1}(x, y) - P_i(x, y), \quad \tilde{y}_i = y_i + P_{i-1}(x, y) - P_i(x, y),$$

$$H_{B^{2,l} \otimes B^{1,l'}}(x \otimes y) = P_0(x, y). \tag{B.8}$$

$$R_{B^{2,l} \otimes B^{2,l'}} : \quad B^{2,l} \otimes B^{2,l'} \longrightarrow B^{2,l'} \otimes B^{2,l}; \quad x \otimes y \longmapsto \tilde{y} \otimes \tilde{x} \tag{B.9}$$

$$\tilde{x}_i = x_i + Q_{i-1}(y, x) - Q_i(y, x), \quad \tilde{y}_i = y_i - Q_{i-1}(y, x) + Q_i(y, x),$$

$$H_{B^{2,l} \otimes B^{2,l'}}(x \otimes y) = Q_0(y, x). \tag{B.10}$$

These formulas do not depend on $l, l'$ explicitly. When $n = 2$, the relevant objects are (B.1), (B.3) and (B.4) only, and one can just set $n = 2$ there.

For instance $Q_0(x, y) = \min\{y_2 + y_3, x_1 + y_3, x_1 + x_2\}$ for $n = 3$ and $Q_0(x, y) = \min\{y_2, x_1\}$ for $n = 2$.

## C KSS bijection

### C.1 Paths

Consider the product set of the form $\mathcal{B} = B^{k_1,1} \otimes B^{k_2,1} \otimes \cdots \otimes B^{k_N,1}$ with $k_i \in [1, n-1]$ containing column shape tableaux only. For the definition of $B^{r,s}$, see Sec. 2.2. States of our cBBS correspond to the choice $k_i \equiv i \mod n - 1$ and $N = (n-1)L$. In this appendix, we do not assume the boundary condition of the BBS, and call the elements of $\mathcal{B}$ *paths*. Our aim is to explain the special case of the KSS bijection [50] corresponding to the above $\mathcal{B}$ along the convention adapted to this paper.

For a path $\mathfrak{p} = c_1 \otimes \cdots \otimes c_N \in \mathcal{B}$ with $c_j \in B^{k_j,1}$, the $n$-array $\lambda = (\lambda_1, \ldots, \lambda_n)$ defined by

$$\mathrm{wt}(\mathfrak{p}) = (\lambda_1, \ldots, \lambda_n) \in (\mathbb{Z}_{\geq 0})^n, \qquad \lambda_a = \sum_{j=1}^{N}(\text{number of letter } a \text{ contained in } c_j) \tag{C.1}$$

is called the *weight* of $\mathfrak{p}$. Let $c_{jl}$ be the $l$ th entry of the tableau $c_j \in B^{k_j,1}$ from the top. Consider the word

$$w_1 w_2 \ldots w_{N'} := c_{11} c_{12} \ldots c_{1k_1} c_{21} c_{22} \ldots c_{2k_2} \ldots c_{N1} c_{N2} \ldots c_{Nk_N}, \tag{C.2}$$

where $N' = k_1 + \cdots + k_N$. The path $\mathfrak{p}$ is *highest* if

$$\#_1(w_1 w_2 \ldots w_m) \geq \#_2(w_1 w_2 \ldots w_m) \geq \cdots \geq \#_n(w_1 w_2 \ldots w_m) \quad (\forall m \in [1, N']), \quad \text{(C.3)}$$

where $\#_a(w_1 w_2 \ldots w_m)$ stands for the number of occurrences of $a$ in the subword $w_1 w_2 \ldots w_m$. We introduce the sets of paths

$$\mathcal{P}(\mathcal{B}, \lambda) = \{\mathfrak{p} \in \mathfrak{B} \mid \text{wt}(\mathfrak{p}) = \lambda\}, \quad \text{(C.4)}$$
$$\mathcal{P}_+(\mathcal{B}, \lambda) = \{\mathfrak{p} \in \mathcal{P}(\mathcal{B}, \lambda) \mid \mathfrak{p} \text{ is highest}\}. \quad \text{(C.5)}$$

By the definition, $\mathcal{P}_+(\mathcal{B}, \lambda)$ is empty unless $\lambda_1 \geq \lambda_2 \cdots \geq \lambda_n$. An example from $\mathcal{P}_+(B^{\otimes 7}, (10, 7, 4))$ with $B = B^{1,1} \otimes B^{2,1}$ and $n = 3$ is

$$\mathfrak{p}_0 = \boxed{1} \otimes \boxed{\begin{smallmatrix}1\\2\end{smallmatrix}} \otimes \boxed{1} \otimes \boxed{\begin{smallmatrix}1\\2\end{smallmatrix}} \otimes \boxed{1} \otimes \boxed{\begin{smallmatrix}1\\2\end{smallmatrix}} \otimes \boxed{2} \otimes \boxed{\begin{smallmatrix}1\\3\end{smallmatrix}} \otimes \boxed{3} \otimes \boxed{\begin{smallmatrix}2\\3\end{smallmatrix}} \otimes \boxed{2} \otimes \boxed{\begin{smallmatrix}1\\2\end{smallmatrix}} \otimes \boxed{1} \otimes \boxed{\begin{smallmatrix}1\\3\end{smallmatrix}}. \quad \text{(C.6)}$$

## C.2 Rigged configurations

We keep $\mathcal{B} = B^{k_1,1} \otimes \cdots \otimes B^{k_N,1}$ as above. Consider a multiset (a set allowing multiplicity of elements)

$$S = \{(a_i, j_i, r_i) \in [1, n-1] \times \mathbb{Z}_{\geq 1} \times \mathbb{Z} \mid i = 1, 2, \ldots M\}, \quad \text{(C.7)}$$

where $M \geq 0$ is arbitrary. Each triplet $s = (a, j, r)$ in $S$ is called a *string*. It carries *color*, *length* and *rigging* denoted by $\text{cl}(s) = a$, $\text{lg}(s) = j$ and $\text{rg}(s) = r$, respectively. $S$ is a *rigged configuration* for $\mathcal{B}$ (or just a rigged configuration for short) if $0 \leq r_i \leq p_{j_i}^{(a_i)}$ for all $i$ in (C.7), i.e.,

$$0 \leq \text{rg}(s) \leq p_{\text{lg}(s)}^{(\text{cl}(s))} \quad (\forall \text{ string } s \in S). \quad \text{(C.8)}$$

Here $p_j^{(a)}$ is the *vacancy* defined by

$$p_j^{(a)} = \sum_{i=1}^{N} \delta_{a,k_i} - \sum_{t \in S} C_{a,\text{cl}(t)} \min(j, \text{lg}(t)). \quad \text{(C.9)}$$

See (2.2) for the definition of $C_{ab}$. If the multiplicity of the color $a$ length $j$ strings in $S$ is denoted by $m_j^{(a)}$, the definition (C.9) is rephrased as

$$p_j^{(a)} = \sum_{i=1}^{N} \delta_{a,k_i} - \sum_{b=1}^{n-1} C_{ab} \mathcal{E}_j^{(b)}, \qquad \mathcal{E}_j^{(a)} = \sum_{k \geq 1} \min(j, k) m_k^{(a)}. \quad \text{(C.10)}$$

From (C.8) it is necessary to satisfy $p_{\text{lg}(s)}^{(\text{cl}(s))} \geq 0$ for all $s \in S$, which is already a nontrivial constraint on the multiset $\{(a_i, j_i) \mid i = 1, \ldots, M\}$ depending on $k_1, \ldots, k_N$. This is the reason why the rigged configurations are defined with respect to a given $\mathcal{B} = B^{k_1,1} \otimes B^{k_2,1} \otimes \cdots \otimes B^{k_N,1}$. The dependence on $k_1, \ldots, k_N$ comes from (C.8) via the first terms in the RHS of (C.9) or (C.10).

We regard a string $s = (a, j, r)$ as a length $j$ row with the rigging $r$ attached to its right side. Collecting such color $a$ strings yields a Young diagram whose rows are assigned with riggings. Further collecting them for the colors $a = 1, 2, \ldots, n-1$ leads to a combinatorial object, which was the original rigged configuration in the literatures [51, 52]. For instance for $n = 3$,

$$S_0 = \{(1, 3, 1), (1, 1, 2), (2, 2, 2), (2, 1, 3), (2, 1, 3)\} \quad \text{(C.11)}$$

is depicted as

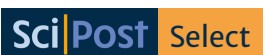

$$(C.12)$$

where the left and the right objects represent the collection of color 1 and 2 strings, respectively. The multiplicity $m_j^{(a)}$ appearing in (C.10) is given by $m_3^{(1)} = 1, m_1^{(1)} = 1, m_2^{(2)} = 1, m_1^{(2)} = 2$. In general, strings with the same color and length form a rectangular block. Objects obtained by permuting the riggings attached to them should not be distinguished since originally it is just the multiset (C.7). Practically, one needs to keep track of the vacancy (C.10) to check (C.8) and this requires the data $\mathcal{B}$ as well. So it is customary and convenient to also include these information in the graphical representation. For instance if $\mathcal{B} = B^{\otimes 7}$ with $B = B^{1,1} \otimes B^{2,1}$, (C.12) is more detailed as

$$(C.13)$$

The number assigned in the left of each rectangular block (*not* a row) is the vacancy. In this example they are $p_3^{(1)} = 3, p_1^{(1)} = 6, p_2^{(2)} = 2, p_1^{(2)} = 3$, and the condition (C.8) is certainly satisfied.

In general we have the Young diagrams $\mu^{(1)}, \ldots, \mu^{(n-1)}$ encoding the color and length of strings, which form the partial data $(\mu^{(1)}, \ldots, \mu^{(n-1)})$ called *configuration*. It corresponds to the multiset $\{(a_i, j_i) \mid i = 1, \ldots, M\}$ obtained from (C.7). The symbol $r^{(a)}$ in (C.13) stands for the rigging attached to $\mu^{(a)}$. To compute the vacancy $p_j^{(a)}$, it is useful to recognize that $\mathcal{E}_j^{(a)}$ in (C.10) is the number of boxes in the left $j$ columns of $\mu^{(a)}$.

The *weight* of a rigged configuration $S$ for $\mathcal{B}$ is an $n$-array $\lambda = (\lambda_1, \ldots, \lambda_n)$ defined by

$$\text{wt}(S) = (\lambda_1, \ldots, \lambda_n) \in (\mathbb{Z}_{\geq 0})^n, \qquad \lambda_a = \sum_{i=1}^{N} \theta(k_i \geq a) - \mathcal{E}_\infty^{(a)} + \mathcal{E}_\infty^{(a-1)}, \qquad (C.14)$$

where $\mathcal{E}_j^{(0)} = \mathcal{E}_j^{(n)} = 0$ is taken for granted and $\theta(\text{true}) = 1, \theta(\text{false}) = 0$. One can show $\lambda_1 \geq \cdots \geq \lambda_n \geq 0$ from $\forall p_j^{(a)} \geq 0$ by noting $\lambda_a - \lambda_{a+1} = p_\infty^{(a)}$.[17] Note that the weight only depends on the configuration and not on the rigging. Now we introduce the set of rigged configurations

$$\text{RC}(\mathcal{B}, \lambda) = \{S : \text{rigged configuration for } \mathcal{B} \mid \text{wt}(S) = \lambda\}. \qquad (C.15)$$

Note that the same symbol $\lambda$ has been used to denote the weight of a path in (C.1) and that of a rigged configuration in (C.14). This we did intentionally since they are going to be identified under the KSS bijection (C.17) in the next subsection.

The number of rigged configurations is given by the so called *Fermionic formula* [3, 40, 51–54]:

$$\#\text{RC}(\mathcal{B}, \lambda) = \sum_{\{m_j^{(a)}\}}^{(\lambda)} \prod_{a=1}^{n-1} \prod_{j \geq 1} \binom{p_j^{(a)} + m_j^{(a)}}{m_j^{(a)}}, \qquad (C.16)$$

---

[17]A slightly nontrivial argument is necessary to derive $\forall p_j^{(a)} \geq 0$ from $p_{\text{lg}(s)}^{(\text{cl}(s))} \geq 0$ for the *existing* strings $s$ only.

where the superscript of the sum means the weight constraint (C.14) on $\{m_j^{(a)}\}$ via (C.10). This follows simply from (C.8) since the number of choices of the riggings attached to the color $a$ length $j$ strings are equal to the number of integers $r_1, \ldots, r_{m_j^{(a)}}$ satisfying

$$0 \leq r_1 \leq \cdots \leq r_{m_j^{(a)}} \leq$$

$p_j^{(a)}$, which is the binomial coefficient in (C.16).

## C.3 KSS bijection

There is a one to one correspondence between the finite sets $\mathcal{P}_+(\mathcal{B}, \lambda)$ and $\mathrm{RC}(\mathcal{B}, \lambda)$. It was shown in a more general case of $\mathcal{B} = B^{k_1, s_1} \otimes \cdots \otimes B^{k_N, s_N}$ in [50] generalizing the pioneering works [51, 52]. Their original motivation was to provide a bijective (combinatorial) proof of the so called *Fermionic* character formula for the finite dimensional representations of quantum affine algebras. It originates in the string hypothesis in the Bethe ansatz. See the introductions of [40, 53] for the rich history going back to Bethe himself [3]. Here we describe the algorithm for the bijection

$$\Phi : \ \mathcal{P}_+(\mathcal{B}, \lambda) \xrightarrow{1:1} \mathrm{RC}(\mathcal{B}, \lambda). \tag{C.17}$$

It works recursively with respect to $\mathcal{B}$ and $\lambda$ along the commutative diagram[18]:

$$
\begin{array}{ccc}
\mathcal{P}_+(\mathcal{B} \otimes B^{k,1}, \lambda) & \xrightarrow{\ \Phi\ } & \mathrm{RC}(\mathcal{B} \otimes B^{k,1}, \lambda) \\
\mathrm{rb} \downarrow & & \downarrow \delta \\
\bigcup_{\nu \in \lambda^-} \mathcal{P}_+(\mathcal{B} \otimes B^{k-1,1}, \nu) & \xrightarrow{\ \Phi\ } & \bigcup_{\nu \in \lambda^-} \mathrm{RC}(\mathcal{B} \otimes B^{k-1,1}, \nu)
\end{array}
\tag{C.18}
$$

Here $\lambda^- = \{\lambda - \mathbf{e}_1, \lambda - \mathbf{e}_2, \ldots, \lambda - \mathbf{e}_n\} \cap (\mathbb{Z}_{\geq 0})^n$ with $\mathbf{e}_a$ being the elementary vector whose only nonvanishing component is 1 at the $a$ th position. The map rb (right box removal) is defined by

$$\mathrm{rb}(c_1 \otimes \cdots \otimes c_N) = \begin{cases} c_1 \otimes \cdots \otimes c_{N-1} \otimes c'_N & (k > 1), \\ c_1 \otimes \cdots \otimes c_{N-1} & (k = 1), \end{cases} \tag{C.19}$$

where $c'_N \in B^{k-1,1}$ is obtained from the tableau $c_N \in B^{k,1}$ just by removing its bottom box and letter. If the letter removed by rb in (C.18) is $x$, then $\nu = \lambda - \mathbf{e}_x$ by the definition (C.1). From (C.3) it is clear that if $\mathfrak{p}$ is highest, $\mathrm{rb}(\mathfrak{p})$ is also highest. In the example (C.6), $\mathrm{rb}(\mathfrak{p}_0)$ differs from $\mathfrak{p}_0$ only by its rightmost component which becomes $\boxed{1}$. The main part of the algorithm lies in $\delta$ in (C.18), which we shall explain below in two ways which fit the calculation of $\Phi^{-1}$ and $\Phi$. A string $(a_i, j_i, r_i)$ in a rigged configuration is *singular* if the rigging attains the allowed maximum, i.e., $r_i = p_{j_i}^{(a_i)}$.

*Algorithm for* $\Phi^{-1}$. Given $k \in [1, n-1]$ and a rigged configuration $(\mu, r) = ((\mu^{(1)}, r^{(1)}), \ldots, (\mu^{(n-1)}, r^{(n-1)})) \in \mathrm{RC}(\mathcal{B} \otimes B^{k,1}, \lambda)$ on the top right corner of (C.18), we are going to define $\mathrm{rk}(\mu, r) \in [1, n]$ called *rank* and a new rigged configuration $\delta(\mu, r) \in \mathrm{RC}(\mathcal{B} \otimes B^{k-1,1}, \lambda - \mathbf{e}_{\mathrm{rk}(\mu, r)})$ which is 'smaller' than $(\mu, r)$. The rank $\mathrm{rk}(\mu, r)$ is the letter inscribed in the removed box in (C.19) in the corresponding path $\Phi^{-1}(\mu, r)$. Therefore, once we know $\mathrm{rk}(\mu, r)$ and $\delta(\mu, r)$, the calculation of the image $\Phi^{-1}(\mu, r)$ is reduced to $\Phi^{-1}(\delta(\mu, r))$. This reduction can be iterated until $\emptyset = \Phi^{-1}(\emptyset)$ is reached, producing the image path $\mathfrak{p} \in \Phi^{-1}(\mu, r)$ letter by letter from the right in the notation of (C.2).

---

[18]By $\Phi$ we actually mean the totality of the maps (C.17) for all $(\mathcal{B}, \lambda)$.

(i) Set $\ell^{(k-1)} = 1$. Do the following procedure for $a = k, k+1, \ldots, n$ in this order until stopped. Find the shortest color $a$ singular string $(a, l, p_l^{(a)})$ in $(\mu^{(a)}, r^{(a)})$ such that $\ell^{(a-1)} \leq l.$[19] If no such $l$ exists, set $\mathrm{rk}(\mu, r) = a$ and stop. Otherwise set $\ell^{(a)} = l$ and continue with $a+1$. This procedure certainly ends at most with $\mathrm{rk}(\mu, r) = n$ since there is no $(\mu^{(n)}, r^{(n)})$.

(ii) Suppose $\mathrm{rk}(\mu, r) = b$ and let $(k, \ell^{(k)}, *), (k+1, \ell^{(k+1)}, *), \ldots, (b-1, \ell^{(b-1)}, *)$ be the so found singular strings, where $*$ denotes the rigging equal to the respective vacancy.

(iii) The new rigged configuration $\delta(\mu, r)$ is obtained by replacing them with $(k, \ell^{(k)}-1, \sharp)$, $(k+1, \ell^{(k+1)}-1, \sharp), \ldots, (b-1, \ell^{(b-1)}-1, \sharp)$ keeping all the other strings unchanged.[20] Here the riggings $\sharp$ are taken so that these strings again become singular, namely, they are chosen to be the respective vacancy in the new environment $\mathcal{B} \otimes B^{k-1,1}$. When $\mathrm{rk}(\mu, r) = k$ in particular, we have no string to replace, hence $\delta(\mu, r) = (\mu, r)$. It is known that $\delta(\mu, r) \in \mathrm{RC}(\mathcal{B} \otimes B^{k-1,1}, \lambda - \mathbf{e}_{\mathrm{rk}(\mu,r)})$.

*Algorithm for $\Phi$.* Let $\mathfrak{p} \in \mathcal{P}_+(\mathcal{B} \otimes B^{k-1,1}, \lambda - \mathbf{e}_d)$ be a highest path in the bottom left corner of (C.18) for some $d \in [1, n]$. Let further $\mathfrak{p}' \in \mathcal{P}_+(\mathcal{B} \otimes B^{k,1}, \lambda)$ be the highest path such that $\mathrm{rb}(\mathfrak{p}') = \mathfrak{p}$. Then (C.19) tells that $\mathfrak{p}'$ is obtained from $\mathfrak{p}$ by adding a box containing $d$ to its 'bottom right' component. (Note that $d \geq k$ holds due to the semistandardness of the rightmost component tableau of $\mathfrak{p}'$.) Given such $k \in [1, n-1], d \in [1, n]$ and a rigged configuration $(\mu, r) = ((\mu^{(1)}, r^{(1)}), \ldots, (\mu^{(n-1)}, r^{(n-1)})) = \Phi(\mathfrak{p})$, we are going define a new one $(\mu', r') = \Phi(\mathfrak{p}')$ on the top right corner of (C.18). This enables us to go backward vertically in (C.18) and to translate the growth of the paths into that of rigged configurations starting from $\Phi(\emptyset) = \emptyset$. Here the growth means $w_1, w_1 w_2, w_1 w_2 w_3, \ldots$ in the notation of (C.2).

(i)' Set $\ell^{(d)} = \infty$. Do the following procedure for $a = d-1, d-2, \ldots, k$ in this order. Find the longest color $a$ singular string $(a, l, p_l^{(a)})$ in $(\mu^{(a)}, r^{(a)})$ such that $l \leq \ell^{(a+1)}$ and continue with $a-1$.[21] If there is no such string, set $\ell^{(a)} = 0$ and continue with $a-1$.[22]

(ii)' Let $(d-1, \ell^{(d-1)}, *), (d-2, \ell^{(d-2)}, *), \ldots, (k, \ell^{(k)}, *)$ be the so found singular, or 'length 0' strings, where $*$ denotes the rigging equal to the respective vacancy.[23]

(iii)' The new rigged configuration $(\mu', r')$ such that $\delta(\mu', r') = (\mu, r)$ is obtained by replacing them with $(d-1, \ell^{(d-1)}+1, \sharp), (d-2, \ell^{(d-2)}+1, \sharp), \ldots, (k, \ell^{(k)}+1, \sharp)$ keeping all the other strings unchanged.[24] Here riggings $\sharp$ are taken so that these strings become singular, namely, they are chosen to be respective vacancy in the new environment $\mathcal{B} \otimes B^{k,1}$. When $k = d$ in particular, we do not have any string to replace, hence $(\mu', r') = (\mu, r)$.

**Example C.1.** Let us show $\Phi(\mathfrak{p}_0) = S_0$, where $\mathfrak{p}_0$ is given by (C.6) and $S_0$ is by (C.11) or graphically (C.12). We utilize the detailed representation (C.13) and display how $\mathfrak{p}_0 = \Phi^{-1}(S_0)$ is constructed by applying $\delta$ successively. In each line, $\mathrm{rk}(\mu, r)$ is calculated and it is newly inscribed in the box of the path in the next line which is just the removed one in $\mathcal{B}$ by rb. The leftmost data $\mathcal{B}$ corresponds to the blank boxes of the path which are yet to be determined. The image of $\Phi^{-1}$ of the rigged configuration on each line is the part of $\mathfrak{p}_0$ (C.6) that corresponds

---

[19]If there are more than one such strings, pick any one of it.

[20]The selected strings with length $\ell^{(c)} = 1$ (if any) are to be just removed.

[21]If there are more than one such strings, pick any one of it.

[22]Once this happens, the rest becomes $\ell^{(a)} = \ell^{(a-1)} = \cdots = \ell^{(k)} = 0$ by the definition.

[23]In case $\ell^{(a)} = 0$, $*$ is undefined but it does not matter.

[24]When $\ell^{(a)} = 0$, replacing $(a, \ell^{(a)}, *)$ by $(a, \ell^{(a)}+1, \sharp)$ is to create a color $a$ length 1 singular string.

to the unfilled boxes of the path on the same line.

$$\tag{C.20}$$

After this, we will only get $\mathrm{vac}^{\otimes 3}$ and end up with $\mathfrak{p}_0$ in (C.6).

Let us finish with a remark on the situation of our cBBS. The states are elements of $B^{\otimes L}$ of the form $\mathrm{vac}^{\otimes j-1} \otimes b_j \otimes \cdots \otimes b_l \otimes \mathrm{vac}^{\otimes L-l}$ satisfying the boundary condition $1 \ll j \ll l \ll L$. Taking $j \gg 1$ assures the highest condition (C.3) automatically for any $b_j \otimes \cdots \otimes b_l$. So we can consider the image of the BBS states by $\Phi$. From the definition of $B$ in (2.21), the vacancy (C.10) becomes (3.3). Thus $l \ll L$ implies $\forall p_j^{(a)} \gg 1$, which allows us to increase the rigging of $\Phi$(state) without breaking the condition (C.8). As the above example indicates, if

$\Phi(\mathfrak{p}) = (\mu, r) \in \mathrm{RC}(B^{\otimes L}, \lambda)$, supplement of the vacuum tail does not influence the rigged configuration in the sense that $\Phi(\mathfrak{p} \otimes \mathrm{vac}^{\otimes l}) = (\mu, r) \in \mathrm{RC}(B^{\otimes L+l}, \lambda + ((n-1)l, (n-2)l, \ldots, l, 0))$. Similarly, supplement of the vacuum in front just causes a uniform shift of the rigging $\Phi(\mathrm{vac}^{\otimes l} \otimes \mathfrak{p}) = (\mu, r + l) \in \mathrm{RC}(B^{\otimes L+l}, \lambda + ((n-1)l, (n-2)l, \ldots, l, 0))$, where $r + l$ means that the rigging of every string gets increased by $l$.

# Acknowledgments

A.K. is supported by Grants-in-Aid for Scientific Research No. 16H03922, 18H01141 and 18K03452 from JSPS.

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
