# Peer review of "Generalized hydrodynamics in complete box-ball system for $U_q(\widehat{sl}_n)$"

_SciPost Physics, doi:SciPost Phys. 10, 095 (2021)_

## Round 1 · Referee Report · Anonymous (Referee 5) · 2020-12-21

Strengths
1- a new integrable cellular automaton with nice scattering property is introduced
2- comprehensive analytical and numerical study of the model
3- very accurate and comprehensive numerical verification of generalised hyrdodynamics
Weaknesses
1- slightly technical
Report
In this paper, the authors study a new cellular automaton called the complete box-ball system (cBBS). This is a model (in fact, this is a family of models) is based on the $U_q(\hat{sl}_n)$ algebra, and is a particular case of a large family of models that has been considered the literature. The cBBS itself has not been studied before. It is shown by the authors to have very specific and appealing dynamical properties, the main one being the fact solitons exist and scatter in a diagonal fashion. The existence of solitons in other BBSs has been seen, but this seems to be the first time a BBS is found to have diagonal scattering.
This offers a very good playground where non equilibrium dynamics can be studied. The TBA and GHD for this model is constructed and compared with numerical simulations. The agreement with GHD for the partitioning protocol is strikingly good, both for the Euler and diffusive GHD predictions. This is not only an interesting new integrable model to study, but also gives one of the most accurate numerical verifications of GHD that has been done until now.
The paper is well written. The model is very involved and its construction relatively technical, but the authors have done a very good work in explaining it and putting it in the context of the box-ball systems. The GHD is well constructed, and the numerical analysis is excellent.
I do not have any particular request concerning the paper, and I believe it can be published as it is.
But on the science side, the one thing I am curious about is what is the correction to the diffusive behaviour seen in the numerics. It is mentioned that the diffusive GHD prediction is not quite reached by the numerics, but that the error decreases. How does it decrease? With a power law? Given how accurately GHD can be recovered, this would be an interesting thing to observe in order to guide future work on corrections beyond diffusive. Maybe the authors can make a quick remark about this?
Small thing:
Page 3: the dot before “non diagonal”?
monotonous -> monotonic
Anonymous on 2020-11-22 [id 1055]
I have only some general comments on the paper.
The box-ball system is a rather degenerate classical integrable many-body
system, which has been mostly studied in Japan, but also more recently by Pablo
Ferrari, who is a well known probabilist. The submitted paper comes with a new twist,
namely internal degrees of freedom, which for computational reasons are taken to form
a quantum group. The authors have a previous recent paper on the standard box-ball with
no internal degrees of freedom. The interesting and novel approach is to use the box-ball
system as a testing ground for generalised hydrodynamics. They derive the equations and
arrive at predictions for particular initial data of domain wall type. The results are
compared with extensive numerical simulations
of the box ball system.
The study is timely and of considerable interest, by teaching us more about GHD in
the context of a concrete model. I strongly support publication.
The paper contains rather lengthy computations, which are unavoidable. But I am
not in a position to check the details. I suggest to ask T. Prosen. He has Ph.D. and
more advanced collaborators, which are well experienced in classical cellular
automata. They might be interested to understand some details of the
submitted paper. An additional option is Makiko Sasada, who is more on
the mathematical side (in this case presumably welcome). She is an expert
on box-ball.

---

## Round 1 · Referee Report · Anonymous (Referee 4) · 2021-1-12

Strengths
1-Introduces a new $sl_n$ generalisation of the original box-ball system that is both natural and considerably simpler than the existing generalisation. In particular the soliton S-matrix is diagonal, involving only phase shifts unlike the earlier generalisation.
2-The simplicity of this model is exploited in developing both the GGE-TBA and GHD analysis of the model.
3-The numerical simulation presents compelling evidence for the validity of the GHD description of the plateau arising in the n=3 domain wall problem.
Weaknesses
1-The presentation of the numerics could benefit from polishing.
2-It would be nice to see a comparison of the diffusive correction predicted in Section 5.2 with the observed broadening of the plateaus in the numerical simulation.
Report
The paper revisits the $sl_n$ generalisation of the Takahashi-Satsuma Box-Ball model. It develops a simpler and in many ways more natural generalisation than the existing one in the literature (described for example in [5]). As in [5] the time-evolution operator is formulated in terms of $sl_n$ combinatorial R-matrices. However, in the current work the local quantum-space representations are changed and given by (1.2). The soliton description of the full quantum space is then elegantly described in terms of fixed $sl_n$ colour solitons of different lengths/amplitudes. These solitons have the particular nice property that their factorised S-matrix is diagonal - unlike the considerably more complex non-diagonal S-matrices of the previous $sl_n$ model. The S-matrix itself is very simple and is given by (2.40).
To digress:
this new $sl_n$ model should be useful in many contexts where a higher-rank quantum integrable system with a simple and very explicit algebraic description is needed. In particular, it will be interesting to see how the parallel 'classical' description of this model works (that is the description of the latter Chapters of [5], including the associated ultra-discrete classical equation, tropical spectral curve, Jacobian, etc ). Hopefully the authors and their collaborators will develop this picture.
In the current paper the simplicity of the model is exploited in order to develop the GGE-TBA and GHD descriptions of the $sl_n$ model. I am not an expert in either of these approaches, but the presentation is clear and I can follow it to see how the plateau description given at the end of Section 5.2 arises.
In the final Sections of the paper, the authors carry out a numerical simulation of the $n=3$ model with 'domain wall' conditions and compare to the GHD description. The results themselves are qualitatively convincing - in that there is a clear coincidence of the predicted GHD plateaus and the smoothed plateaus of the numerics. I have some minor reservations about the presentation style of the numerics that I point out below. A more substantive point is that the numerical plateau are smoothed, and the obvious question is whether this smoothing can be described by the diffusive corrections to the ballistic description already developed in Section 5.2.
Overall I think this is a very nice paper, and that the $sl_n$ generalisation of the Ball-Box model is a useful contribution to the field in itself. Of course the paper goes way beyond this and develops the GHD description of the model and compares with numerics. The consistency with the numerics provides more convincing evidence of the validity of the GHD description of such quantum integrable systems.
Requested changes
1-In Section 5.3, I can't see the difference between the different $t$ plots (even on my 'retina' display). While I understand that this is the point - it seems like bad practice to graphically represent two sets of data in a way where you can't properly see either because of overlap with the other. I leave it to the authors to consider if they can solve this issue.
2- Minor point, but the figures are referred to variously as fig 1, Fig. 1, Figure 1. Please be consistent.
3- There seems to be a problem with the ordering of figures: the discussion of Figure 10 comes directly after the discussion of Figure 6. Please fix.
4- In Section 6 it is stated that '[...] the simulations match perfectly the GHD expectations'. Either make this quantitive (with a percentage error) or don't state it.
5-Section 6 gives a nice summary of what is a fairly long and technical paper. It would help with the readability of the paper if the authors could cross-reference (i.e. include equation numbers for) the earlier results in the paper they are referring to in this concluding section.
6-I think there deserves to be some discussion of the role of the diffusive corrections presented in Section 5.2 in the comparison with the numerics.

---

## Round 1 · Referee Report · Anonymous (Referee 3) · 2021-1-18

Strengths
- Presents a novel and very rich integrable model with an arbitrary rank underlying symmetry algebra, yet the model is simple and has only a finite spectrum of velocities of solitons
- The model has a very rich set of commuting dynamical automorphisms
- Inverse scattering formulated in a very explicit manner
- GHD equations of the model take remarkably simple form in terms of Y system of TBA
Weaknesses
- Some terminology is a bit unusual for statmech (see below)
- English needs a bit of polishing
Report
This is a remarkable paper!
It proposes a rich family of models, the so-called complete box-ball systems, whose local phase space is defined on the product set of column semistandard Young tableauxs, and whose underlying symmetry algebra is given b U_q(sl_n).
The elementary solution excitaitons can be completely classified and shown to undergo purely diagonal scattering, a feature which largely simplifies hydrodynamic treatment of the model. The deterministic local time evolution map is given in terms of combinatorial R satisfying a Yang-Baxter equation with a discrete version of a spectral parameter. This is a beautiful discrete space-time-state version of a higher-rank integrable theory, and can serve either as a playground for developing nonequilirbium statistical mechanics of integrable systems, or as a cenceptually new kind of integrable systems.
Despite being very technical, the paper is not too hard to follow. It is clearly structured and quite well written. I enthusiastically recomment its publication in SciPost.
Requested changes
1- Is comutative diagram (2.48) completely correct, namely the two verstical mappingfs T^{(k)}_l are not identical. One is a conjugation of the other by the map \Phi. See also last in-text formula on the page 16.
2- maybe notation on the RHS of (2.52) has to be explained?
3- I think it is an awkard nomenclature to call cBBS ystem’s configurations as “states”, and then treatment of distributions over configurations - which in the more standard language of statistical mechanics would correspond to “states” - as “randomized cBBS”. Maybe the authors can rethink that, but my suggestion would be to use “configurations”, for the former, and “states” for the later, while the system (cBBS) has no intrinsic randomness (it is deterministic in both cases) so it does not make sense to call it “randomized cBBS” in case where statistical ensembles of cBBS configurations are considered.
4- There were other integrable deterministic cellular automata in recent literature, where the interplay between ballistic and diffusive transport has been established, even beyond the hydrodynamic assumtptions. See e.g. Commun. Math. Phys. 371, 651 (2019), or PRL 119, 110603 (2017). I don't expect that such explicit results could be achieved for cBBS system, but perhaps the authors can comment on this and/or make make an appropriate link to this relevant literature.

---

## Round 2 · Referee Report · Anonymous (Referee 1) · 2021-2-16

Report

The answers from the authors are satisfying and I think the manuscript should be published.

---

## Round 2 · Referee Report · Anonymous (Referee 2) · 2021-2-26

Strengths

See original report

Weaknesses

See original report

Report

I am satisfied that the authors have seriously addressed all the minor issues raised in my earlier report in their revised version. I am happy to recommend publication of the paper in its current form.

---

## Round 2 · Author Response

We thank all the referees for their positive, detailed and constructive reports. We have taken their suggestions into account in the new version of the manuscript. We answer all the comments and suggestions just below as well as in the ‘List of changes‘ box.

[Referee #1] But on the science side, the one thing I am curious about is what is the correction to the diffusive behaviour seen in the numerics. It is mentioned that the diffusive GHD prediction is not quite reached by the numerics, but that the error decreases. How does it decrease? With a power law? Given how accurately GHD can be recovered, this would be an interesting thing to observe in order to guide future work on corrections beyond diffusive. Maybe the authors can make a quick remark about this?

The shape of the transition between two consecutive plateaus is analysed at the end of Sec 5.3. In Fig. 10 the results of the simulations are compared to the error functions that are predicted by diffusive corrections to GHD (Sec 5.2). The agreement is good in all cases, but, as pointed out by the referee, we observed one case where the agreement is quantitatively not as good (but it improves as time grows). As written at the end of Sec 5.3 the reason for such behaviour is not clear to us. The referee’s suggestion to carry a quantitative analysis of the convergence is certainly good. One reason why this has not yet been done is the fact that a reliable analysis would presumably require even more precise simulation results (increased number of initial conditions) while the numerical data presented in Fig. 10 already represent some rather significant amount of CPU time (~10^16 applications of combinatorial R and thus several months of CPU time).

[Referee #2] In Section 5.3, I can't see the difference between the different t plots (even on my 'retina' display). While I understand that this is the point - it seems like bad practice to graphically represent two sets of data in a way where you can't properly see either because of overlap with the other. I leave it to the authors to consider if they can solve this issue.

Presenting a large quantity of numerical data carrying a lot of information in a few plots can often be tricky. For the problem raised by the referee one option would be to keep a single value of t in each plot, but this would no longer show the readers that the data actually overlap, which is an important result. In addition we stress also that the curves do not overlap everywhere since they visibly differ in the steep regions of each step, and this illustrates the diffusive broadening. A common practice in this kind of situation is to shift vertically the curves associated to different values of t. But in the present case, because of the “thickness” of the signal (due to the fluctuations effect) this does not seem doable in practice (it would require very large shifts to separate the curves, and they would no longer fit inside each panel). For the above reasons we do not see any satisfactory way to improve the presentation of the data.

[Referee  #3] I think it is an awkward nomenclature to call cBBS ystem’s configurations as “states”, and then treatment of distributions over configurations - which in the more standard language of statistical mechanics would correspond to “states” - as “randomized cBBS”. Maybe the authors can rethink that, but my suggestion would be to use “configurations”, for the former, and “states” for the later, while the system (cBBS) has no intrinsic randomness (it is deterministic in both cases) so it does not make sense to call it “randomized cBBS” in case where statistical ensembles of cBBS configurations are considered.

In fact, state is used for single microscopic configurations everywhere except in Sec. 5, and we believe that the precise meaning is always quite clear from the context. Still, we have added a comment (footnote #12) at the beginning of Sec. 5 to warn the readers that, in this section, state should be understood in a thermodynamic (or probabilistic) sense. We have removed the expression “randomize cBBS” from the abstract and from the beginning of the introduction. It is then used for the first time in page 4, where we have added a sentence to stress that randomized refers to the initial conditions, and that the dynamics remains fully deterministic. We also note that “randomized BBS” has already been used several times in the literature.

---

## Round 2 · List of Changes

[Referee #1] the dot before “non diagonal”? Corrected.

[Referee #1] monotonous -> monotonic

Corrected.

[Referee #2] (...) the figures are referred to variously as fig 1, Fig. 1, Figure 1. Please be consistent.

Done, thanks.

[Referee #2] There seems to be a problem with the ordering of figures: the discussion of Figure 10 comes directly after the discussion of Figure 6. Please fix.

The (previously numbered) figure 10 has been moved after Fig. 6. So the old Fig. 10 is now Fig. 7.

[Referee #2] In Section 6 it is stated that '[...] the simulations match perfectly the GHD expectations'. Either make this quantitative (with a percentage error) or don't state it.

Since the quality of the agreement between the simulations and the GHD theory is quite obvious in all figures, we have not carried out a quantitative estimate of the error. We have therefore removed “ match perfectly” and have replaced it with the weaker statement “are in good agreement with”.

[Referee #2] Section 6 gives a nice summary of what is a fairly long and technical paper. It would help with the readability of the paper if the authors could cross-reference (i.e. include equation numbers for) the earlier results in the paper they are referring to in this concluding section.

Done.

[Referee #2] I think there deserves to be some discussion of the role of the diffusive corrections presented in Section 5.2 in the comparison with the numerics.

The calculation of the diffusive corrections described in Sec. 5.2 is indeed directly relevant to the broadening of the steps between plateaux, as observed in the numerics and in Fig. 10 in particular. This is in fact already discussed in the paper. We have added one comment at the end of Sec. 5.2 to indicate that the error-function form (and the quantity \Sigma) is compared quantitatively to the simulation data at the end of Sec. 5.3.

[Referee #3] Is commutative diagram (2.48) completely correct, namely the two vertical mappings T^{(k)}_l are not identical. One is a conjugation of the other by the map \Phi. See also last in-text formula on the page 16.

The referee is right, formally the two vertical mappings are different functions. However, we feel that it would be unnecessarily heavy to introduce here a new notation here, since the two mappings are naturally identified. We have added a footnote (called just above the diagram (2.48)) to indicate that the same name is used for two formally distinct objects.

[Referee #3] Maybe notation on the RHS of (2.52) has to be explained?

We have added a sentence below (2.52) to explain that the graphical (box) notation is explained in the appendix C.2.

[Referee #3] There were other integrable deterministic cellular automata in recent literature, where the interplay between ballistic and diffusive transport has been established, even beyond the hydrodynamic assumptions. See e.g. Commun. Math. Phys. 371, 651 (2019), or PRL 119, 110603 (2017). I don't expect that such explicit results could be achieved for cBBS system, but perhaps the authors can comment on this and/or make make an appropriate link to this relevant literature.

We thank the referee for drawing our attention to these two interesting references. They are now cited in the revised manuscript (in Sec. 6).

---

## Editorial Decision

published